# Cyanine dyes in the mitochondria-targeting photodynamic and photothermal therapy
Zdeněk Kejík [1,2] ✉, Jan Hajduch[1,2], Nikita Abramenko[1,2], Frédéric Vellieux [1,2], Kateřina Veselá[1,2], Jindřiška Leischner Fialová[1], Kateřina Petrláková[3], Kateřina Kučnirová[1,2], Robert Kaplánek[1,2], Ameneh Tatar [1,2], Markéta Skaličková[1,2], Michal Masařík[1,2,3,4], Petr Babula[4], Petr Dytrych[5], David Hoskovec[5], Pavel Martásek [2] ✉ & Milan Jakubek [1,2] ✉

Mitochondrial dysregulation plays a significant role in the carcinogenesis. On the other hand, its destabilization strongly represses the viability and metastatic potential of cancer cells. Photodynamic and photothermal therapies (PDT and PTT) target mitochondria effectively, providing innovative and non-invasive anticancer therapeutic modalities. Cyanine dyes, with strong mitochondrial selectivity, show significant potential in enhancing PDT and PTT. The potential and limitations of cyanine dyes for mitochondrial PDT and PTT are discussed, along with their applications in combination therapies, theranostic techniques, and optimal delivery systems. Additionally, novel approaches for sonodynamic therapy using photoactive cyanine dyes are presented, highlighting advances in cancer treatment.

Despite advancements in diagnostic techniques, medical research, therapies, and technologies, cancer remains one of the most challenging issues[1,2]. Recent studies have increasingly focused on the role of mitochondrial dysregulation in carcinogenesis and its potential as a therapeutic target. As the cell's powerhouses, mitochondria are essential for various cellular processes, including energy metabolism, signal transduction, and apoptosis[3]. Their dysfunction can lead to increased metastatic potential and therapy resistance in cancer cells. However, targeting mitochondrial biogenesis and functionality presents a promising avenue for cancer treatment[4,5].

Photodynamic and photothermal therapies (PDT and PTT) have emerged as effective treatment modalities that exploit the unique properties of photosensitizers (PS) to induce cancer cell death. Cyanine dyes, known for their strong tumor selectivity and mitochondrial accumulation, show great promise in enhancing the efficacy of PDT and PTT[6–9]. These dyes not only target cancer cells with high precision but also induce potent anticancer effects when activated by specific wavelengths of light. The dual capability of cyanine dyes to function in both PDT and PTT makes them versatile tools in cancer treatment. It is noteworthy that mitochondria are particularly sensitive to PDT, PTT, and especially to their combination[10]. Beyond effectively eliminating cancer cells and reducing tumor mass, this therapeutic approach can also stimulate the immune system and inhibit the metastatic process[11–13].

This comprehensive review investigates the potential and limitations of cyanine dyes in mitochondria-targeted PDT and PTT. It explores their applications in combination therapies, theranostic techniques, and optimal delivery systems. Furthermore, novel methods in sonodynamic therapy using photoactive cyanine dyes are highlighted, illustrating innovative advances in cancer treatment. The discussion is enriched with numerous examples that demonstrate the effectiveness of cyanine dyes as mitochondria-targeting photosensitizers, emphasizing their crucial role in the advancement of cancer therapeutics.

"The role of mitochondria in biological processes" briefly introduces mitochondrial functionality and its role in carcinogenesis. "Basic principle of the photodynamic and photothermal therapy" describes the basic mechanisms of photodynamic and photothermal therapy. "Cyanine dyes" introduces cyanine dyes, with the first subsection discussing their usability in mitochondrial targeting. "Mitochondrial targeting in PDT" and "Mitochondrial Targeting in PTT" discuss the principles and specific mechanisms of mitochondrial targeting in photodynamic and photothermal therapy, respectively.

[1]BIOCEV, First Faculty of Medicine, Charles University, 252 50 Vestec, Prague, Czech Republic. [2]Department of Paediatrics and Inherited Metabolic Disorders, First Faculty of Medicine, Charles University and General University Hospital in Prague, Ke Karlovu 455, 120 00 Prague, Czech Republic. [3]Department of Pathological Physiology, Faculty of Medicine, Masaryk University, Kamenice 5, CZ-625 00 Brno, Czech Republic. [4]Department of Physiology, Faculty of Medicine, Masaryk University, Kamenice 5, 625 00 Brno, Czech Republic. [5]1st Department of Surgery-Department of Abdominal, Thoracic Surgery and Traumatology, First Faculty of Medicine, Charles University and General University Hospital in Prague, U Nemocnice 2, 121 08 Prague, Czech Republic. ✉e-mail: Zdenek.Kejik@lf1.cuni.cz; Pavel.Martasek@lf1.cuni.cz; Milan.Jakubek@lf1.cuni.cz

In both sections, the "Multifunctional photodynamic cyanine dyes" illustrates and discusses photodynamic and photothermal cyanine dyes with mitochondrial selectivity, as well as describing their potential theranostic applications. "Combination of PDT and PTT" focuses on the combination of photodynamic and photothermal therapy in the context of mitochondrial targeting, with the "Mitochondria-targeted dual photodynamic and photothermal cyanine dyes" presenting and discussing cyanine dyes with dual photodynamic and photothermal activity. "PDT and PTT in the combination therapy" focuses on the usability and design of photodynamic and photothermal therapy in combination therapy within the context of mitochondrial targeting, with the "PDT and PTT in the combination therapy in the context mitochondrial targeting" presenting individual agents that affect mitochondrial functionality. "Delivery system" introduces the role of nanoparticles for drug delivery of cyanine dyes in the context of mitochondrial targeting. "Self-assembly nanoparticles," "Liposomes," "Polymeric micelles," "Biopolymer," and "Inorganic nanoparticles" present and discuss in detail the types of nanoparticles used for the targeted transport of phototoxic cyanine dyes (self-assembly, liposomes, polymeric micelles, biopolymers, and inorganic nanoparticles, respectively). "Sonodynamic therapy—novel therapeutic applicability of phototoxic cyanine dyes" describes novel applications of sonodynamic therapy with cyanine dyes. "Future direction" discusses possible novel strategies in the design and application of phototoxic cyanine dyes in the context of mitochondrial targeting.

## The role of mitochondria in biological processes

Mitochondria are essential organelles in eukaryotic cells, playing key roles in a diverse range of biological processes, including energy metabolism, signal transduction, and cell survival[14,15]. They consist of five distinct parts: two separate membranes (outer and inner) with characteristic phospholipid composition, the intermembrane space, cristae (folds in the inner mitochondrial membrane) and the matrix. Mitochondria are semi-autonomous, containing their own mitochondrial DNA (mtDNA), which allows them to replicate, transcribe, and translate independently of nuclear DNA (nDNA)[16]. Unlike the much larger nDNA (which has billions of base pairs in humans),

mtDNA is small (16,569 bp)[17,18], circular and features hypomethylated CpG motifs. mtDNA codes 37 genes; 22 tRNAs, 2 rRNAs, and 13 polypeptides that are components of complexes I (CI), III (CIII), IV (CIV), and V (CV) of the respiratory chain. These polypeptides include 7 subunits of CI (NADH dehydrogenase), cytochrome b (a main component of CIII), 3 subunits of CIV (cytochrome c oxidase), and 2 units of CV (ATPase). The noncoding region of mtDNA, called displacement-loop (D-loop), regulates mtDNA replication and maintenance. All other mitochondrial proteins and components of the respiratory chain are encoded by nDNA.

Given the critical functions of mitochondria, their dysfunctions, and altered activity are implicated in numerous diseases. For instance, higher ATP levels in cancer cells can be associated with stem-like phenotype (multidrug resistance, invasiveness, and spontaneous metastasis; Fig. 1)[19]. Additionally, higher mitochondrial activity and mitochondrial biogenesis in African American patients correlate with worse therapeutic prognoses (higher cancer mortality rates, and shorter survival times) across multiple cancer types compared to European American patients with lower mitochondrial activity[3].

Studies confirm that African American patients generally have higher cancer mortality rates and shorter survival times compared to European American patients, across multiple cancer types including breast, prostate, colorectal, lung, pancreatic, liver, cervical, multiple myeloma, stomach, ovarian and esophageal cancer[20-26]. Piyarathna et al. have identified a group of genes that are upregulated in various tumor types in African American cancer patients compared to European American patients[3]. These genes are linked to enhanced oxidative phosphorylation and the upregulation of transcription factors that promote mitochondrial biogenesis, leading to an increased number of mitochondria in tumor samples from African American patients. These findings suggest that mitochondrial dysfunction may contribute to the higher cancer incidence and poorer outcomes observed in African American patients[27].

Mitochondrial oxidative phosphorylation is the main cellular producer of reactive oxygen species (ROS). Some electrons escape from the mitochondrial respiratory complexes I and III, directly reacting with oxygen to generate the superoxide anion radical[28]. The production of ROS is essential

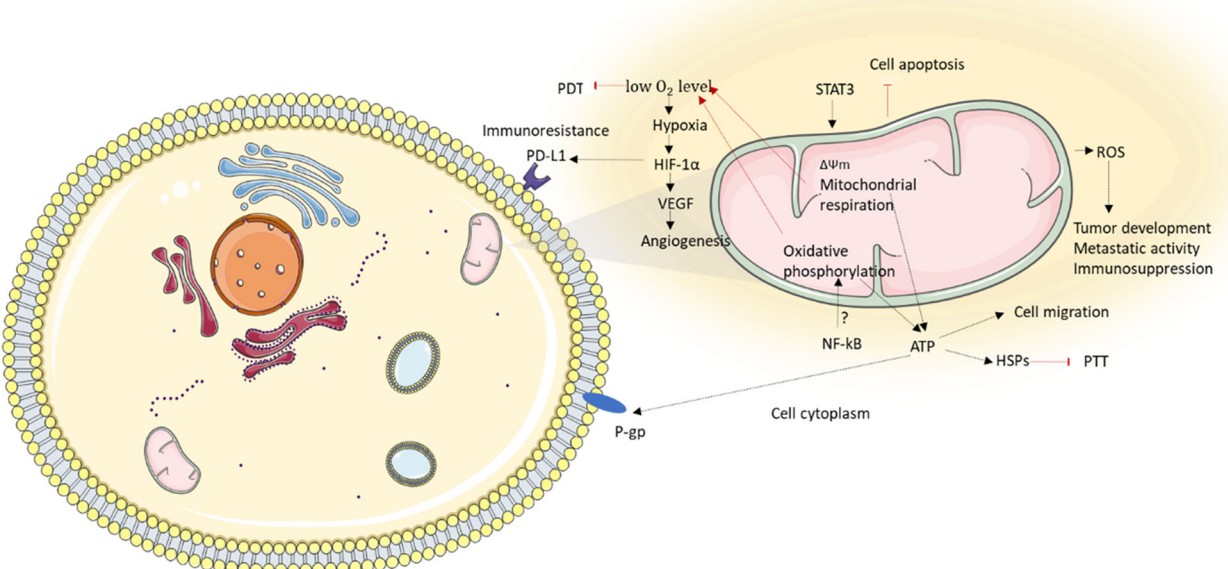

**Fig. 1 | Influence of mitochondrial respiration on the carcinogenesis.** Increased activity of STAT3 and NF-κB signaling can enhance mitochondrial respiration and oxidative phosphorylation, leading to increased ATP production. Elevated ATP levels support cell migration and the activity of P-gp and HSPs, which can reduce the effectiveness of chemotherapy and PTT, respectively. Higher $\Delta\Psi m$ can promote the production of reactive oxygen species (ROS), contributing to tumor development, metastatic activity, and immunosuppression. Heightened mitochondrial activity may also lead to decreased oxygen levels, initiating hypoxia. HIF-1α (activated by hypoxia), which in turn induces the expression of numerous tumorigenic factors such as VEGF (a stimulator of angiogenesis) and PD-L1 (protection against immune cells). The figure was partly generated using Servier Medical Art, provided by Servier, licensed under a Creative Commons Attribution 3.0 unported license. $\Delta\Psi m$, mitochondrial membrane potential; HIF-1α, hypoxia inducible factor 1 subunit alpha; HSP, heat shock protein; NF-κB; Nuclear factor NF kappa B; PD-L1, programmed death-ligand 1; PDT, photodynamic therapy; P-gp, P-glycoprotein; PTT, photothermal therapy; ROS, reactive oxygen species; STAT3, signal transducer and activator of transcription 3; VEGF, vascular endothelial growth factor.

**Fig. 2 | Jablonski's diagram for the PDT and PTT.** Upon light (photon) absorption, PS transitions from $S_0$ to the first excited $S_1$. The excited PS can return to the $S_0$ through fluorescence emission or vibrational relaxation, which can generate heat and potentially cause thermal damage of tissue and cells. Another possibility is the transition of the PS to the $T_1$, where the PS can interact with molecular oxygen $^3O_2$ and form highly reactive singlet oxygen $^1O_2$ in a type II reaction. In a type I reaction, ROS are produced via electron transfer from the PS, which is subsequently reduced. In both cases, ROS can lead to oxidative damage in cells. However, the PS can also return to $S_0$ via phosphorescence emission. $^1O_2$, singlet oxygen; $^3O_2$, molecular oxygen; $^1O_2$, singlet oxygen; PDT, photodynamic therapy; PTT, photothermal therapy; PS, Photosensitizer; ROS, reactive oxygen species; $S_0$, ground state; $S_1$, first singlet excited state; $T_1$, triplet excited state.

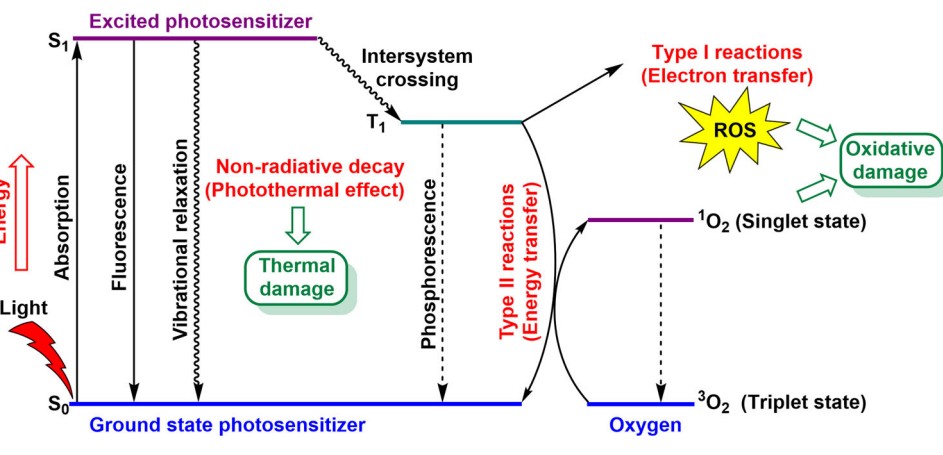

for physiological processes within cells, as they act as signaling molecules. In cancer, ROS play a dual role, capable of both promoting and inhibiting malignant behavior. The cellular response to ROS critically depends on their levels. Elevated ROS have been shown to contribute to higher viability and invasiveness of cancer cells[29]. On the other hand, unregulated ROS production and accumulation lead to various forms of cell death[30,31]. In general, cancer cells have higher levels of ROS, contributing to mutagenesis and ultimately tumor progression[32]. Nevertheless, higher ROS levels make cancer cells more susceptible to the ROS-induced treatments, as their ability to balance distribution, accumulation and detoxification capacities is limited[28]. Many anticancer drugs exploit ROS-induced cell death as their mechanism of action[33,34]. Essentially, cancer cells necessitate elevated ROS levels compared to normal cells. However, excessively high ROS levels induced by certain therapies such PDT can effectively eliminate cancer cells[28].

Recent research has shown that metastatic tumors display active mitochondrial respiration, which significantly promotes tumor growth and metastatic activity[35–37]. On the other hand, some metastatic factors, such as signal transducer and activator of transcription 3 (STAT3) and nuclear factor kappa B (NF-κB), can directly affect mitochondrial functionality[21,38]. STAT3 supports mitochondrial membrane potential and maintenance. The role of NF-κB is not fully understood, and published results depend on the experimental condition and cell type[21]. Nevertheless, it was reported, that NF-κB has been found to stimulate oxidative phosphorylation in colon cancer cells[39]. STAT3 activity can have a positive impact on ATP production[38]. Higher activity of mitochondrial ATP synthase (depending on tumor type) can be associated with higher metastatic activity[40].

High impact studies suggest that mitochondrial oxidative phosphorylation is significantly involved in the development of drug resistance under conditions of hypoxia and circulation of metastatic cells that depend on mitochondrial respiration[41,42]. In the case of breast cancer patients, higher expression of β-F1-ATPase correlates with higher risk of metastasis and poor survival. Higher mitochondrial ROS production and mitochondrial membrane potential ($\Delta\Psi m$) are also associated with loss of therapeutic efficiency[43,44]. Higher activity of antioxidative factors (e.g., superoxide dismutase, glutathione, thioredoxin, and peroxiredoxins) induced by ROS may contribute to the protective effect against anticancer drugs[45–48].

## Basic principle of the photodynamic and photothermal therapy

PDT and PTT rely on the combination of specific photosensitizers (PS) and targeted light irradiation[49,50]. By focusing light directly on specific tissue areas, these therapies effectively concentrate their effects while minimizing side effects. In the absence of light, PS at therapeutic concentrations typically exhibit minimal cytotoxicity, and light alone has no impact on the affected tissue.

The therapeutic efficacy of PDT and PTT is constrained by the light absorption properties of biological tissues. The clinically used excitation wavelength is chosen as a compromise between low tissue absorption and the requirement for sufficient light energy for PDT and/or PTT application. For PDT, the spectral region from 600 to 800 nm (the so-called "first spectral or biological window") is typically used[51], while PTT employs wavelengths between 650–1100 nm[52]. In addition to the first biological window, PS used for PDT can be irradiated within the second biological window (1000–1350 nm) and the third biological window (1550–1850 nm)[53]. Upon photoabsorption, PS transitions from the ground state to an excited singlet state (Fig. 2). This short-lived state may emit gained energy as fluorescence or heat (as utilized in PTT) or it can transit to a more stable triplet state. In this triplet state, PS can engage in type I or type II reactions. In type I reactions, PS produces ROS such as hydroxyl radicals ($^\bullet$OH), hydrogen peroxide ($H_2O_2$) and superoxide anions ($O_2^{\bullet-}$) through electron transfer[54,55]. These ROS are further involved in biochemical reactions, like the Fenton reactions (catalyzed by $Fe^{2+}$ ion), which generate highly reactive hydroxyl radicals[56]. In type II reactions, PS interacts with molecular oxygen ($^3O_2$) to form $^1O_2$ (singlet oxygen) causing oxidative damage. Singlet oxygen can further produce other types of ROS. This mechanism is expected for the majority of PS[57]. Higher levels of ROS can cause oxidative damage of biomolecules (e.g., proteins, nucleic acids, and lipids), dysregulation of the redox homeostasis, and subsequently cell death.

The type I reaction mechanism initiates the production of $O_2^{\bullet-}$ by electron transfer from PS in the triplet state (monovalent reduction)[54,55]. $O_2^{\bullet-}$ is converted to $H_2O_2$ by superoxide dismutase. In the Fenton reaction (catalyzed by $Fe^{2+}$ ion), $H_2O_2$ is decomposed into $^\bullet$OH and OH$^-$[56]. $O_2^{\bullet-}$ can also react with $^\bullet$OH, or NO, and form $^1O_2$ and peroxynitrate (OONO$^-$), respectively. In type I reaction, PS in the triplet state can also directly interact with an organic molecule from its surroundings, such as lipids (in the case of cellular membranes), and bind a hydrogen atom or electron to form a radical[58]. Conversely, organic radicals can react with oxygen to form ROS[59]. However, type I reactions often lead to more severe damage, and PS is consumed and needs to be resupplied. On the other hand, the initiated radical production can propagate itself and multiply the oxidative damage[60,61]. Moreover, type I reactions show less sensitivity to oxygen levels compared to type II reactions[62,63]. In solid tumors, oxygen levels may be reduced (due to local hypoxia) and therefore the efficiency of PDT, especially of type II reactions, can be limited[62,64].

It should be noted that some cyanine dyes display selective localization in the mitochondrial membrane (which is very sensitive to ROS and good source of organic radicals)[60,61,65–68]. While investigations have also delved

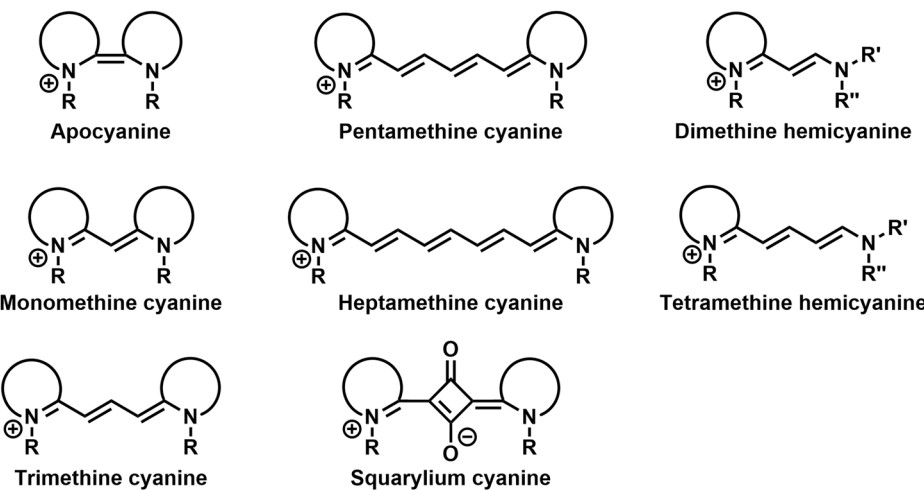

**Fig. 3 | Type of cyanine dyes.** Cyanine dyes are delineated as cationic compounds characterized by two terminal nitrogen aromatic units interconnected by a poly-methine chain. The classification of cyanine dyes is contingent upon the length of the polymethine chain, leading to distinctions such as apocyanine, monomethine, tri-methinine, pentamethine, and heptamethine cyanines. Those cyanine dyes incorporating a squaraine group within the polymethine chain are denoted as squarylium cyanines. Notably, the structural configuration of unsymmetric hemi-cyanines (e.g., dimethine and tetramethine) may encompass solely one nitrogen aromatic unit, setting them apart from their counterparts.

into the targeting of other organelles, notably lysosomes and endosomal systems, for PDT, the intricate relationship between photosensitizer intra-cellular localization and efficacy is extensively examined in the compre-hensive review by Wang et al.[69]. Moreover, notable studies suggest that mitochondrial photosensitizers may exhibit superior efficiency compared to their lysosomal counterpart[70,71] Conversely, cyanine dyes may display localization within lysosomes or co-localization[72,73], emphasizing the need to consider their potential impact on endosomes/lysosomes. In this case, cyanine dyes can enter cells via endocytosis[74], a process where the cell membrane engulfs extracellular material to form vesicles known as endosomes[75]. These endosomes can mature into lysosomes and lysosomal localization of cyanine dye can be observed. In the case of nanoparticles modified with cyanine dyes, cellular uptake occurs through endocytosis, with observable mitochondrial localization only after an extended period[76].

In PTT, the absorbed energy is dissipated through non-radioactive decay, raising the local temperature. Temperatures above 42 °C can lead to the destruction of cancer cells, primarily through heat-induced protein denaturation. The higher the temperature, the more rapid and effective the cell killing becomes; for instance, thermal ablation can occur within a few minutes at temperatures between 48–60 °C[77]. The effectiveness of PTT depends heavily on the precise control of the treatment temperature and the optimal temperature range can vary depending on the desired therapeutic outcome. If the temperature increase is too high, heat shock proteins (HSPs) are activated. HSPs are known as a family of ATP-dependent chaperone molecules that play a diverse role in the regulation of signal transduction, have a protective effect against adverse stressful conditions and can help cancer cells survive under stressful conditions[78,79]. It has also been proven that HSPs play a critical role in initiating the defense mechanism of tumor in thermoresistance[80]. Several HSPs inhibitors have been exploited to reduce thermoresistance in PTT[81–90]. Mild hyperthermia (39-42 °C) induces cel-lular stress and enhances the effects of other treatments like chemotherapy and radiotherapy by increasing tumor cell permeability and blood flow. This temperature range minimizes damage to surrounding healthy tissues and enhances immune responses. Initially, it was thought that hyperthermia aids radiotherapy primarily by enhancing oxygen delivery to tumors, thus reducing radiation-resistant hypoxia, supported by refs. 91–93. Hyper-thermia (42-45 °C) in this range causes denaturation of proteins and damage to cellular structures, leading to apoptosis or necrosis of cancer cells[94]. It can provide a balance between efficacy and safety, being effective in inducing significant cancer cell death while sparing normal tissues if con-trolled precisely. Although lower temperatures may transiently improve

tumor oxygenation and thus enhance radiation response, higher tempera-tures (above 42 °C) can damage vascular structures, leading to increased hypoxia post-treatment, which may paradoxically protect the tumors from radiation, as indicated in ref. 92. Thermal ablation in PTT involves tem-peratures above 48 °C to induce necrosis in tumor cells. However, this can cause damage to surrounding normal tissues due to the heat diffusion. The high temperatures necessary for ablation can also damage adjacent healthy cells, provoking inflammatory responses and other side effects. Therefore, it is crucial to find a balance between effectively destroying the tumor and minimizing damage to healthy tissues[95].

Unlike PDT, PTT's efficacy is less dependent on oxygen levels and the intratumoral distribution of PS, as long as the target temperatures are reached in the desired volume[96]. PTT can be particularly effective in the avascular regions due to the decreased heat-sink effect and improved light tissue penetration caused by lower blood absorption.

An ideal PS or thermal sensitizer should exhibit low dark toxicity and preferentially accumulate in the tumor tissue to minimize side effects. These agents should also have high photostability to resist photobleaching and high absorbance in the 600-850 nm range to balance tissue transparency and photoreaction requirements, especially for producing $^1O_2$. For PDT, the sensitizer should ideally localize in mitochondria and/or lysosomes, avoiding the nucleus to prevent DNA mutations[97].

Selective targeting of cellular organelles, particularly mitochondria, is an emerging strategy in developing novel anticancer agents. Mitochondria represent attractive targets for phototoxic agents due to their susceptibility to oxidative stress and their crucial roles in regulating intracellular Ca$^{2+}$ levels, oxidative stress, survival/apoptotic signaling pathway, cellular dif-ferentiation and the cell cycle[98]. Cyanine dyes, due to their structural motif as hydrophobic cations, are suitable agents for targeting mitochondria.

## Cyanine dyes
The development of mitochondria-specific agents and therapies is therefore one of the hot topics in medical research[36,37,99]. One of most promising mitochondria-targeted agents are cyanine dyes.

Cyanine dyes are a large group of compounds with high structure variability (Fig. 3). Their structural motif includes two terminal heterocyclic units (containing nitrogen atom; e.g., pyrrole, imidazole, thiazole, ben-zothiazole, pyridine, or quinoline) linked by a π-conjugated polymethine chain ending with a nitrogen atom with a positive charge. Depending on the length of polymethine chain, they can be further subdivided into mono-methine cyanine, trimethine cyanine, pentamethine cyanine

(dicarboxycyanine), and heptamethine cyanine (tricarbocyanine), or apocyanines (directly linked heterocyclic units). Some of them (squarylium cyanine) may have cyclic group (squaraine) in the middle of polymethine chain. Hemicyanins (prepared from the heptamethine cycanine), which may contain only one nitrogen heterocyclic unit, can also be classified as cyanine dyes.

Cyanine dyes are promising agents for clinical use due to their inherent merits such as well-defined chemical structure, high purity and good reproducibility. Their synthetic protocols enable the preparation of cyanine dyes as single pure compounds under Good Manufacturing Practice (GMP) conditions with quality control and low production costs.

Cyanine dyes represent a very promising scaffold with high applicability in bioanalytical and medicinal chemistry[6,100,101]. Due to their structural motif (hydrophobic cation), cyanine dyes and their derivatives can be used as specific probes in the recognition of anionic polysaccharides and lipids[102,103], or mitochondrial labelling[65,104]. As some of them show significant selectivity for tumor tissue and cytoselective effect against cancer cells, they are intensively studied as anticancer agents. Specifically, heptamethine cyanine dyes like IR-780, IR-783, and MHI-148 exhibit preferential accumulation in cancer cells[105]. This selectivity is largely attributed to the structural features of these dyes, such as the presence of fused cyclic rings and specific alkyl chains that enhance their hydrophobicity and uptake by cancer cells. These dyes accumulate in tumors due to their binding to serum albumin, which is preferentially accumulated in the tumor tissue[9]. Additionally, modifications like adding ligands allow these dyes to bind to receptors overexpressed on cancer cells (e.g., heparan sulfate and gp130 part of IL-6R)[106,107], further enhancing selectivity[105]. It should be also noticed that tumorigenic transformation is also associated with an increase in the level of anionic phospholipids (typical binding partner of cationic cyanine dyes)[65–67,108,109]. The synergy of the above-mentioned phenomena results in a higher concentration of the dyes within cancerous tissues compared to normal tissues.

Cyanine dyes also show promising photochemical properties such as a broad absorption spectral range with high extinction coefficients and the position of their absorbance band (usually between 600 and 800 nm) corresponding to the phototherapeutic window and are therefore used and studied in the PDT and PTT[7,97,110]. In addition to therapeutic applications, numerous studies have shown that their excellent photophysical properties are also suitable for fluorescence imaging. The living system displays extremely low autofluorescence and low absorbance in the NIR spectral range (700-900 nm)[111]. The use of NIR light in imaging and therapy can significantly decrease background interference and enhance diagnostic usability. For example, indocyanine green (ICG, one of cyanine dyes studied for the PTT and PDT) can also be used and has been approved by FDA approved as a medical imaging agent[112].

On the other hand, most of cyanine dyes have extremely low quantum yield of singlet oxygen[113] and their hydrophobicity/hydrophility ratio may not be optimal. Their poor solubility in water can cause a decrease in their bioavailability and a reduction in the generation of reactive oxygen species (ROS).

Currrently, heptamethine indocyanine green (ICG, FDA-approved for clinical imaging)[114] is also being studied for the photodynamic and photothermal therapy[112]. Nevertheless, this substance displays short half-life, nonspecific plasma binding, optical instability, and poor aqueous stability, which limits its clinical applicability[115]. Therefore, other photodynamic and photothermal agents are being intensively developed[21,97,110].

## Cyanine dyes in the mitochondrial targeting

Cyanine dyes predominantly enter cells through passive diffusion or facilitated transport mechanisms. Since cyanine dyes are lipophilic and cationic, and the cellular membrane (especially in the case of cancer cells) contains anionic receptors such as phospholipids[116], cyanine dye can permeate cellular membranes by dissolving in the lipid bilayer. Nevertheless, other transport mechanism such as endocytosis, especially at higher concentrations, cannot be excluded[73,74,103].

Due to the combination of hydrophobic structure and cationic charge, some cyanine dyes display high mitochondrial accumulation[8]. Nevertheless, in addition to the properties of probes themselves (molecular weight, lipophilicity, amphiphilicity, ionic charge and protein/lipid binding characteristics) intracellular distribution can also depend on the phenotype of the target cells[117].

In normal healthy cells, the inner mitochondrial membrane has a strong negative membrane potential [($\Delta\Psi m$) between $-150$ to $-180$ mV relative to the rest of the cytoplasmatic membrane][118]. However, cancer cells can exhibit significantly higher $\Delta\Psi m$ value compared to the corresponding normal cells[119]. Heerdt et al. reported that subcloned lines from SW620 cells with higher $\Delta\Psi m$ display higher VEGF and MMP-7, protein level and invasive potential than the original lines[120]. In the case of subcloned lines with lower $\Delta\Psi m$, the opposite trend was observed. Higher $\Delta\Psi m$ may also increase SW620 cells in hypoxia or nonadherent state[121]. Since cyanic dyes are hydrophobic cations, an increase in $\Delta\Psi m$ is likely to cause their higher accumulation in tumor mitochondria compared to normal cells. Alternatively, the mitochondrial accumulation of some cyanine dyes, such as penthamethine salts, could also be explained by their strong affinity to cardiolipin (localized exclusively in the inner mitochondrial membrane)[65–67,108]. Similarly, the behavior of nonyl acridine orange (a cardiolipin-selective fluorescence probe)[122] is known, and its staining is significantly slower depending on mitochondrial membrane potential[123]. This could suggest that at least some cyanine dyes are localized in the inner mitochondrial membrane.

It should be mentioned that cyanine dyes (even at low micromolar and submicromolar concentration) can disturb mitochondrial metabolism and oxidative phosphorylation and thereby induce cell death[66,124–126]. For example, in MDA-MB-231 cells, IR-783 heptamethine (in tens of micromolar concentrations) induces $\Delta\Psi m$ loss, ATP depletion, opening of the mitochondrial permeability transition pore, and release of cytochrome C[124]. Inhibition of mitochondrial respiration by pentamethines strongly suppresses migration and invasiveness of prostate cancer cells[127]. Application of bis-pentamethine leads to decreased STAT3 phosphorylation and mitochondrial respiration[107]. In addition to conjugation of mitochondrial-selective cyanine dyes and cytotoxic agents, they represent a promising strategy for the targeting of mitochondrial functionality[128,129].

On the other hand, cyanine dyes can have a protective function for mitochondria[130,131]. IR-61 (heptamethine) induces a reduction of mitochondrial damage and reactive oxygen species[130]. In diabetic rats, IR-61 improved bladder function[131]. This effect was associated with suppression of the mitochondrial apoptotic pathway and upregulation of nuclear factor erythroid 2-related factor 2 (NRF2) and associated antioxidant proteins[132]. It is well known, in cancer cells, that increased NRF2 activation is involved in cancer promotion, progression, and metastasis. Conversely, in normal cells, canonical activation of NRF2 prevents cancer initiation and is suitable for cancer chemoprevention strategies. In this context, it should be also mentioned, that NRF2 plays one of key role in the balance of mitochondrial homeostasis[133,134]. Activation of NRF2 has been shown to suppress mitochondrial ROS and promote the clearance of damaged or dysfunctional mitochondria through the process of autophagy, also known as mitophagy. The association between type 2 diabetes and disturbances in mitochondrial dynamics, biogenesis, and mitophagy repression is well-established[135]. It is plausible to consider the use of cyanine dyes as inducers of mitophagy. In a study by Zhu et al., it was reported that the heptamethine dye dc-IR825 can trigger excessive mitophagy in A549 cells[136]. However, further research is warranted to provide additional insights and clarification on this intriguing subject.

## Mitochondrial targeting in PDT

Although mitochondria are oxygen-consuming organelles and therefore have reduced intracellular oxygen levels compared to other cell compartments[137], mitochondria are also very sensitive to PDT compared to other organelles such as the nucleus, endoplasmic reticulum, or lysosomes[138,139]. It was reported that mitochondrial localization of PS, or

mitochondria-related damage, may correlate with their phototoxicity[140–142]. Zhao et al. found that low doses of ROS induced by PDT can cause disruption of mitochondrial respiration, which stimulates other ROS production generated by oxidative phosphorylation[143]. In addition to the lower oxygen consumption caused by targeted mitochondria, PDT can increase the oxygen level in the tumor, thereby making the tumor less hypoxic[144]. In this context, it should be noted that the therapeutic efficiency of direct inhibitors of mitochondrial respiration can be significantly improved by their use as a photosensitizer[145].

Mitochondria might represent a more suitable organelle than lysosome and endoplasmic reticulum (ER) in the case of PDT-stimulated immunogenic cell death (ICD)[11–13]. Mitochondria are highly sensitive to heat and ROS[146] and are closely linked to the endoplasmic reticulum[147], which is typically targeted by ICD inductors[148,149]. It is important to note that certain PS, including cyanine dyes, exhibit potent dual photodynamic and photothermal effects with a strong synergy (section "Combination of PDT and PTT"). Consequently, mitochondrial stress can trigger ER-induced ICD[148]. ICD can also be explained by the immunostimulatory effect of oxidized mtDNA[150–152]. Guo et al. reported a positive correlation between the efficiency of PDT and ICD and the mitochondrial localization of the PS[12]. Although the effect of lysosomal and endoplasmic reticulum-PS on the primary tumor mass (mice with 4T1 tumor) can be comparable to mitochondrial PS[11]. In the case of distant tumor, the lowest and highest efficiency were found for the lysosomal and mitochondrial PS, respectively. Mitochondrial targeting can lead to strong stimulation of the immune system. For example, the ratio of CD4 + T/Treg, and/or CD8 + T/ Treg cells in primary and distant tissue was significantly higher for mitochondrial targeting. In a mouse model featuring 4T1 carcinoma, tumor eradication achieved through mitochondrial-targeted dual phototherapy (PDT and mild hyperthermia) was associated with a robust activation of the immune system[13]. This led to an increase in CD4+ and CD8 + T cell infiltration and a decrease in the population of immunosuppressive cells. Furthermore, the oncogenic M2 phenotype of tumor-associated macrophages was repolarized to the antitumor M1 phenotype.

The above suggests the high potential of PDT targeting mitochondria. However, ROS (produced by PDT), especially $^1O_2$, are highly reactive and their range is very limited. For example, the lifetime and diffusion radius of $^1O_2$ in water are approximately 40 μs and 220 nm, respectively[138]. In the cell, these values will be strongly decreased due to the interaction of singlet oxygen with other molecules. This implies that the affected biomolecules will be in close proximity to the excited PS. Since mtDNA is more sensitive to oxidative damage than nDNA[153], it can be expected that mtDNA damage plays an important role in the phototoxicity of mitochondrial PS. Therefore, mitochondrial oxidative stress may be effective way to target mtDNA[154]. In this context, it should be mentioned that azonia-cyanine can in vitro interact with mtDNA[155]. In comparison, the size of mitochondria ( ~ 1 μm, depending on cell type and conditions) is sometimes larger[156]. Thus, the intracellular distribution of PS has a strong impact on their efficiency[157]. On the other hand, the lifetime of $H_2O_2$ can be between 1 μs-1 ms and other organelles could also be affected[158]. It is well known that organelles can cooperate and form tight connections for regulation of the homeostasis and cell function. Mitochondria-associated endoplasmic reticulum membranes (MAM; specialized membrane region) contain both endoplasmic reticulum (smooth and rough, rat isolated mitochondria) and mitochondria that are in close proximity (9–16 and 19–30 nm, respectively)[147]. It could be suggested that selective PS targeting mitochondria, particularly mitochondrial membranes, could also be effective in the targeting endoplasmic reticulum (ER). Some oxygen radicals, such as $H_2O_2$, ·HO, readily cross membranes and therefore their direct effect should not be limited to mitochondria[56].

On the other hand, the simultaneous co-localization of PS in mitochondria and endoplasmic reticulum can significantly affect their phototoxicity[159,160]. Oxidative stress in endoplasmic reticulum can induce caspase-8, which together with caspase-9 (mitochondrial pathway) is involved in the activation of caspase-3 and subsequently in apoptosis.

**Fig. 4 | Cyanine dyes with improvement photostability.** A photosensitizer exhibiting heightened stability undergoes minimal degradation via irradiation and generated ROS, sustains photoactivity for an extended duration. Various strategies have been documented for enhancing the photostability of cyanine dyes, including aromatic substitution in the γ-position of the pentamethine chain and the alteration of the heptamethine aromatic nitrogen group through alkylpyridinium modification.

The predominant apoptotic effect was observed to occur via the mitochondrial pathway[159]. Similarly, the ECe6 formulation demonstrated localization within the mitochondria, ER, and lysosomes in a time-dependent manner[139]. Notably, PDT targeted at the mitochondria proved to be the most efficacious in eradicating cancer cells, while lysosome-targeted PDT exhibited the least effectiveness in this context. Also, Kassel et al. noted a greater effect on cell viability with the benzoporphyrin derivative (0.5 μM; targeting mitochondria in PDT) in comparison to lyNPe6 (20 μM; targeting lysosomes in PDT)[70,71]. Nevertheless, a synergic effect has also been observed for the combination of mitochondrial and lysosomal targeted PDT[71,161–164].

In addition to the vulnerability of mtDNA, the mitochondrial sensitivity could also be explained by the composition of mitochondrial membranes. The major phospholipid components of mitochondrial membranes are unsaturated and polyunsaturated fatty acids, which are susceptible to oxygen radical attack because of the presence of double bonds that undergo peroxidation through a chain of oxidative reactions[60]. Lipoperoxidation of mitochondrial membranes can be considered not only as a detoxification reaction but also as a new source of radicals due to the self-propagating nature of the highly reactive radicals[61]. Further, ferroptosis-like cell death via mitochondrial $Fe^{2+}$ and/or $Ca^{2+}$ is induced[165].

In accordance with the above, a correlation between mitochondrial membrane affinity and PS phototoxicity has been observed[166]. Nevertheless, their interaction with the mitochondrial membrane may have a strong effect on their aggregation.

Although, in the case of PS (especially porphyrin derivatives), their aggregation (e.g., π–π stacking) significantly decreases ROS generation relative to the monomer forms[167]. However, it could increase their PDT efficiency by reducing the energy gap between singlet and triplet state ($\Delta E_{ST}$). A decrease of the energy gap between the lowest excited singlet ($S_1$) and triplet ($T_1$) state can increase the rate of intersystem crossing[113,168]. Dye aggregates could display a lower energy gap and increase quantum yield of $_1O^2$ ($\Phi_\Delta$) relative to monomer forms[169,170]. At lower $\Delta E_{ST}$, the fluorescent dye is likely to exhibit a higher rate of intersystem crossing and a longer lifetime in the triplet state, both of which are essential for improving the energy transfer from excited dye molecules to oxygen during the photodynamic process[171]. Nevertheless, the localization of cyanine bases in mitochondrial membranes (hydrophobic environment) can cause the decomposition of the aggregation state and thereby reduce the $^1O_2$ production[65,67]. On the other hand, mitochondria with higher $\Delta \Psi m$ exhibit higher concentration cyanine dyes, such as J1, which can induce their aggregation[172]. Also, nonyl acridine orange can form aggregates at higher concentrations (even in the inner mitochondrial membrane)[122].

## Mitochondria-targeting cyanine dyes in PDT
The photodynamic efficiency of cyanine dyes can be significantly enhanced by optimizing their structural motif. For example, the aromatic substitution in the γ-position of the pentamethine chain, or modification of nitrogen groups by alkylaryl groups (in the case of heptamethine) also increases photostability (Fig. 4)[65,67,173].

The aromatic substitution in the γ-position probably increases the steric hindrance of the bonds, where the side arylthiazoles are connected and where the degradation process by molecular oxygen and light occurs. On the other hand, this strategy can decrease their fluorescence quantum yield (Φ), however, their dark cytotoxicity can increase[67]. Nevertheless, different design strategies are used in the preparation of photodynamic cyanine dyes (Fig. 5).

The effect of halogen substitutions in the pentamethine chain (**1a**, Fig. 5) in the γ-position was studied by Huang et al. On the other hand, Cl-substitution (**1b**) did not increase $\Phi_\Delta$ and decreased phototoxicity against MCF-7 cells. Nevertheless, Br-substitution (**1c**) sometimes increases $\Phi_\Delta$ (0.003 vs 0.015; dichloromethane) and photocytotoxicity (IC$_{50}$ = 62 and 1208 nM; MCF-7) compared with parent compound **1a**[174]. The observed effect was associated with stimulation of mitochondrial oxidative stress. Similarly, **1c** exhibited a $\Phi_\Delta$ 1.4% ($\lambda_{ex}$ = 630 nm, EtOH), while no significant value was determined for **1a**[175]. In 4T1 cells, **1a** and **1c** (2.5 μM) did not present obvious cellular toxicity with or without laser irradiation. However, **1a** and especially **1c** have potent phototoxicity (IC$_{50}$ = 62 and 753 nM, 660 nm, 20 mW cm$^{-2}$, 10 min) against 4T1 cells. Similarly, unsubstituted **2a** displayed slow ROS production, but its brominated derivative **2b** had strong photodynamic efficiency. In a mouse model with 4T1 tumor, the combination of **2a** and light irradiation (808 nm, 330 mW cm$^{-2}$) strongly suppressed tumor growth and increased overall survival (OS) of mice. All treated mice were alive on day 60 of the treatment. Irradiation with 630 nm resulted in 70% of mice being alive, while all mice in the control group did not survive the day 50 of experiment.

Bromination of indole cyanine bases (**3a**, **3c** and **3e**; Fig. 5) sometimes increases efficiency in $^1O_2$ generation (**3b**: 12.83-fold, **3d**: 22.74-fold, and **3f**: 8.93-fold)[176]. Nevertheless, in NCI-H460 tumor cells, **3d** displayed only a low increase in ROS production. Incorporation of a triphenylphosphine group (TPP) into the structural motif of cyanine dye can significantly increase their accumulation into mitochondria. Compound **4b** exhibited higher mitochondrial uptake than the corresponding **3b** (~6-fold). However, the localization of others, especially **3d**, was comparable with **4d**. PMSs alone (**3** and **4**) did not show significant cytotoxicity. In a mouse model (NCI-H460), **3d** and its combination with irradiation decreased tumor weight to half and a quarter, respectively.

In this context, Shi et al. reported an interesting cyanine PS (**5-7**) that can target mitochondria by more independent mechanism[177]. Its design was based on the combination of TPP (mitochondrial localization) and chloroacetyl group (protein conjugation) into a structural motif of an indocyanine dye (Fig. 5). Compound **5** (solely TPP) displayed significantly lower fluorescence/accumulation in mitochondria than **6** (solely chloroacetyl group) and **7** (both TPP and chloroacetyl group). The highest dark and phototoxicity and ROS production were found for **7** (IC$_{50}$ = 23.17 and 6.28 μM), and the efficiencies of **6** and especially **7** were sometimes lower. A similar trend in the photodynamic efficiency was observed in vivo.

Increasing the rigidity of cyanine dyes may be a promising way to improve the photodynamic efficiency of cyanine dyes. Polymethine chains (depending on the length) display chain flexibility, thereby increasing the non-radiative dissipation of the excited state energy and reducing the triplet state quantum yield[178]. Zhao et al. prepared NIR cyanine dyes (**8**; Fig. 5) with an incorporated boron difluoride complex in the center of the polymethine chain[179]. The tested dyes displayed strong mitochondrial localization (a correlation coefficient of 0.914). Under light irradiation, they sometimes exhibited higher photostability and $^1O_2$ production (660 nm, 10 mW cm$^{-2}$) than ICG (808 nm, 10 mW cm$^{-2}$). Compound **8** (2 μM, 90 J cm$^{-2}$) displayed potent phototoxicity (cell viability at half) against dark MCF-7 cells, but only low dark toxicity.

In the case of squaraine zwitterionic dye (**9a**; Fig. 5), however, this approach did not lead to an increase in photodynamic efficiency[180]. A possible solution was showed by Lima et al.[181]. Aminosquaraine dyes (**10a-10c**; Fig. 5), although exhibiting marked cytotoxicity compared to the zwitterionic dye **9b**, displayed significantly higher photostability, $^1O_2$ production, mitochondrial localization and phototoxicity. The loss of negative

charge could reduce solubility and lead to a higher mitochondrial accumulation of cyanine dyes and thus higher ROS production in the dark. Nevertheless, their co-localization with rhodamine was low.

The bichromophoric cyanine dye (**11**; Fig. 5) represents a promising structural motif for PDT. It sometimes showed lower IC$_{50}$ values against melanoma cells than Photogem®[182]. In the case of human peripheral blood mononuclear cells, the phototoxicity of **11** did not exceed 70% even for a concentration of 20 μM, whereas concentrations below 5 μM resulted in 100% melanoma cells killed. Nevertheless, the phototoxicity of Photogem® (20 μM) reached 80%. On the other hand, the LC$_{50}$ of **11** for PBMC cells was 3 times lower than for Photogem®.

Cyclic salt (**12**; Fig. 5) displayed very strong phototoxicity against A375 melanoma cells (IC$_{50}$ = 121 nM;). After irradiation (630 nm, 5 J cm$^{-2}$), cell viability was 14-times lower than that of the corresponding treated but not irradiated cells[68]. However, at 100 nM, intracellular ROS levels were only slightly elevated. On the other hand, irradiated compound **12** caused disruption of mitochondrial tubular structures and formation of small vesicular-shaped mitochondria. Apart from mitochondrial structure, salt **12** showed no significant effect on the mitochondrial membrane potential without irradiation. However, the introduction of heavy atoms into the structure of cyanine dyes can cause undesirable cytotoxicity under dark conditions [21], thereby decreasing their specificity for the tumor.

The combination of the PS structural motif with stable radical may lead to stimulation of higher ROS production. In contrast to agents combined with PS alone, the ISC process is promoted through the radical triplet pair mechanism. The process is accompanied by an increase in the static and dynamic free volume, allowing quenching of the PS excited triplet state by oxygen[183]. An alternative mechanism involves irreversible intramolecular electron transfer from the excited singlet of the PS donor to the nitroxide acceptor with subsequent regeneration of the fluorophore segment and hydroxyl formation[184]. The combination of PS with stable radicals can also be used under hypoxia conditions (lower oxygen level)[185]. Incorporation of 2,2,6,6-tetramethylpiperidinyloxy radical into the structural motif of indolium cyanine (**13**; Fig. 5) significantly improves its photodynamic efficiency[186]. The $\Phi_\Delta$ of **13** increased several-fold (from 0.0170 to 0.323), but $\Phi$ and fluorescent lifetime decreased from 0.285 to 0.186 and 3.43 to 2.55 ns, respectively. The maximum absorption and emission wavelengths display a red shift (from 605 to 723 nm and 683 to 746 nm, respectively) compared to the original dye. After irradiation (700 nm), compound **13** (1 μM) strongly reduces the viability of MCF-7 cells (less than half). In the dark, its cytotoxicity was minimal. In a mouse model with 4T1 tumor, compound **13** and especially its nanoparticle formulation (PEG-SS-PCL micelles) significantly decreased tumor volume relative to control.

**Multifunctional photodynamic cyanine dyes.** It is well known that the combination of multiple therapeutic modes in cyanine dyes can significantly enhance their therapeutic efficiency. Examples of these combinations are shown in Fig. 6.

One strategy involves constructing chimeric agents that combine the structural motif of two anticancer agents, such as photodynamic cyanine dyes and cytostatic drugs. For example, the antitumor efficacy of chimeras consisting of xanthene-cyanine dyes and DNA methylating methyl triazene moiety (**14**; Figs. 6 and 7) was studied using triple-negative human breast cancer cell line MDA-MB-238[129]. The toxicity of **14a** and especially the phototoxicity (IC$_{50}$ = 42 vs 4.8 μM; 660 nm, 45 mW cm$^{-2}$, 1 h) were lower than that of the released original dye **15a**. The phototoxicity of **14b** was comparable with **14a**, nevertheless, the released **15b** displayed a significantly lower IC$_{50}$ (2.7 μM). On the other hand, the fluorescence emission of **15a** and **15b** was strongly suppressed compared with the corresponding chimeras **14a** and **15b**.

Another promising agent for the combination of chemo-photodynamic therapy was prepared by Liu et al[128]. Its structural motif contains a NIR photosensitizer, the anticancer drug 5′-deoxy-5-fluorouridine, and a bisboronate group as a linker (**16**; Fig. 6). Conjugate **16** displays very slow fluorescence and photodynamic activity, however, in the presence

**Fig. 5 | Examples of Photodynamic cyanine dyes.** Tested photodynamic cyanine dyes exhibit a significant degree of structural variability, with their photophysical properties being influenced by their structure and substitutions. For example, halogenation of **1a** and **2a** resulted in only a minor shift in the position of the absorption maxima. However, pentamethine derivatives (**1c**, **2b**, **3b**, **3d**, and **3f**) with bromine substitutions occasionally demonstrate notably higher photodynamic efficiency compared to the original compounds. In biological systems, their efficacy is notably impacted by mitochondrial selectivity. In general, cyanine dyes (**3d** and **7**) containing cationic groups (such as triammonium salt and triphosphonium group) and/or protein conjugation groups (preferably both) exhibit improved mitochondrial selectivity and photodynamic efficiency when contrasted with corresponding cyanine dyes (**4d**, **5**, and **6**). A successful strategy involves developing more rigid cationic cyanine dyes (**8**, **10a**-c, and **12**) with enhanced photodynamic efficiency. However, the squaraine zwitterionic dye (**9b**) may not be as efficacious. Encouraging photodynamic efficiency is also noted in the bichromophoric cyanine dye **11** and the hemicyanine dye **13** substituted with a tetramethylpiperidinyloxy radical.

**Fig. 6 | Examples of multifunctional photodynamic dyes.** In addition to its standalone photodynamic therapy function, cyanine dyes can exhibit other therapeutic and diagnostic capabilities. Chimeric compounds **14a, 14b,** and **16** are cleaved to form active hemicyanine photosensitizers **15a, 15b,** and **17**, as well as cytostatic drugs. The conjugation of the phenanthrimidazole Ru²⁺ complex with heptamethine dyes (**18a, 18b,** and **19**) demonstrates enhanced efficacy under hypoxic conditions (thank to cytotoxicity and activation of type I mechanism) compared to the original cyanine dye. The fusion of porphyrin with the heptamethine structural motif (**20**) results in the development of a potent theragnostic photosensitizer. Furthermore, the theranostic heptamethine and hemicyanine dyes (**21** and **23**) can also serve as fluorescence probes for cysteine and aminopeptidase, respectively.

**Fig. 7 | Anticancer effect of xanthenecyanine with DNA methylation ability.** The hydrolysis of the carbamate bond activates the unstable monomethyl triazine group. This group spontaneously releases diazomethane, which further cleaves to form a highly reactive methyl carbocation (DNA methylation agent) and $N_2$. The remaining parts of **14a** and **14b** undergo self-immolative cleavage, liberating 4-iminocyclohexa-2,5-dien-1-one and the photoactive hemicyanine **15a** and **15b**.

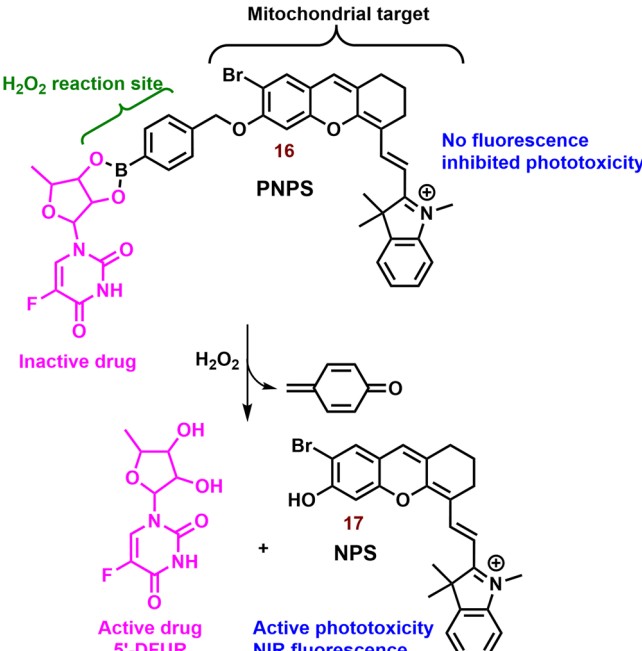

**Fig. 8 | Conjugate of cyanine dye with fluorouridine cytostatic.** In the presence of $H_2O_2$, non-active hemicyanine dye **16** is cleaved on the chemotherapeutics (5-DFUR), active PS **17** and cyclohexadienone byproduct (oxidized linker).

$H_2O_2$ (higher level in cancer cells)[187], **16** is cleaved (Fig. 8). Free PS **17** shows potent mitochondrial localization (Person correlation coefficient (P) = 0.965), strong fluorescence emission ($\lambda_{max}$ = 710 nm) and sometimes ROS production relative to the **16**. In cancer cells (HeLa and HepG2 cells), **16** displays significant dark toxicity ($IC_{50}$ = 16.6 μM and 14.8 μM, 3 h) and phototoxicity (9.32 μM and 8.15 μM), respectively. However, both **16** and **17** display lower photo and dark cytotoxicity against normal HL-7702 cells, for example, more than 75% of the tested HL-7702 cells survived 10 μM of **16**. Nevertheless, the phototoxicity of **17** was stronger ($IC_{50}$ = 8.50 μM). In a

mouse model with HCT116 tumor, strong fluorescence emission was found in the tumor tissue, liver and stomach, and much weaker fluorescence was observed in lung, heart, kidney and spleen after an infection of the **16**.

Incorporation of phenanthrimidazole $Ru^{2+}$ complex into the structural motif of symmetric heptamethine cyanine dyes leads to a highly effective PS for the treatment of hypoxic tumors (**18a, 18b** and **19**; Fig. 6)[188]. The prepared conjugates display strong absorption and emission in the NIR region (approximately 800 nm) and potent $\Phi_\Delta$ (15-20%). The $Ru^{2+}$ complex itself has a significantly higher $\Phi_\Delta$ (64%), but the position of its absorption and emission maxima was observed at shorter wavelengths (458 and 607 nm, respectively). In vitro tested compounds (**18a, 18b** and **19**) exhibit strong mitochondrial localization/co-localization (P = 0.76, 0.67 and 0.69, respectively), comparable to the original PMS (0.74) and significantly higher intracellular uptake than the $Ru^{2+}$ complexes alone. The best phototherapeutic index (PI; ratio between phototoxicity and dark toxicity) was found for **18a** (PI = 106). Potent dark and phototoxicity against CT-26 colon cancer cells was detected under both tested conditions: hypoxia ($IC_{50}$ = 12 and 0.62 μM, respectively) and normoxia (18 and 0.33 μM, respectively).

The combination of heptamethine and the porphyrin structural motif represents an interesting strategy for the preparation of novel PS[189,190]. A conjugate with mitochondrial localization (**20**; Fig. 6) was prepared by Chen et al.[189]. Its photodynamic efficiency against radiation-induced fibrosarcoma cells was not significantly different from that of the original porphyrin and cyanine dye (irradiation at 665 and 810 nm, respectively). However, the higher photocytotoxicity of **20** was sometimes observed with irradiation at 665 nm (porphyrin PDT). In a mouse model, strong fluorescence (780 nm) of **20** in the tumor tissue, even at a subtherapeutic dose (0.3 μmol kg$^{-1}$), was observed more than in blood and other body tissues, and promising efficiency against RIF tumor was observed. Analysis of tumor tissue showed heterogeneous distribution; the majority of **20** was localized in the growing edge, with a minority in the necrotic center. This suggests a promising potential of **20** for the tumor labelling. The applicability of the original porphyrin and cyanine dye was significantly lower. Compound **20** also showed strong efficiency against RIF tumor. More than 75% of the treated mice were alive at day 90 of the experiment (3.5 μM; 665 nm, 135 J cm$^{-2}$), while control group was dead by day 10 of the experiment.

**Fig. 9 | Interaction of cyanine probe with cysteine.** PS **21** is photoactive but lacks fluorescence emission. However, a cysteine nucleophilic attack against the sulfide bond leads to the formation of two fluorescent probes: naphthalimide-thioether and cyanine-thioether ($\lambda_{abs}/\lambda_{em}$ 660/750 nm). $\lambda_{abs}$, absorption wavelength; $\lambda_{em}$, emission wavelength.

**Fig. 10 | Hydrolysis of 23 by Aminopeptidase N (APN).** Although hemicyanine **23** exhibits low fluorescence emission and photodynamic efficiency, it undergoes APN hydrolysis, resulting in the subsequent self-cleavage of the aniline group the photoactive probe **24** is liberated.

Suitably designed PS can also be used as theranostic agents. For example, Shen et al. prepared a fluorescence probe (**21**; Fig. 6) for the determination of mitochondrial thiols (mainly cysteine)[191]. In the presence of cysteine, naphthalimide and **22** are released and fluorescence emission (~750 nm) is sometimes increased (Fig. 9). Compound **21** (independent of the thiol reactions) displayed potent phototoxicity (IC$_{50}$ = 3.7 μM) against A549 cells and strongly inhibited their migration. In the wound healing assay, the scratch width of A549 cells exposed to **21** and irradiation was almost unchanged within 24 h and slightly narrowed after 48 h. However, without irradiation, cytotoxicity was significantly lower, and the majority of the scratch (more than 75%) was closed after 48 h.

An NIR photosensitizer based on the hemicyanine dye (**23**; Fig. 6) was studied for tumor imaging and tumor selective PDT[72]. In the presence of APN (Aminopeptidase N, APN/CD13) expressed on the surface of cancer cells[192], **23** was hydrolyzed to **24** (Fig. 10)[72]. After hydrolysis, $\Phi$ increased from 0.005 to 0.024. In the presence of HepG-2, or 4T1 cells, an increase in increase fluorescence of **23** ($\lambda_{ex}$ and $\lambda_{em}$ = 685 and 717 nm, respectively) was sometimes observed. However, bestatin inhibitor of APN suppressed the increase in fluorescence. However, in the presence of COS-7 and LO2 cells, the fluorescence response was sometimes slower. Similarly, **23** did not display any significant ROS formation, nevertheless, in the presence of HepG-2 cells, strong ROS formation was observed. **23** exhibited strong selectivity for mitochondria (P = 0.94) compared to lysosomes and nucleus (P = 0.60 and 0.04, respectively) in HepG-2 cells. PN-CyI (2 μM) had significant phototoxicity to cancer cells (viability ~15%) and relatively little damage to normal cells (cell viability ~70%). In the 4T1 mouse model, **23** displayed strong fluorescence in a tumor tissue and potent phototoxicity against tumor.

**Cyanine dyes for the type I PDT.** In the context of hypoxia, type I PDT emerges as a highly promising therapeutic avenue. However, compared to type II PDT, the availability of low molecular weight agents is notably limited. The examples of type I PDT based on the cyanine dyes are showed on Fig. 11.

This is where the modulation of ISC becomes crucial, presenting a potential strategy for designing type I PS[193]. The ISC rate constant is directly and indirectly dependent on the spin orbital coupling and $\Delta E_{ST}$. Therefore, a small $\Delta E_{ST}$ and a substantial spin-orbit coupling (SOC) play pivotal roles in the efficacy of PS. There are two primary approaches for modulating ISC: introducing heavy atoms (e.g., halogens)[73,194,195] introducing reductive strong electron donors (D) and acceptors (A)[196,197].

Zhang et al. studied a series of halogen-substituted hemicyanine dyes (**25-29**) as activatable/theranostic type I photosensitizers (Fig. 11)[73]. It is interesting to note that brominated hydroxy hemicyanine **17**, possessing a closely related structure to **26** and **27**, demonstrated strong mitochondrial localization (P = 0.965)[128]. The maximum absorption and emission wavelengths were around 670 and 720 nm, respectively[73]. When it comes to halogenated dyes, only a slight red shift and a decrease in $\Delta E_{S1-T1}$ (from 1.29 to 1.26) were noted. However, the spin-orbit coupling (SOC) of **26-28** saw a significant increase from 0.06 to 0.08, 0.24, and 0.42, respectively. Unlike methylene blue, none of the tested hemicyanine scaffold (**25-28**) displayed only a mild effect on the absorbance spectra of 9,10-anthracenediyl-bis(methylene)dimalonic acid (ABDA; $^1O_2$ selective probe)[198] after irradiation (660 nm; 30 mW cm$^{-2}$, 10 min). In the case of dihydrorhodamine 123 (DHR 123; $O_2^{\cdot-}$ probe)[199,200] a significant increase in fluorescence intensity was observed, suggesting that their photodynamic effect is based on the production $O_2^{\cdot-}$. Compound **27** exhibited the highest ROS production, followed by **26**. However, there was no significant change in the fluorescence spectra of terephthalic acid (TA; $^{\cdot}$OH probe)[201]. The fluorescent quantum yields were calculated to be 0.21 for **25**, 0.12 for **26**, 0.07 for **27**, and 0.03 for **28**. Upon phosphorylation of the **27** phenyl group, a blue shift to 600 nm was observed, leading to strong repression of fluorescence ($\lambda_{ex}$ = 660 nm). This process also improved photostability and inhibited ROS production. In the presence of alkaline phosphatase (ALP; overexpressed by tumor tissue)[202], the phosphate ester **29** was hydrolyzed, releasing **27** (Fig. 12). Compound **29** exhibited strong fluorescence in HePG2 cells with high ALP expression, while showing low fluorescence in LO2 normal liver cells without ALP overexpression. Notably, significant cytotoxicity of **29** was not

**Fig. 11 | Examples of type I PS based on the cyanine dyes.** The majority of the reported PS belong to type II. However, some type I PS, based on the structural motif of cyanine dye (mainly hemicyanine), have also been developed. For instance, brominated hydroxy hemicyanines (e.g., **26** and **27**) represented intriguing scaffolds for the design of type I PS. Their phosphorylated derivatives, such as **29**, can be selectively activated by alkaline phosphatase in tumor tissues. The findings related to **30b** demonstrate how structural optimization through hexyl substitution can enhance mitochondrial selectivity and photodynamic efficiency. In the case of suitably designed PS (e.g., **31**), their aggregation can significantly bolster type I PDT.

observed in LO2 cells under dark or light irradiation conditions. In HePG2 cells, **29** (3 and 4 μM) was able to kill cancer cells under normoxia and hypoxia. However, the ALP inhibitor $Na_3VO_4$ (100 μM) suppressed **29** phototoxicity and fluorescence in HePG2 cells. Conversely, the effect of $NaN_3$ ($^1O_2$ scavenger) on cell viability was insignificant. Nevertheless, **29** showed a significantly higher preference for the lysosome (P = 0.75) and ER (0.32) than the mitochondria (0.25). In a mouse model with HeG2 carcinoma, **29** (100 μM, 100 μL) exhibited strong fluorescence emission in tumor tissue and halted tumor growth after irradiation (660 nm; 0.5 W cm$^{-2}$, 10 min). Co-application with $Na_3VO_4$ significantly decreased the phototherapeutic effect of **29**.

The combination of A and D in PS design can facilitate the separation in the distribution of the lowest unoccupied and highest occupied molecular orbitals (LUMO and HOMO), leading to a significant reduction in $\Delta E_{ST}$ and effective ISC[196]. Zhao et al. developed innovative D-A-π-A photosensitizers (**30a** and **30b**) featuring a hybrid structure of aminophenoxazinone and hemicyanine[203]. The maximum absorption peaks and fluorescence emission of **30a** and **30b** (Fig. 11) were around 655 and 682 nm in Tris buffer (pH = 7.4), respectively. Following irradiation (red LED light, 50 mW cm$^{-2}$, 100 s), DHR 123 exhibited a comparable fluorescence response to both tested dyes, while the decomposition rates of ABDA were very slow. Notably, the more lipophilic **30b** demonstrated significantly higher mitochondrial localization compared to **30a** (P = 0.88 vs 0.55) and induced ROS production in MCF-7 cells. Both dyes, particularly **30b**, exhibited potent phototoxicity, with doses of approximately 1 μM and 0.2 μM of **30a** and **30b** reducing cell viability by up to 40% post-irradiation (red LED light; 50 mW cm$^{-2}$, 3 min).

In scenarios characterized by high ISC, competitive processes such as nonradiative decay from single and triplet exceptions ($S_1 \rightarrow S_0$ and $T_1 \rightarrow S_0$, respectively) can hinder photodynamic efficiency[196]. Intermolecular interactions like π-π stacking observed in aggregates of aromatic compounds strongly support these processes. A potential solution to this challenge lies in luminogens with aggregation-induced emission[204]. Within their aggregates, all motions (e.g., rotation and vibration) are constrained due to short intermolecular distances, preventing π–π stacking and consequently enhancing fluorescence emission. Moreover, these aggregates exhibit significantly smaller $\Delta E_{ST}$, favoring type I reactions via charge-separated states for electron transport compared to type II monomer forms[205].

Building on this concept, Li et al. explored ortho-dimethyl-substituted derivatives **31-33**) as electron donors in combination with various electron acceptors (pyridinium and imidazolium cations; Fig. 11)[197]. Compound **31** ($\lambda_{ex}$ = 430 nm and $\lambda_{em}$ = 700) notably enhanced the fluorescence of 20,70-dichlorodihydrofluorescein diacetate (DCFH-DA; ROS probe)[206] and DHR 123 post-irradiation (white light 30 mW/cm$^2$, 4 min), while the absorption peaks of ABDA remained largely unchanged. The replacement of iodate atoms with hexafluorophosphate suppressed photodynamic efficiency. In the case of **32** ($\lambda_{ex}$ = 540 and $\lambda_{em}$ = 795 nm) and **33** ($\lambda_{ex}$ = 580 and $\lambda_{em}$ = 710 nm), their photodynamic efficiency was occasionally lower (~8-fold). However, **32** exhibited poor ROS generation due to the absence of a pyridinium iodide salt unit crucial for supporting ROS generation. The rigid planar conformation of **33** (Fig. 11) can promote dye aggregation through intermolecular π–π interactions, leading to unintended energy loss via fluorescence emission[207–209]. Dye compound **33** exhibits minimal fluorescence in water but significantly higher fluorescence in toluene[197]. The fluorescence emission intensity of **31** and **33** was comparable. In HeLa cells, **31** demonstrated notable mitochondrial localization (P = 0.94) but exhibited low colocalization with lipid droplets and lysosomes (P = 0.24 and 0.58, respectively). Compound **31** displayed potent phototoxicity (IC$_{50}$) against HeLa (11.5 μM), MCF-7 (9.7 μM), and A549 (4.8 μM) under hypoxic conditions (less than 1% O$_2$), with lower dark toxicity observed in some instances.

**Fig. 12 | Activation of 29 in presence alkaline phosphatase (ALP).** Phosphorylated hemicyanine **29** exhibits very low fluorescence emission and ROS production. However, upon ALP hydrolysis, it releases the potent photoactive probe **27**.

## Mitochondrial Targeting in PTT

The effect of PPT is not as localized as PDT, but the temperature change depends on the concentration of agents used in addition to properties of the environment[210]. Since a temperature gradient in the cell/tissue cannot be excluded[211], organelle-targeted therapies are intensively studied[212]. In the case of PPT, mitochondria could represent very promising target[213]. Heat shock leads to a disruption of mitochondrial homeostasis such as an increase in mitochondrial ROS production and subsequently oxidative damage of mitochondria. Heat shock also inactivates CI and thus the electron transport chain, oxygen consumption and ATP synthesis[214]. Since intracellular oxidative stress plays an important role in heat shock-induced apoptosis, an increase in ROS levels could stimulate cytotoxic effects of heat shock[60]. On the other hand, the application of antioxidants such as glutathione or Mito-TEMPO could decrease the sensitivity of cells to heat shock[215]. For example, Mito-TEMPO (mitochondria-targeted ROS scavenger) inhibited hyperthermia-induced malonyldialdehyde production, cardiolipin peroxidation and platelet apoptosis[216]. However, PTT is much more independent of the oxygen level than PDT.

Although inorganic nanoparticles (e.g., metal oxide nanoparticles, and quantum dots) are usually studied for the PPT[217], they are not readily biodegradable, and their non-negligible long-term toxicity may strongly limit their application. Therefore, testing of organic systems such as cyanine dyes for PPT can be initiated[7].

### Mitochondria-targeting photothermal cyanine dyes

Heptocyanin dyes with strong absorption in the NIR region are usually studied for photothermal therapy. However, this does not necessarily mean that substances with absorption at significantly lower wavelengths could not be used. Examples of photothermal cyanine dyes are shown in Fig. 13.

Thiazole orange substituted with triphenylphosphonium ($\lambda_{max} \sim 600$ nm; **34**, Fig. 13) showed potent photothermal efficiency (0.5 mM; $\Delta T = 15$ °C, 600 nm, 1.5 W cm$^{-2}$, 5 min), whereas the efficiency of thiazole orange alone ($\lambda_{max} \sim 500$ nm) was negligible[218]. This difference was most probably caused by very low absorbance of thiazole orange itself at 600 nm. In vitro (MCF-7 and U87) **34** (50 μg/ml) displayed killing of more than 80% of cells after 5 min of irradiation. In a mouse model with MCF-7 tumor, **34** exhibited sometimes higher fluorescence in tumor tissue compared with liver, kidney, lung, heart, and spleen after application (24, 48 and 72 h). Fourteen days after treatment (5 mg kg$^{-1}$; 1.5 W cm$^{-2}$, 5 min), a reduction of tumor tissue by more than one magnitude was observed.

Due to their unique properties (high fluorescence in the NIR region and selectivity for tumor tissue), photothermal agents represent promising theranostic agents, for example anionic heptamethine cyanine dyes (**35a-d**; Fig. 13) developed by Zhang et al.[219]. The prepared compounds displayed mitochondrial and lysosomal distribution in deprotonated and protonated forms, respectively. The deprotonated dyes have the HOMO localized both on the Hcyanine scaffold and the bridgehead amine, whereas the LUMO is distributed only along the polymethine chain, indicating a potent charge transfer from the terminal amine to Hcyanine. After protonation, electrons in the HOMO of the conjugated polymethine chain predicted a large overlap (from 1.989 to 2.074 eV) with electrons in the LUMO. Their protonated

form (observed in the acidic pH of lysosomes) exhibited an order of magnitude larger $\Phi$ relative to unprotonated forms in the mitochondrial pH (Fig. 14). On the other hand, deprotonation of these dyes increases their photothermal efficiencies (7.1-fold relative to the protonated form). For example, after laser irradiation (750 nm, 6.0 W cm$^{-2}$, 3 min), **35b** (10 μM) increased the temperature of the buffer (7.4 pH) nearly to 70 °C. More importantly, cyanine **35b** fluorescence was observed only in the lysosomes of cancer cells (HepG2 and HeLa), i.e. not in normal cells (HL-7702). Compound **35b** (20 μM, 12 h) killed more than 80% of cancer cells after laser irradiation (750 nm, 6.0 W cm$^{-2}$, 10 min), but only 10% of normal cells.

A potential limitation of fluorescent photothermal agents in bioanalytical/diagnostics applications can be their phototoxicity. A possible solution could be to separate the analytical and therapeutic functions. Compound **36** (Fig. 13) displayed two separated excitations (580 and 808 nm) for both red fluorescence imaging and NIR photothermal therapy (PTT), respectively[220]. Upon green light irradiation (in organic solvent and nonpolar media), **36** exhibits high fluorescence ($\lambda_{max} \sim 720$ nm, $\Phi > 43\%$) at 580 nm excitation. Nevertheless, in the NIR region below 808 nm, the fluorescence emission is slow ($\Phi < 0.14\%$) On the other hand, NIR irradiation shows efficient light-to-heat conversion (17.4%). Compound **36** shows strong preference for the mitochondria (P ~ 0.95). In addition, significant selectivity towards cancer cells was found. Under the same incubation and imaging conditions, mitochondria in A549 cells were clearly observed with strong red fluorescence; however, very weak fluorescence was seen in AT II cells. After laser irradiation (808 nm 1.0 W cm$^{-2}$, 5 min), **36** (5 and 10 μg mL$^{-1}$) killed most of exposed cancer cells ( ~ 70% and >95%, respectively). Dark cytotoxicity was low at the concentrations below 20 μg ml$^{-1}$.

One of the key conditions for the success of photothermal and generally anticancer agents is their selectivity for the tumor tissue, which may depend significantly on the structure of the cyanine base. Li et al. reported that chloro-cyclohexene ring and indolium unit with carboxyl group on the heptamethine chain (**37a**; Fig. 13) are key structural features for improved distribution in a tumor[221]. In vitro, heptamethinium without carboxylated group (**37a**) sometimes exhibited higher cytotoxicity in NIH/3T3 cells. Nevertheless, in a mouse model with NCI-H460 tumor, its tumor distribution was significantly lower with carboxylated heptamethine (**38**; Fig. 13). According to the above, the therapeutic efficiency of **37b** in the combination with laser irradiation (1.1 W cm$^{-2}$, 5 min) was insignificant, whereas in the case of **37a**, the tumor was practically eradicated. In this context, it should be noted that the position of their absorption maximum did not differ significantly, although **37a** exhibited significantly higher excitation coefficient. This effect can be explained by the binding of **37a** to serum albumin (BSA), which can accumulate in tumor tissue[9].

## Combination of PDT and PTT

The combination of PDT and PTT can have a significant synergic effect in the treatment of cancer[96,222]. For example, PDT can disrupt tumor physiology (e.g., decreased pH), thereby increasing heat sensitivity[223]. PTT efficiency can be significantly increased by the Warburg effect, which induces tumor acidification in a poorly oxygenated tumor tissue[224]. On the other hand, PTT can stimulate an increase in blood flow by the heat

**Fig. 13 | The examples of photothermal cyanine dyes.** While heptamethine cyanine dyes are predominantly reported as PSs, triphenylphosphonium-substituted thiazole orange (**34**) has shown enhancements in photophysical properties and especially photothermal efficiency. Compounds **35a-d** and **36** serve as intriguing examples of switchable PSs, demonstrating photothermal activity and near-infrared (NIR) fluorescence depending on pH and excitation wavelength, respectively.

However, the impact of the cyanine dye structure on its tumor selectivity can be a crucial consideration. For instance, heptamethine **37a**, featuring a chloro-cyclohexene ring and an indolium unit with a carboxyl group, exhibits significantly higher selectivity, possibly through interactions with serum albumin, compared to **37b** (solely chloro-cyclohexene) and **38** (solely carboxylate groups).

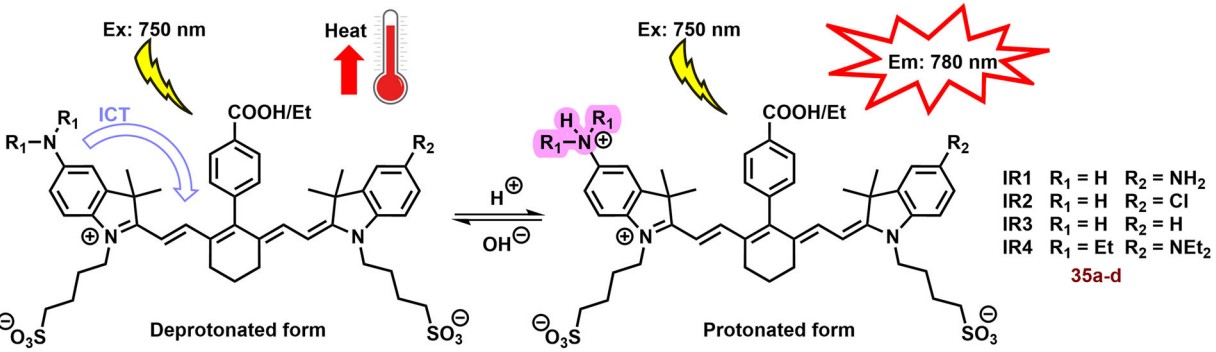

**Fig. 14 | Functionality of Hcyanine dyes in dependence of pH.** The deprotonated form of Hcyanine dyes exhibits potent photothermal efficiency due to intramolecular charge transfer (ICT) from the terminal amine to Hcyanine. However, protonation of Hcyanine results in the loss of ICT and photothermal activity, while strong fluorescence emission can be observed.

produced and thereby increase oxygen levels in cancer cells[223]. Additionally, hyperthermia can increase the damage induced by PDT[225]. The increase in cellular temperature may participate to cellular damage by the denaturation of DNA repair enzymes[226]. Although it is not so well known, nDNA repair proteins also serve in in maintenance and repair of mtDNA[227]. However, it should not be dismissed that heat shock proteins play an important role in the resistance of cells to photodynamic therapy[228]. On the other hand, mitochondria are very sensitive organelles to ROS produced during photodynamic and partially photothermal therapy.

Nevertheless, it has become evident that the combination of PDT and PTT is very promising tool against superficially localized tumors such as melanoma, or hypoxia tumor[217,222,229]. To this end, nanoparticles with two independent excitation wavelengths or more advanced systems activatable by one specific wavelength in the NIR region are being studied[230]. It should be noted that some small molecule compounds, such as cyanine dyes, exhibit both photodynamic and photothermal effects[229,231]. For example, NIR irradiation of indocyanine green stimulates heating and ROS production[231]. This dual effect has also been observed for mitochondria-targeted dyes such as heptamethine (see next subsection). However, the experimental results and the potentially synergistic effect can be strongly dependent on the experimental design. In a mouse model of radiation-induced-fibrosarcoma, PDT alone and heat followed by PDT cured less than 10% of the animals, and heat alone had no significant effect[225]. Nevertheless, PDT followed by heat cured almost half of the treated mice (45%). In the case of dual cyanine bases, both therapeutic modalities PDT and PTT can be expected to be activated simultaneously after

**Fig. 15 | Examples of dual photodynamic and photothermal cyanine dyes.** Heptamethine dyes with a central chloro-cyclohexene group, like **41**, are commonly explored as photosensitizers for dual PDT and PTT. However, this does not preclude structural optimization. For instance, **39g** exhibits remarkable selectivity and efficacy against tumors. Conversely, compounds substituted with triphenylphosphonium (**40**) and fluorinated amphiphiles (**42a** and **42b**) in the γ-position demonstrate enhanced photoefficiency compared to the original **41**, particularly in ROS production.

**Fig. 16 | Phototoxicity of HC-1 (compound 17) on the dependence of light irradiation wavelength.** Upon exposure to 640 nm and 808 nm irradiation, compound **17** display photodynamic one therapeutic effect (photodynamic and photothermal, respectively). For the effective dual PDT and PTT simultaneously, irradiation at both 640 and 808 nm is necessary.

irradiation, and any novel phototherapeutic applications may require a new dose of the agents.

## Mitochondria-targeted dual photodynamic and photothermal cyanine dyes

Light-irradiated cyanine dyes can simultaneously produce ROS and heat. The examples of dual cyanine dyes are showed on Fig. 15.

Whether a cyanine base will display photodynamic or rather photothermal efficiency may also depend on the balance between monomeric and aggregate forms. As a result, the excitation wavelengths used for PDT and PTT in the case of hemicyanine could differ significantly. In DMSO, brominated hemicyanine (**17**; Fig. 15) displayed an absorption maximum (represented by the monomer form) at 693 nm and the absorbance in the NIR region was very low[232]. In the aqueous environment, the intensity of this peak ($\lambda_{max} \sim 650$ nm) is lowered and a new peak ($\lambda_{max} \sim 800$ nm) can be observed. While light irradiation of the original peak maximum (640 nm, 300 mW, PBS) induces strong ROS production and slower ΔT (7.8 °C), in

the case of NIR light (808 nm, 100 mW, DMSO) ΔT was sometimes higher (17 °C) and ROS production significantly lower (Fig. 16). However, 640 nm-30 mW irradiation triggers only ROS production. Considering the absorption spectra of **17**, the result of photothermal efficiency is surprising. A possible explanation could be provided by Chen et al. They reported that **36** (IR825-Cl; chlorinated cyanine dye) had less than 1% $\Phi$ (similarly **17**) upon excitation at 808 nm, but displayed high photothermal conversion efficiency in PBS[220]. More importantly, dual light irradiation (640 nm-300 mW and 808 nm-1000mW) led to a higher ROS production and ΔT (21.3 °C)[232]. In Hela cells, **17** produced more ROS and exhibited higher effect on the cell viability under 640 nm-300mW than 640 nm-30mW. Whereas in the SW480 cells, this difference (significantly slower) was observed only at a higher dose of **17** (7.5 ug/ml), but ROS production was higher. A lower dose of **17** (2.5 ug/ml) displayed a higher effect on the cell viability under 640 nm-30mW. Under 808 nm-1000 mW irradiation, ROS production was higher but the effect on the cell viability was weaker. In the case of dual irradiation (640 nm-300 mW and 808nm-1000 mW, simultaneously), a strong synergic

effect was observed. The live/dead assay indicated almost complete killing of SW480 (10 μg mL$^{-1}$) and HeLa cells (7.5 μg mL$^{-1}$) after 5 min co-irradiation.

Although the combination of PDT and PTT may exhibit a strong synergistic effect, other therapeutic properties, such as selectivity for tumor tissue, should be neglected. Compounds **39a-c** (Fig. 15) with water-soluble groups (-OH, -COOH, -SO$_3$Na) or **39d** and **39e** (Fig. 15) with strong hydrophobic groups (difluorobenzene and triiodobenzene) did not show tumor targeting and accumulation[233]. Heptamethine cyanine dyes with lipid-water partition coefficients about six, such as compounds with alkyl substituted benzoic and methoxybenzoic acid **39f** and **39g** (Fig. 15), showed better tumor targeting ability. Heptamethine cyanine dyes with methoxybenzoic acid **39g** also exhibited potent cytotoxic effect (based on the PDT and PPT) against cancer cells, especially against 4T1 after irradiation at 880 nm. In a mouse model, the application of cyanine and irradiation completely inhibited tumor growth, and no tumor recurrence was observed with 100% survival rate during the observation period (60 days). In the control group and the cyanine group without irradiation, each mouse died before day 40 and 45.

It is well known that Cl group in γ-position of heptamethine dyes, through binding to serum albumin, plays an important role in their selectivity for tumor tissue[9]. On the other hand, this does not mean that there are no other suitable groups for substitution in this region, e.g., TPP. Zhang et al. tested a conjugate of IR780 with TPP (**40**; Fig. 15) for the mitochondrial targeting of PDT and PTT. Compound **40** displayed a blue shift of λmax against **41** (IR-780; from ~ 800 to 600 nm)[234]. Their photothermal efficiencies were comparable, however, **41** showed significantly higher ROS production after irradiation (808, or 660 nm). In 4T1 cells, **40** displayed a stronger effect on mitochondrial membrane potential and much stronger ROS production. In a mouse model with 4T1 tumor, the application of light and **40** led to higher suppression of tumor growth, metastatic activity, higher OS and higher CD4+ and CD8 + T cells/ Treg ratio.

Another promising strategy based on the substitution in γ-position was published by Jing et al.[235]. The conjugation of IR780 with fluorinated amphiphilic building blocks (**42a** and **42b**; Fig. 15) led to a significant increase in fluorescence intensity (MeOH, especially **42b**) and a decrease of the blue shift of the maximum absorbance from 780 nm (**41**) to 696 nm (**42a**) and 694 nm (**42b**), respectively. In water, their fluorescence was very slow. The conjugation of **41** strongly enhanced its photodynamic and photothermal efficiency. Compared with **41** (50 μM), both **42a** and **42b** produced more singlet oxygen after laser irradiation (750 nm, 1 W cm$^{-2}$), especially **42b** which generated almost 2 times more ROS. Compound **42b** displayed significantly higher ΔT (above 20 °C vs below 20 °C), and the effect of IR780 was very slow.

## PDT and PTT in the combination therapy

However, despite the promising therapeutic potential of cyanine dyes, including targeting, their anticancer effect may not be sufficient for the complete tumor eradication. The application of phototoxic cyanine dyes could enhance used anticancer therapies (such as chemotherapy and immunotherapy). For example, in the cancer cells, higher temperature can decrease the expression of P-glycoprotein (P-gp) and multidrug resistance-associated protein 1 (MRP1) associated with drug resistance[236]. In the case of PDT, cancer cells that survived ALA-PDT were found to have reduced mitochondrial function and metastatic potential associated with increased HDAC activity[237,238].

Both PDT and PTT can strongly stimulate anti-tumor immunity (Fig. 17). Tumors destroyed by PDT and PTT release tumor antigens that can activate the anti-tumor immune response[239]. In addition to killing tumor cells, activated phototoxic agents remodel the tumor microenvironment and convert the immunogenic "cold" environment to a "hot" one, including activation of tumoricidal macrophages, dendritic cell (DC) maturation, infiltration of CD4+ and CD8 + T cells, and suppression of myeloid-derived suppressor cells (MDSC)[240,241].

This suggests that phototoxic agents could be a promising tumor sensitizer for subsequent immunotherapy. On the other hand, monoclonal antibody against cancer antigen may serve as a highly selective delivery system for phototoxic agents[239]. However, PTT stimulates higher overexpression of programed cell death 1 (PD-1; repression of T cells)[242,243]. Nevertheless, a combination with inhibitors of transforming growth factor β (TGF-β) or mitochondrial signaling can strongly decrease PD-1 level.

On the other hand, a combination therapy could increase the efficiency of PDT and PTT by targeting mechanisms of resistance against them. In the case of PDT, the survival of cancer cells may be associated with NF-kB, hypoxia-inducible factor 1-alpha, B-cell lymphoma family proteins (HIF-1α and Bcl-2, respectively) and mitochondrial HSP-60[244–248]. It was observed that PDT can induce an increase of NF-κB and heat shock proteins (HSP-60 and HSP-70)[247,249,250]. Since NF-κB is also associated with chemotherapy resistance and radioresistance, loss of therapy sensitivity cannot be excluded. In this context, it is worth mentioning that the anticancer effect of penthamethinium dyes in HCT116 cells was also associated with activation of NF-κB signaling[106]. Kevin et al. reported that p53-induced apoptosis was repressed by inhibition of NF-κB activity[251].

In addition, NF-κB is essential for the activation of the immune system[246] and targeting it non-specifically could be counterproductive. In addition, the role of other factors should not be neglected. Shen et al. reported that the expression of Bcl-2 interacting protein 3 (BNIP3), Bcl-2 and HSP-27 was increased in PDT-resistant cells relative to the parental HT29 cell line[252]. In contrast, the expression of glutamate dehydrogenase, hepatoma-derived growth factor, mutant p53 and mitochondrial genes encoding 16 s ribosomal RNA and CI subunit 4 was decreased. In melanoma cells, superoxide dismutase, microphthalmia transcription factor and NF-κB were associated with cell survival[142]. In PC-3 cells exposed to PDT, mitochondrial oxidative stress caused upregulation of nitric oxide synthase-2 and NO associated with resistance to PDT[253].

In the case of PTT, HSPs are one of the main factors that stimulate resistance to therapy[254]. Cancer cells exposed to sub-lethal heat can express HSP-70 (colocalized in mitochondria)[255,256]. Temperature-denatured protein can form aggregates[257], which can display cytotoxicity[258]. HSP-70 in a complex together with Bag-1 degrades the denatured protein, and the Bag-1/HSP-70 complex also has an antiapoptotic function[259]. In neuroblastoma cells (SH-SY5Y), BAG-1L protects cells from hypoxia/reoxygenation damage via interaction with HSP-70 and activation of PI3K/AKT pathways[260]. HSP-90 can also repair heat damage to proteins and is associated with a resistance to PTT[261,262]. Inhibition of HSP-90 enables targeting of cancer cells under mild-temperature conditions (45 °C), and decreases thermal effect side effects to surrounding tissues[82,83]. It should be mentioned that HS-90 plays an important role in the maintenance of NF-KB functionality[263].

## PDT and PTT in the combination therapy in the context mitochondrial targeting

Mitochondrial activity plays an important role in the chemoresistance of cancer and its suppression can be an effective way to enhance the efficiency of chemotherapy[264]. For example, reduced ATP production can be associated with lower expression of permeability glycoprotein (P-gp; which effluxes drugs from cells)[265].

On the other hand, some studied/used anticancer agents seem to increase the efficiency of mitochondrial PDT and PTT. Many terms for agents directly targeting mitochondria are used and intensively studied for the anticancer treatment. Some cytostatic agents, such as doxorubicin (oxidative phosphorylation)[266], cisplatin (mtDNA damage and ROS production)[267], and 5-fluorouracil (mitochondrial membrane depolarization)[268], have displayed therapeutic effects on mitochondrial functionality. For example, doxycycline can induce inhibition of autophagy and mitochondrial dysfunction, where it promotes photodynamic effects and presentation of tumor antigens[269].

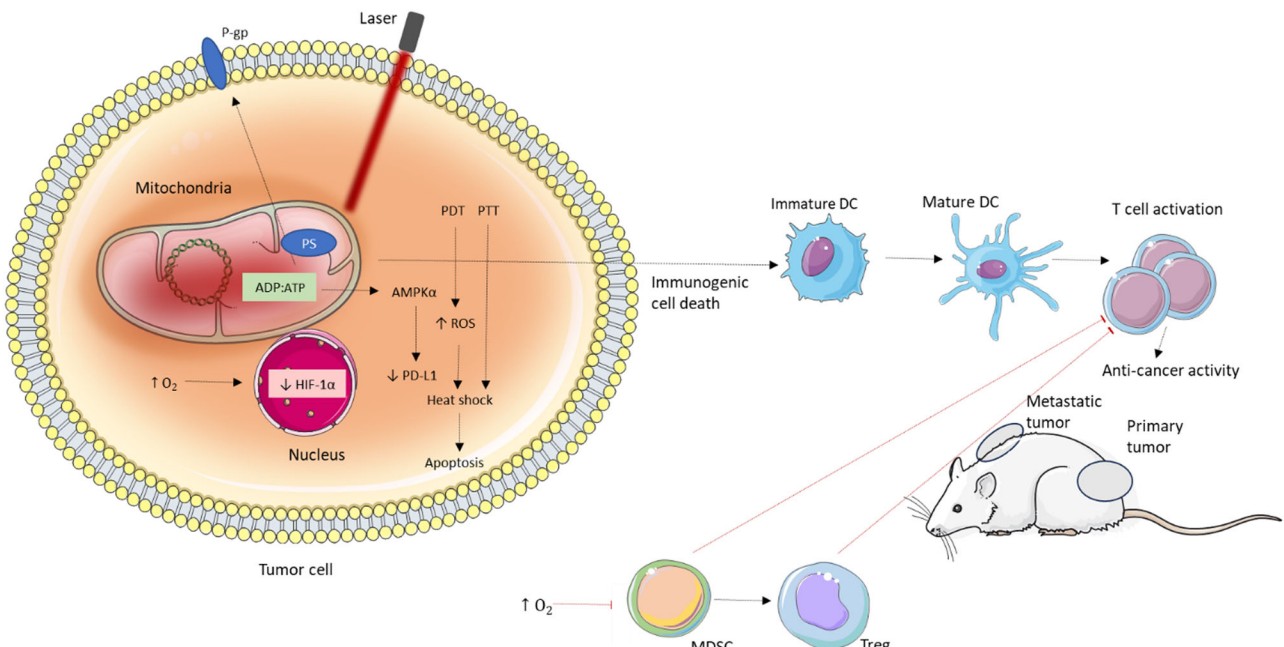

**Fig. 17 | Activation of anti-tumor immunity by mitochondria-targeted PDT and PTT.** Increased mitochondrial activity leads to higher ATP production and decreased oxygen levels, which are associated with cancer chemoresistance and a hypoxic phenotype. Mitochondria-targeted PDT and PTT result in ATP depletion, activating AMPKα (leading to the downregulation of PD-L1 expression). Increasing oxygen levels can suppress the hypoxic phenotype and HIF-1α activity in cancer cells. The generation of reactive oxygen species (ROS) and heat shock during PDT and PTT initiates apoptosis in cancer cells. It is also worth mentioning that mitochondria are a very suitable target for immunogenic cell death; for example, oxidized mitochondrial DNA is a potent immunostimulant that supports dendritic cell maturation. In the tumor microenvironment, higher oxygen levels (stimulated by repression of mitochondrial activity) suppress myeloid-derived suppressor cells (MDSCs) and subsequently regulatory T cells (Tregs). Both dendritic cell maturation and Treg suppression lead to the reactivation of T cell activity against primary and metastatic tumors. The figure was partly generated using Servier Medical Art, provided by Servier, licensed under a Creative Commons Attribution 3.0 unported license. AMPKα 5'-AMP-activated protein kinase catalytic subunit alpha-1, DC dendritic cells, HIF-1α hypoxia inducible factor 1 subunit alpha, MDSC myeloid-derived suppressor cells, mtDNA mitochondrial DNA, PD-L1 programmed death-ligand 1, P-gp P-glycoprotein, PDT photodynamic therapy, PTT photothermal therapy, PS photosensitizer, ROS reactive oxygen species, Treg regulatory T.

Some hexokinase-2 inhibitors (2-deoxyglucose and 3-bromopyruvate; clinically used anticancer drugs) have enhanced PDT-induced ROS production and oxidative damage in MDA-MB-231 breast cancer cells[270]. Currently, oxidative phosphorylation inhibitors such as lonidamine (CI and CII inhibitor)[271] and metformin (CI inhibitor)[272] have also been studied for combination with PDT/PTT. [195] In addition, it has been found that lonidamine can induce suppression of lactate efflux and intracellular acidification via inhibition of mitochondrial hexokinase II[273]. Metformin can target immunosuppressive mechanisms in tumor tissue, for example, inducing degradation of programmed death-ligand 1 (PD-L1; immunosuppression) in T cells[272]. A decrease in PD-L1 protein levels was found in the tumor tissue from the breast cancer patients after metformin treatment[274].

The application of NO scavengers (e.g., L-$N^G$-nitroarginine methyl ester) can decrease NO levels and thereby prevent the induction of resistance to PDT[253]. In this context, HSP inhibitors such as BIIB021 (HSP-90 inhibitor)[263] and gambogic acid (inhibitor of HSP-70 and HSP-90)[275,276] should also be mentioned, which can significantly increase the efficiency of phototherapy, especially PTT[275–278].

Other promising agents may be inhibitors of STAT3 and/or NF-κB activation. In addition to the nucleus, activated STAT3 can also be taken up into mitochondria, where it promotes mitochondrial functionality and maintains mitochondrial membrane potential[38]. Targeting its mitochondrial localization may increase the sensitivity of mitochondria to oxidative stress. It should be mentioned that the loss of mitochondrial membrane potential plays an important role in the mechanism of PDT and its recovery can lead to the survival of exposed cells[279]. Its disruption was observed even in the case of PTT[278,280]. In the case of NF-κB, its mitochondrial localization was associated with stimulation of oxidative phosphorylation (colon carcinoma cells)[39], nevertheless, other cell types may show opposite results[21]. Curcumin (direct inhibitor of STAT3 and NF-κB activation)[281–283] is studied in the combination with PDT and/or PTT/hyperthemia[256,284,285] Curcumin also decreases HSP efficiency in cancer cells[286] (including HSP-70)[256], effectively suppressing the hypoxic phenotype in tumor tissues and activating the anti-tumor immune system[287,288]. In leukemic cells, curcumin stimulates HSP-70 but decreases the level of Bag-1 (HSP-70 co-chaperone) in K62 cells[289].

Another strategy could be based on stimulating the Fenton reaction. Higher temperature also increases the efficiency of the Fenton-type reaction (˙OH generation from $H_2O_2$, which is highly abundant in mitochondria)[290]. Quiogue et al. reported that bafilomycin (vacuolar proton-pumping ATPase inhibitor) causes a collapse of the acidic lysosomal/endosomal pH gradients, thereby increasing cytoplasmic free $Fe^{2+}$ ions[291]. Subsequently, $Fe^{2+}$ ions can be taken up into mitochondria via the $Ca^{2+}$ uniporter[291,292]. Ru360 (inhibitor of mitochondrial calcium and iron uptake) significantly decreases the effects of bafilomycin on PDT-targeted mitochondria[292]. In this context, it should be mentioned that some polyphenols, such as EGCG, can significantly increase ROS production in the Fenton reaction by $Fe^{2+}$ ions[293].

Inhibition of mitochondrial respiration and the subsequent increase of cellular oxygen can also strongly influence other therapies, such as radiotherapy[294]. On the other hand, ROS production is part of the cytotoxic mechanism of radiotherapy and mitochondria are very sensitive to ionizing radiation. Specific targeting of tumor mitochondria can strongly enhance the sensitivity and efficiency of radiotherapy. Incorporation of a radiosensitizer (2-nitroimidazole) into the structure of heptamethine cyanine dyes has led to the development of new promising agents **43a** (called IR-83; Fig. 18) suitable also for PTT and PDT[295]. Compound **43a** (after irradiation; 808 nm) induces a sharp decrease in oxidative phosphorylation (lower tissue

**Fig. 18 | Structure of IR-83 (43a) and NHS-N782 (43b).** The heptamethine moiety within **43a** serves dual roles as agents for photodynamic therapy (PDT) and photothermal therapy (PTT). Incorporating 2-nitroimidazole, a radiosensitizer, inhibits oxidative phosphorylation, thereby amplifying mitochondrial and cellular sensitivity to both PDT and radiotherapy. The Salmonella-modified by **43b** effectively integrates mitochondria-targeted PTT with the stimulation of an anticancer immune response.

oxygen consumption and a remarkable increase in ROS levels) and thus sensitizes to radiotherapy. At the molecular lever, a decrease in the glutathione (GSH) context and NRF2 and heme oxygenase-1 expression and an increase in the NADP + /NADH ratio versus control and single irradiation were observed. In mouse model of Lewis lung carcinoma, **43a** displayed strong accumulation in tumor tissue (3 h) relative to heart, liver, spleen, kidney, muscle and intestine. The combined approach can decrease tumor mass relative to single therapies. Additionally, mice treated with **43a** in combination with laser irradiation and radiotherapy achieved longer survival time ( > 45 days) than mice receiving other treatments (20–35 days). Expression levels of HIF-1α and VEGF were obviously downregulated in the combined group compared to the other groups. A similar trend was observed in lung metastatic nodes.

Mitochondrial apoptosis induced by PDT is significantly involved in the induction of signal transduction pathways, which participate in the development of immune responses. The application of a mitochondria-targeted photosensitizer can effectively stimulate the antitumor immune response[296].

An interesting combination of photothermal and immune therapy was studied by Chen et al.[297]. The authors decorated the bacterial surface (Salmonella VNP20009) with heptamethine cyanine dyes **43b** (NHS-N782) and JQ-1 (BET inhibitor; reduction of PD-L1 expression on tumor and tumor-associated dendritic cells). The modified bacteria (Fig. 18) exhibited mitochondrial localization and an inhibitory effect on PD-L1 expression in B16-F10, the complexed agents displayed potent photothermal efficiency against cancer cells. Addition of irradiated cells stimulated a maturation of dendritic cells. In a mouse model, complexed agents displayed significant tumor accumulation and a very strong reduction in tumor mass and an increase in median OS (from 14 to 82 days) after irradiation compared to the control. This effect was associated with an increased level of CD8 + T cells, CD69 + T cells, CD45R + B cells, and natural killer cells in tumor tissue.

## Delivery system

The selectivity of cyanine dyes for tumor tissue is one of their pharmaceutically important properties. Small molecules, such as cyanine dyes, display a low half-life in the blood and are rapidly taken up by surrounding tissues. Nevertheless, some of them can bind to serum proteins such as serum albumin, and their selectivity is greatly enhanced by an increase in the retention effect[9,298]. Tumor tissue exhibits leaks and pores, and its lymphatic drainage is strongly limited. This causes high accumulation of macromolecules (e.g., proteins and nanoparticles) in tumor tissues. This strategy can lead to the design of novel promising anticancer/theranostic nanoagents[299–302]. A certain cyanine dyes (Fig. 19) have the ability to self-assemble into nanoparticles, facilitating effective accumulation in tumor tissues[303,304].

Due to the requirement for biodegradability (in the context of the prevention of long-term toxicity of nanoparticles) silica, sulfide or phosphate nanoparticles can be used. In the case of calcium phosphate, calcium overdose induced by hydrolysis of nanoparticles can severely impair mitochondrial functionality[305]. However, biodegradable supramolecular nano systems (e.g., lipids, polymeric micelles) are more frequently studied. The self-assembly process (molecular aggregation; in the process of nanoparticle formation) can control the physical, chemical, or biological properties of the anticancer agents by various mechanisms[306]. In this context, it is also worth mentioning perfluorinated micellar systems, which have the capability to transport large amounts of oxygen, thereby enhancing PDT efficiency[307]. Another potential strategy involves the complexation of cyanine dyes with natural biopolymers such as serum albumin, enabling their accumulation in tumor tissues based on the size of the complex[308,309]. Examples of tested cyanine dyes are shown on Fig. 20. These approaches are described in more detail in the following subsections.

## Self-assembly nanoparticles

Aggregation in aqueous medium is common behavior for cyanine dyes[310]. Nevertheless, some of them (depending on their structural motif) can form self-assembled nanoaggregates of suitable size for targeting tumors[303,304]. Their binding and aggregation are controlled by noncovalent interactions such as electrostatic, π–π stacking, hydrogen bonding, and hydrophobic interactions[310].

Compound **41** (called IR-780; Fig. 15) represents a promising structural motif for the preparation of new phototherapic agents. For example **41** TPE conjugates **44**-**46** (Fig. 19) were studied by Zhao et al.[303]. Compounds **44**-**46** exhibit different aggregation behaviors (hexane: EtOH, 1:1) in terms of particle size and dispersity, where **44** was in the state of a nanoparticle chain of about 50 nm but did not aggregate significantly. In contrast, **45** was tightly aggregated in large solid particles of 1 μm, whereas **46** dispersed in 4 nm and loosely aggregated to a size of 1 μm. At low concentration (10 μM, EtOH), **41** displayed significantly higher photothermal efficiency (ΔT ~ 10 °C) than **44**-**46**. However, at high concentration (50 μM), the highest ΔT ~ 18 °C was found for **45**, and the thermal efficiency of other dyes was comparable. **44** and **46** sometimes showed higher intracellular uptake into Hela cells than **41**, nevertheless, the uptake of **45** was strongly limited. In mitochondria, **44** and especially **46** had brighter emission and higher photostability than IR-780. The highest phototoxicity (808 nm, 0.5 W/cm², 5 min) was found for **46**, which (0.6 μM, 24 h) decreased the viability of cancer cells by approximately one-fourth. The photothermal efficiencies of **44** and **41** were sometimes lower and very low, respectively. In the case of normal cell lines (bEnd.3 and MLG and NIH-343), the dark toxicity and phototoxicity efficiencies of **41** and **44** were sometimes lower. In a mouse model with 4T1 tumor, the highest temperature in the tumor region was 64.6 °C and 83.7 °C (**41** and **46**; 300 μM, 100 μL) after irradiation (808 nm, 0.5 W cm⁻², 4 min.), respectively.

Zheo et al. reported a potent photodynamic and photothermal agent **47** (Fig. 19) based on the structural motif of bis-cyanine base[304]. In water, the $\lambda_{max}$ of **47** exhibited blue shift from 782 nm to 701 nm and extremely low $\Phi$ against DMSO (0.05% vs 8.34%). This difference was caused by the nanoaggregation of **47** in water (d ~ 200 nm). In DMSO, **47** displayed significantly higher photostability, ROS production and ΔT than **41**

**Fig. 19 | Examples of aggregated cyanine dye PS.** Certain cyanine dyes exhibit the propensity to self-assemble into nanostructures of specific sizes, influenced by their molecular structure or the presence of complementary cations (in this case of anionic cyanine dyes) for targeted tumor delivery. Notably, dyes **44** and **47** have the ability to form nanochains (approximately 50 nm) and nanoaggregates (around 200 nm in diameter), respectively, in contrast to **45** and **46**. Compound **48** is a conjugate with indomethacin, serving as a binding partner for cyanine dyes and a cyclooxygenase inhibitor. Upon interaction with cyclooxygenase 2, the self-assembled nanoparticles of **48** undergo disassembly, leading to an augmented photodynamic efficiency. Furthermore, in the presence of dodecyl(triphenyl)phosphonium and metal ions ($Fe^{2+}$ and $Mn^{2+}$), anionic cyanine dyes (**49** and **50**, respectively) form nanoaggregates with significantly higher phototoxicity compared to the individual dyes.

(Fig. 15), and especially ICG. After irradiation with 808 nm (0.3 W cm$^{-2}$) **47** (5 μM), **41** (10 μM) and ICG (10 μM) exhibit ΔT approximately 12, 8 and 2 °C, respectively. Compound **47** has a narrower HOMO–LUMO gap of 0.2028 eV than **41**. Since the nonradiative decay rate increases exponentially with the decreased optical energy gap[303], this difference could explain its strong photothermal efficiency. In A549 and HELA cells, it exhibits low dark toxicity and strong phototoxicity[304]. Compound **47** (2.5 μM) decreased cell viability approximately on 20% after irradiation with 808 nm (0.3 W cm$^{-2}$, 5 min). In a mouse model with 4T1 carcinoma, **47** (0.8 mg kg$^{-1}$) after irradiation with 808 nm (0.8 W cm$^{-2}$, 8 min; 24 h and 48 h after application) strongly reduced tumor mass. Tumor temperature increased to 52 and 57 °C and tumor tissue was strongly eradicated, the expression of Bcl-2 and Ki67 in tumor tissue was significantly decreased.

Prostaglandins (lipid signaling molecules) produced by cyclooxygenases (COXs) play an important role in the suppression of antitumor immunity[311]. Indomethacin (a potent inhibitor of COX-1 and COX-2 activity[312]; hydrophobic anion) also has a strong affinity for squaraine dyes[313]. Their conjugate (**48**; Fig. 19) forms self-assembled nanoparticles with low fluorescence emission. In cells with high expression of COX-2 (cancer cells), the nanoparticles disaggregate, inhibiting the enzyme activity and strongly increasing the fluorescence of the dye. In normal cells, this effect was not observed. Monomeric dyes are subsequently transported into mitochondria (Pearson's coefficient = 0.96). The $\Phi_\Delta$ of **48** in the absence and presence of COX-2 were $5.7 \times 10^{-3}$ and 0.035, respectively. In accordance with the above, the **48** displayed submicromolar phototoxicity against cancer cells, but efficiency of original squaraine dye was significantly smaller.

**Fig. 20 | Photodynamic and photothermal cyanine dyes tested in the form of nanoparticles and biopolymer complexes.** Nanoparticles play a crucial role in augmenting the therapeutic efficacy of various anticancer agents, such as photoactive dyes. In this context, supramolecular systems like liposomes (**51a**) and polymeric micelles (**51b** and **52**) are extensively studied. Encouraging results have also been demonstrated with the biopolymer application of serum albumins (**54a**, **54b**, and **55**) and hyaluronic acid (**56**). Additionally, inorganic nanoparticles such as molybdenum disulfide nanoflakes (**57a**, **57b**) and Ag₂S quantum dots (**58**) have shown promise in this area. Compounds like **52** and **55** exhibit other anticancer activities, such as reducing glutathione levels (a defense against oxidative stress) and inhibiting oxidative phosphorylation, respectively. Notably, in the case of the no-active compound **52**, its reaction with glutathione results in the release of active PS **53**.

Li et al. reported a new type of anionic cyanine dye (**49**; Fig. 19) that can form J-aggregates in the presence of hydrophobic cation such as dodecyl(triphenyl)phosphonium[314]. Nanoaggregations cause a strong red shift of $\lambda_{max}$ (~ 50 nm). In the case of pyridinium (**49a**) and sodium ion (**49b**), this effect was insignificant. The photodynamic efficiency of **49** (EtOH) alone

was lower than that of chlorin. However, **49c** nanoparticles sometimes displayed higher production of $^1O_2$. Nanoparticles **49a** and **49b** did not generate any $^1O_2$. On the other hand, their photothermal efficiency (655 nm, 0.3 Wcm$^{-2}$, 5 min) was better ($\Delta$T ~ 25 °C vs 17 °C). DFT calculations suggest that J-type dimers decrease the singlet-triplet energy gap $\Delta E_{ST}$

compared to the monomeric dye (0.40 eV vs. 0.64 eV) by reducing the energy level of the $S_1$ state. Usually, with the $\Delta E_{ST}$ below 0.3 eV, the ISC process will be favorably triggered[315]. In 4T1 cells, the highest phototoxicity was found for **49c** nanoparticles[314]. In a mouse model, **49c** nanoparticles (200 μg mL$^{-1}$, 100 μL) caused a very strong reduction of tumor mass after irradiation.

Cyanine dyes can aggregate not only by themselves, but can be used to deliver, for example, anticancer agents such as BIIB021 (HSP-90 inhibitor)[263]. Targeting HSP-90 can significantly improve the anticancer effects of PTT and its usability even at lower temperatures and therapy with decreased damage of surrounding normal tissues. A clinical trial of BIIB021 is currently underway[316]. Zhang et al. tested self-assembled nanoparticles of PEG conjugate with **41** (IR-780, Fig. 15; 25 mg) for PTT and delivery of BIIB021 (1 mg) in a breast cancer model[278]. The nanoparticles (20 μg ml$^{-1}$) displayed mild dark toxicity, but strong phototoxicity ( ~ 80% vs ~50% cell viability) after irradiation with 808 nm (1.0 W cm$^{-2}$, 10 min). In the case of nanoparticles containing only **41** conjugate, the cell viability was approximately 60%. In both cases, ΔT was below 15 °C. The photothermal effect was associated with a decrease of $\Delta\Psi m$ and activation of the mitochondrial apoptotic pathway. In a mouse model with MCF-7 carcinoma, the application of nanoparticles (5 mg kg$^{-1}$) after irradiation with 808 nm (1.0 W cm$^{-2}$, 10 min) caused ΔT 12 °C (tumor tissue) and below 8 °C (surrounding normal tissues). The tested nanoparticles with and without BIIB021 caused a strong decrease in tumor volume, and significantly higher effects were observed for nanoparticles with BIIB021.

In the case of cyanine dyes with anionic groups (e.g., sulphate or carboxy), their possible interaction with metal ion should be also considered. For example, Lv et al. demonstrated that photophysical properties and phototoxicity can also be influenced by complexed metal ions (e.g., $Fe^{2+}$ and $Mn^{2+}$; Fig. 19)[317]. In the presence of metal ions, **50** (Fig. 19) forms H aggregates via carboxy groups. This aggregation is linked to a blue shift (from 800 to 720 nm) and a reduction in fluorescence intensity. The hydrodynamic size of both aggregates was approximately 200 nm in PBS at pH 7. The metal aggregates, particularly those with $Fe^{2+}$, exhibited significantly enhanced thermal stability and photostability compared to **50** under laser irradiation (808 nm; 0.3 W cm$^{-2}$, 10 min). While both aggregates sometimes showed a lower increase in fluorescence of singlet oxygen sensor green (SOGS, $^1O_2$ probe)[318] than free dyes, $Mn^{2+}$ aggregates notably displayed a significant increase in $O_2^{\bullet-}$ production compared to free dyes. $Fe^{2+}$ aggregates exhibited high phototoxicity (808 nm, 0.4 W cm$^{-2}$, 10 min), with a 90 μM dose of $Fe^{2+}$ aggregates resulting in the death of over 80% of 4T1 cells, whereas $Mn^{2+}$ aggregates and free dye only caused approximately 70% and 20% cell death, respectively. Interestingly, in the absence of light, $Mn^{2+}$ aggregates only killed half the number of cells compared to $Fe^{2+}$ aggregates.

## Liposomes

Liposomes (closed colloidal structures) containing self-assembling lipid bilayers (e.g., phospholipids) and an aqueous core have long been studied for PS transport[319]. Hydrophilic agents are transported into the core and lipophilic agents, mostly PS, are transported into the lipid membranes. Phosphocholine liposomes (due to their thermal lability) can be a very promising system for targeted transport of photothermal agents[277,320]. They are stable at body temperature (37 °C), but undergo a phase transition after mild hyperthermia treatment (at 41–42 °C) that allows rapid release of loaded agents[321].

Liu et al. studied phosphocholine lyposomes with and without a small amount of triphenylphosphonium conjugate for the transport of **47** (Fig. 19) and gambogic acid (inhibitors of HSP-90; **47&GA@TP** and **47&GA@LP**, respectively)[277]. **47&GA@LP** displays lysosomal localization, whereas **47&GA@TP** accumulates in mitochondria (A549 cells). After irradiation with an 808 nm laser (0.3 W cm$^{-2}$, 5 min), **47@TP** showed a decrease in the protein levels of HSP-70, HSP-90 and Bcl-2. **47&GA@TP** caused a similar effect, however, the protein level of HSP-90 was sometimes lower. On the other hand, irradiation of **47@LP** caused an increase in the protein levels of HSP-70 and HSP-90. In the case of **47&GA@LP**, a decrease in Bcl-2 expression was observed. Mitochondrial localization was also associated with higher ROS production/ destabilization of mitochondria (due PDT and PTT), leading to lower production of ATP, which is necessary to maintain HSP functionality[322]. According to the above, the highest phototoxicity was observed for **47&GA@TP** (6 μg ml$^{-1}$), the cell viability was approximately 30% relative to the control[277]. In a mouse model with A549 carcinoma, the irradiated **47&GA@LP** displayed the highest decrease in tumor mass compared to the other tested liposomes. The expression of Bcl-2, HSP-90 and HSP-70 was dramatically downregulated and the inhibition of HSP-70 was significantly stronger than that of HSP-90. Nevertheless, ΔT after irradiation (808 nm, 0.6 W cm$^{-2}$, 10 min) was the lowest.

To target the hypoxia phenotype, PEG-PCL liposomes (60 nm) containing combination of **41** (Fig. 15) and metformin (direct inhibitor of CI)[323] (1:1; **41&Me@NP**) were studied[324]. **41&Me@NP** (1 mg ml$^{-1}$) displayed potent ROS production and ΔT (20.4 °C to 45.8) under irradiation with 808 nm (0.75 W cm$^{-2}$, 5 min; PBS). The photothermal conversion efficiency was approximately 34%. The dark toxicity (MKN-45P gastric cancer cells) of **41&Me@NP** was associated with a decrease in CI activity and $\Delta\Psi m$ and an increase in oxygen levels. Irradiated **41@NP** (4 μg ml$^{-1}$) stimulated a hypoxia phenotype, and the protein level of HIF-1α was approximately doubled. Nevertheless, **41&Me@NP** decreased the expression of HIF-1α to one-third of the original value. In a mouse model with MKN-45P carcinoma, the combination of **41&Me@NP** (20 μg, **41**) and irradiation with 808 nm (1 W cm$^{-2}$, 5 min) led to an approximately threefold increase in oxygen-saturated hemoglobin protein, and the protein level of unsaturated hemoglobin was decrease to one-third. Nevertheless, without irradiation, no significant changes in the haemoglobin context were observed. In the case of **41@NP**, the opposite trend was observed. HIF-1α protein was increased approximately six times after irradiation of **41@NP**, but only 2.5x in the case of **41&Me@NP**.

Liposomes (**41&SB@LP**) containing **41** (Fig. 15) and SB-505124 (TGF-β pathway inhibitor) in the combination with PD-1 antibody were studied in the treatment of breast cancer[242]. **41&SB@LP** displayed strong uptake and mitochondrial localization into 4T1 cells, and liposomes accumulated in the center in a 3D spheroid model. However, in the case of SB@LP, the uptake into 4T1 cells and spheroids and mitochondria was strongly lower. After irradiation with 808 nm (1 W cm$^{-2}$, 5 min), **41&SB@LP** and **41@LP** displayed strong, comparable phototoxicity against 4T1 cells (viability was lower than 10%). However, in a mouse model with 4T1 tumor, **41&SB@LP** showed a significant antitumor effect against **41@LP**. This phenomenon was probably caused by an increase in populations of CD4+ and CD8 + T cells and a decrease in Treg infiltration. Also, the increase in PD-1 protein level was slower compared to the single therapy. The combination of **41&SB@LP** with PD-1 antibody displayed a significant improvement in antitumor efficiency against, primary, distant and metastatic tumors. For example, at day 60 of treatment, OS was 40 and 60% for **41&SB@LP** and its combination with PD-1 antibody, respectively.

Thermosensitive liposomes (**51a&CAT@TSL**) containing **51a** (called also IR-820; Fig. 20) and catalase were tested for the treatment of cutaneous squamous cell carcinoma[325]. Incorporation of **51a** into liposomes had a strong effect on its UV-Vis spectrum, $\lambda_{max}$ of **51a** was 690 nm (aqueous solution). However, the liposomal formulation displayed two comparable peaks ($\lambda_{max}$ before and after 800 nm). Liposome **51a&CAT@TSL** (125 μg mL$^{-1}$) showed strong photothermal efficiency, the temperature increased to 60 °C after irradiation with 808 nm (1.5 W cm$^{-2}$, 6 min.). Mitochondrial and lysosomal localization was observed in A431 cells. Under hypoxia conditions, the efficiency **51a@TSL** was significantly limited, however, **51a&CAT@TSL** exhibited strong production of oxygen. In mice, **51a&CAT@TSL** (6.4 ug) selectively accumulated in tumor tissue, decreased HIF-1α expression by half, and virtually attenuated tumor growth after irradiation with 808 nm (1.5 W cm$^{-2}$, 10 min). However, the efficiency of **51a@TSL** (limited PDT) was sometimes decreased. In the case of lower energy irradiation of about 1.0 W cm$^{-2}$ (limited PTT), a similar effect was observed.

Although liposomes containing transported agents are localized in lysosomes after uptake, this does not necessarily imply the impossibility of mitochondrial targeting. For example, Shi et al. reported that lecithin liposomes with **1c** (Bromated penthamethine; Fig. 5) and paclitaxel (**1c&Pa@LP**) were taken up into cells by the endocytosis approach and localized in lysosomes[76]. Nevertheless, within 90 min, **1c** was relocalized in mitochondria. The application of **1c&Pa@LP** (Paclitaxel: **1c** ~ 1uM) and irradiation with 660 nm (6 J cm$^{-2}$) strongly decreased the viability of paclitaxel-resistant A549 cells (20% of control cells). This effect was associated with a decrease in the P-gp expression and ATP levels, and the effect of **1c** 5@Li was significantly slower. In the case of Pa@LP, or **1c@LP** without irradiation, an increase in P-gp protein level was found. In a mouse model with paclitaxel-resistant A549 cells and especially 4T1 tumor, these applications sometimes reduced tumor volume relative to a control and single therapy.

A lipid-like delivery system can also be obtained from the circulating cells. This approach could significantly reduce immune clearance and prolong the biological half-life of the delivered drug compared to conventional delivery systems[326]. Mai et al. prepared lipid nanoparticles derived from platelet membranes for the transport of **41** (Fig. 15) and metformin (**41Met@NP**; d = 135 nm)[327]. In 4T1 cells (dark), **41&Met@NP** induced a decrease in mitochondrial respiration and hypoxia suppression. After irradiation with 808 nm (1 W cm$^{-2}$, 3 min), significantly higher ROS production was observed for **41&Me@NP** than for **41@NP**. Irradiated 4T1 cells (in the same trend) induced the maturation of bone marrow derived dendritic cells (BMDC) associated with HMGB1 and IL-12 production. In a mouse model with 4T1 tumor, the combination of **41&Met@NP** (16 mg kg$^{-1}$ Met, 20 mg kg$^{-1}$ **41**) and irradiation with 808 nm (1 W cm$^{-2}$, 10 min) led to the suppression of primary tumor growth and metastatic activity in the lung. This effect was associated with a strong boost of antitumor immunity (DC maturation, decrease in frequency of Treg and MDSC) and an increase in tumor oxygen level. Nevertheless, in the case of **41@NP** or **41**, there was an increase in Treg and MDSC levels. In this context, it should be mentioned that metformin, in addition to inhibiting CI, can also have an anticancer immunity, such as by direct activation of PD-L1 degradation in T cells[272].

## Polymeric micelles

Polymeric micelles are formed by the self-aggregation of amphiphilic block or graft copolymer/copolymers in an aqueous environment[328]. They have a hydrophilic corona (hydrophilic structural motif such as PEG, polyoxazolines, chitosan, dextran, and hyaluronic acids) on their surface and a hydrophobic core in the middle of micelles. PS (mostly hydrophobic agents) are captured in the core of the micelles by hydrophobic and electrostatic interactions.

The use of nanoparticles for the combined application of cyanine dyes and mitochondria-targeting agents is a promising method in cancer treatment[327,329,330]. This strategy can significantly increase their effect on mitochondrial functionality. For example, Wen et al. prepared poly(lactic-co-glycolic acid) micelle **41&3BP@PLGA** with **41** (Fig. 15) and 3-bromopyrate (an inhibitor of mitochondrial respiration)[272,329]. Fluorescence microscopy showed that **41** exhibited a strong effect on micelle distribution. DiI-labeled 3BP@PLGA ($\lambda_{ex}/\lambda_{em} = 549/565$ nm) sometimes displayed lower fluorescence in breast cancer cells and their mitochondria than the corresponding **41&3BP@PLGA**. The same trend was observed for the increase in intracellular levels and activation of the mitochondrial apoptotic pathway (in dark). Strong phototoxicity of **41** was observed after irradiation, and the synergy of 3BP caused that **41@PLGA** (200 ug/ml) decreased cell viability (4T1) by approximately 60%, while irradiated **413&BP@PLGA** decreased cell viability by approximately 80%. In the case of 3BP@PLGA and **41&3BP@PLGA** (dark), the cell viability was below 80% and below 70% of the control, respectively. In a mouse model with 4T1 carcinoma, **41&3BP@PLGA** displayed a strong antitumor effect after irradiation with 808 nm (1 W cm$^{-2}$, 5 min)[329]. Its effect was associated with a

decrease in the protein level of HIF-1α (1/4 control groups) and a strong increase in oxygen level in tumor tissue.

Self-assembled polymeric micelles (PF127) were studied as a delivery system for the heptamethine cyanine dye **51b** (me-IR825; Fig. 20)[331]. The prepared nanoparticles **51b@NP**) display two excitation maxima (550 and 780 nm; PBS). The position of $\lambda_{max}$ is similar ( ~ 750 and 850 nm). Compound **51b** itself displays only one absorption maxima in ethanol (828 nm), but two (before and after 800 nm) in water. Emission intensities of **51b@NP** in water at both 610 and 845 nm were stronger than those of **51b** in water but weaker than those of **51b** in ethanol ($\Phi$ of < 1%). In the coculture (A549 and ATII), **51b@NP** shows strong red fluorescence signals in cancer cells and slower signals in normal cells. Micelle **51b@NP** (1 mg/ml) displayed significant photothermal efficiency ($\Delta$T ~ 25 °C) after irradiation with 808 nm (1.0 W cm$^{-2}$, 5 min). In a mouse model with U14 cervical carcinoma, the tumor growth was virtually arrested and tumor tissue temperature was significantly higher after irradiation than in the control groups ($T_{max}$ = 47.5 vs 39.6 °C). However, the dark cytotoxicity of **51b@NP** was not significant.

GSH increases the antioxidant capacity and the resistance to oxidative stress in many cancer cells[332]. An interesting overcoming of this approach was found by Yang et al.[333]. The authors prepared di-cyanine (**52**; Fig. 20) pro-PS with a disulfide bridge. Compound **52** does not exhibit any significant photodynamic activity, however, in the presence of GSH, the disulphide bond is cleaved and Cy7-S − NH2 (**53**; potent PS) is generated (you can see in Fig. 21). The GSH reaction also strongly alters spectral properties of the dye, such as the position of emission (from 755 to 800 nm) and excitation maxima (from 630 to 780 nm). In vitro, **52** nanoparticles (**52@NP**; POEGMA-b-PDPA) were found to significantly decrease cytoplasmatic GSH levels prior to uptake of **52** by mitochondria. After irradiation with 808 nm, the amount of nanoparticles corresponding to 4 uM of **52** decreased the cell viability (HepG2 cells) by approximately half. In a mouse model, under irradiation with 808 nm (1 W cm$^{-2}$, 5 min), **52@NP** (20 mg kg$^{-1}$) decreased tumor weight more than an order of magnitude, whereas the corresponding dose of **52** alone reduced tumor mass by approximately half.

A very promising system for the delivery of cyanine dyes could be designed based on phospholipid micelles. Phospholipid micelles with **41** (Fig. 15) were studied for tumor recognition in the U87M2/luc orthotopic tumor model[334]. High fluorescence was observed in the mouse brain 24 h after injection and lasted for at least 4 days. Skin and kidney emitted less fluorescence, while no significant fluorescence signal was detected in other organs (lung, heart and liver, muscle, spleen, stomach and intestines). In contrast, in the control group, a significant fluorescence signal was detected in the normal brain.

**Perflorinated micellar system**. A nanoemulsion of perfluoropolyether in water could be an effective system for targeting hypoxia cells and tumors[307,335,336]. Perfluoroalkyl can transport high levels of O$_2$ and the efficiency of PDT is significantly improved. In the case of **41** (Fig. 15), perfluoropolyether nanoemulsion significantly increased production of singlet oxygen and phototoxicity against cancer cells[335]. The efficacy of this approach can be improved by combining it with an inhibitor of mitochondrial respiration. A nanosystem with a perfluoralkyl core with **41** and atovaquone sometimes decreased tumor weight and suppressed HIF-1α expression after NIR irradiation in mice with CT26 colorectal tumor[307].

Micelles based on soybean oil and 1,11-diperfluoro-tert-butoxyundecane (F-oil) were studied for targeted transport of tamoxifen and conjugates of fluorinated amphiphiles and heptamethine dyes **42b** (Fig. 15; derived from **41**) (**42b&Tm@Mi**)[235]. The micellar formulation of **42b** had a significant impact on the photophysical properties. Soybean oil reduced self-quenching of **42b** in water and enhanced the fluorescence emission. F-oil was used to improve the $^{19}$F MRI signal intensity without introducing chemical shift artifacts and oxygen solubility. An increase in its

**Fig. 21 | Reaction of Dcy7 (compound 52) with glutathione (GSH).** While di-cyanine **52** does not show significant fluorescence emission or photodynamic efficiency, in the presence of GSH, it is cleaved to form the photoactive probe **53**.

photodynamic efficiency was also observed. However, after the formulation of **42b** into nanoparticles, ΔT was less than 16 °C, whereas comparable amounts of **42b** showed ΔT below 20 °C. In a mouse model with MCF-7 tumor, **42b** &Tm@Mi displayed long-term accumulation (10 days) in the tumor tissue. Fluorescence observed in other parts of body, such as heart, liver, spleen, lung and kidney, was sometimes lower. After irradiation, the temperature in the tumor tissue was higher than 60 °C and the tumor was eliminated.

### Biopolymer

The interaction of cyanine dyes with serum proteins such as serum albumin can significantly affect their selectivity. Tumor cells require a significantly higher supply of energy and building blocks (serum albumin represents both) necessary for their maintenance and proliferation[337]. Thus, cyanine dyes bound to serum albumin can display higher uptake into tumor cells. Therefore, cyanine dyes with higher affinity to albumin may exhibit higher tumor accumulation and therapeutic efficiency[308].

Tan et al. reported that **54a** (Fig. 20; ICG derivative) displays significantly higher affinity to HSA than ICG[309]. In addition, pH shift (from 7.22 to 6.02) or higher temperature (50 °C) caused a decrease in the stability of their complex (**54a**@HSA). In 4T1 cells, **54a**@HSA increased (in the dependence on irradiation time) the activation of caspase-3 and 9, and cytochrome c and Hsp70 expressions were increased. In the dark, a potent effect on cell viability (IC$_{50}$ ~ 16 μM, 24 h) was also observed. This suggests that **54a** combination of chemotherapy, PDT and PT, has an anticancer effect. The hydrodynamic diameter of **54a**@HSA was 15 nm, indicating its selective accumulation in tumor tissue and potent antitumor effects. Furthermore, **54a**@HSA after laser irradiation (808 nm, 1.5 W cm$^{-2}$, 8 min) caused a significant reduction in tumor weight in a mouse model with QBC-393, Hela and 4T1 carcinomas. For example, in a mouse model with 4T1, the temperature in the tumor tissue was between 60-70 °C and an excellent therapeutic effect (complete tumor eradication and strong increase in OS) was observed. Mice from control groups and non-irradiated groups with the applied formulations were all dead before day 30 (post-injection), nevertheless, all mice exposed to the combination of formulated agent were alive on day 90 of the experiment.

Another effective strategy that exploits the affinity of albumin for cyanine dyes in tumor targeting is based on the preparation of albumin nanoparticles. Albumin self-assembled nanoparticles as drug delivery system (DDS) for sorafenib (multi-kinase inhibitor) and **54b** (NHI-148; Fig. 20) in a mouse model with MB49 tumor were studied by Zhou et al.[338]. The antitumor efficiency was sometimes higher relative to a control and single therapy. In contrast to dyes alone, the prepared nanoparticles (**54b**&SRF@BSA) displayed approximately double selectivity for the tumor tissue compared to heart, liver, spleen, lung and kidney. In tumor tissue, nanoparticles **54b**&SRF@BSA decreased the level of HIF-1α protein ( ~ 25% relative to the control) after irradiation via inhibition of mitochondrial respiration and normalization of the tumor vasculature. ICD induced by the produced ROS enhanced T cell infiltration and improved their ability to kill tumor cells. The tumor immune suppression in the tumor microenvironment (TME) was also suppressed by decreasing vascular endothelial growth factor A (VEGF-A) protein levels.

BSA nanoparticles were tested for the transport of doxorubicin and **40** (conjugate of IR780 with triphenylposphonium; Fig. 15)[234]. Firstly, **40** was co-assembled with doxorubicin through hydrophobic interactions such as π − π interaction. This interaction was associated with a decrease in the intensity of the absorption peak of doxorubicin. In the case of the formation of BSA nanoparticles (**40**&Dox@BSA), a mild blue shift of **40** (in the form of doxorubicin complex) was observed. Nanoparticles **40**&Dox@BSA exhibited significantly higher photodynamic efficiency than **40** alone. However, the efficiency of the original **41** was significantly higher. The photothermal efficiency was comparable. Nevertheless, in 4T1 cells, **40** and especially nanoparticles **40**&Dox@BSA displayed strong ROS generation after irradiation (660 nm or 808 nm, respectively, 1 W cm$^{-2}$, 2 min.), the effect of **41** was negligible. In a mouse model with 4T1 tumor, **40**&Dox@BSA displayed significant phototoxicity and induction of immune system, such as an increase in the T cells (CD4+ and CD8 + )/Treg ratio. In the case of surgical extraction of the primary tumor, the nanoparticles virtually suppressed metastatic activity in the lung.

Compound **55** (Fig. 20), a conjugate of heptamethine cyanine dye and lonidamine (LND; CI and CII inhibitors)[271], displayed approximately a hundred times greater inhibitory activity against CI (IC$_{50}$ = 6.4 uM) and CII ( = 9.5 uM) than lonidamine alone[339]. HSA formulation of **55** displayed potent dark and phototoxic effects against breast cancer cells. The IC$_{50}$ value representing the dark toxicity for **55**@HSA was 1.59 uM against MB49 cells. In the case of **55**@HSA and of LND@HSA, the observed values were 25.92 and 262.15 uM, respectively. A similar trend was observed in the case of their phototoxicity. In cancer cells, the application of **55** (1 μM, dark) decreased the protein levels of HIF-1α and PD-L1 (repressor of T cells) via activation of 5' AMP-activated protein kinase. In co-culture of T cells and cancer cells (MB49 or CT26), cancer cells exposed to **55**@HSA were killed by uninhibited T cells. In a mouse model with MB49 carcinoma, **55**@HSA showed strong accumulation in primary and metastatic tumors relative to liver, spleen and kidney. Nanoparticles **55**@HSA (5 mg kg$^{-1}$ **55**) caused a strong reduction of tumor mass associated with downexpression of PD-L1 and HIF-1α in the tumor tissue. In the case of phototherapy (808 nm, 200 mW cm$^{-2}$, 1 min), the tumor was smaller, but the protein levels of PD-L1 and HIF-1α did not change significantly compared to the dark application.

Self-assembled HSA nanoparticles for the combined delivery of **36** (IR-825Cl called also dc-IR825; Fig. 13) and gambogic acid (HSP90 inhibitor; **36&GA@HSA**) were tested by Gao et al.[340]. In A549 cells, **36&GA@HSA** mainly accumulated in mitochondria (P = 0.96), significant colocalization was also observed in the lysosome (P = 0.24), ER (P = 0.67), and Golgi apparatus (P = 0.29). After irradiation with 808 nm (0.3 W cm$^{-2}$, 10 min), the fluorescence signal was removed into the cytosol and vacuolization of mitochondria was observed. Their irradiation caused a significant ΔT (25 °C, 10 ug ml$^{-1}$). In a mouse model with A549 carcinoma, **36&GA@HSA** completely suppressed the growth of tumor mass after irradiation, the efficiency of **36@HSA** and **GA@HSA** was lower.

Anionic polymers such as hyaluronic acid are suitable drug delivery systems for transporting cationic agents such as cyanine dyes. Due to CD44, hyaluronic acid (overexpressed in tumor tissue) exhibits tumor selectivity[341] and may also be intrinsic to the anticancer effect (depending on the length)[342]. The therapeutic potential of hyaluronic nanoparticles with **56** (Fig. 20; **56@HA**) was tested by Thomas et al.[173]. Nanoparticles **56@HA** exhibited significantly higher dark stability (in PBS) and higher ROS production (in Hela cells) than **56**. The Pearson's co-localization coefficients for mitochondrial localization (in Hela cells) were 0.73 and 0.77 for **56** and **56@HA**, respectively. In these non-cancerous HeK293T cell lines with significantly lower CD44 expression compared to Hela cells, **56** showed strong mitochondrial localization, however, **56@HA** displayed only negligible accumulation in HeK293T cell lines. The IC$_{50}$ of **56@HA** after irradiation with 808 nm (3 min, 200 mW cm$^{-2}$) was found to be 5–7 μM in both HeLa and MDA-MB-231 cells, however, in the case of HeK293T cells, **56@HA** (20 μM **56**) displayed only negligible phototoxicity. In a mouse model with SSC7 tumor, **56@HA** exhibited higher tumor selectivity than **56** and especially **41** and strong antitumor efficiency.

## Inorganic nanoparticles

Inorganic nanoparticles with solid cores have been studied as transport systems for anticancer agents containing PS. In addition, they may exhibit intrinsic photodynamic, or photothermal activity. Paradoxically, the main drawback of some promising inorganic compounds is their stability and thus low biodegradability. Only nanoparticles with biodegradable or renal clearable properties such as silica, calcium phosphate, oxide or sulfide nanoparticles, have reasonable chances for potential clinical translation[343].

Mesoporous silica nanoparticles modified by hyaluronic acid were studied for the transport of **40** (conjugate of heptacyanine dye with TPP; Fig. 15) and lactate oxidase (**40&LOD@Si-HA**)[344]. The prepared nanoparticles exhibit potent photodynamic and photothermal effect, although the photothermal effect of **40** alone is significantly higher than the corresponding amount of nanoparticles. Nevertheless, in the case of MCF-7 and especially 4T1 cells, **40&LOD@Si-HA** shows a stronger effect on the viability, colony formation, migration and invasion of cancer after irradiation with 660 nm (1 W cm$^{-2}$, 2 min) than the dyes alone. This effect was associated with downexpression of glutathione peroxidase 2, and matrix metalloproteases (MMP-2 and MMP-9) and overexpression of transferrin and an increase of iron levels. The effect of **40&LOD@Si** on cancer cells was significantly slower. In a mouse model with 4T1 carcinoma, the combination of nanoparticles (**40&LOD@Si** and especially **40&LOD@Si-HA**; 2 mg kg$^{-1}$ dye) with irradiation caused a strong reduction of the tumor mass and increase in OS of mice. On day 30 of the therapy, 60% and 80% of mice treated by **40&LOD@Si** and **40&LOD@Si-HA**, respectively, were alive. Nevertheless, all mice in the control group and mice treated with the combination of dyes and lactate oxidase were dead after day 20 and 30 of the therapy. The same trend was observed for the reduction of lung metastasis and decrease in expression of HIF-1α, MMP-2, and MMP-9 and lactate levels in the tumor tissue.

For the combination of PTT with mitochondria and endoplasmic reticulum-targeted PDT, molybdenum disulfide nanoflakes (MoS) with mitochondria and endoplasmic reticulum selective dyes (**57a** and **57b** (Fig. 20), respective; **57a&57b@MoS**) were tested[345]. To increase their selectivity for cancer cells, they were modified with hyperbranched glucose-substituted polyglycerol. Cancer cells can overexpress glucose transporter 1 (GLUT1), which can increase the intracellular uptake of glucose-modified compounds and nanoparticles[346,347]. MoS alone exhibit significant photothermal activity (ΔT > 20 C°; 50 ug ml$^{-1}$) after irradiation with 808 nm (1.0 W/cm, 8 min) and selectivity for cancer cells[345]. In Hela spheroids, **57a&57b@MoS** (30 ug/ml corresponding to the cumulative amount of both dyes) decreased the number of spheroids below 10% after irradiation. The efficiency of **57a@MoS** and **57b@MoS** was significantly lower, but higher reductions were sometimes found for **57a@MoS**. The combination of dyes alone had slow effects. At the intracellular level, the activation of C/EBP homologous protein pro-apoptotic signaling pathway (induced by **57b** PTT) and promotion of cytochrome C release from mitochondria (induced by **57a** PTT) were observed. Mitochondrial disfunction led to lower ATP production and thus suppressed MDR phenotype (lower Pg-P expression). In a mouse model, a high accumulation of **57a&57b@MoS** was found in the tumor tissue compared to spleen and especially to heart liver and lung. The combination of **57a&57b@MoS** (2 and 2 mg kg$^{-1}$) and the irradiation with 808 nm (1.0 W cm$^{-2}$, 8 min) suppressed the tumor growth and Kiel 67 (Ki-67; marker of proliferation), Pg-P and expression. The antitumor efficiency of **57a@MoS** (4 mg) and **57b@MoS** (4 mg) and especially the combination of the corresponding amounts of dyes alone was sometimes lower.

Ag$_2$S quantum dots (QDs) are notable for their lack of toxic heavy metals and extremely low solubility constant (Ksp = $6.3 \times 10^{-50}$), indicating high biocompatibility[348–350]. These QDs exhibit favorable spectroscopic properties with absorption and emission in the near-infrared (NIR) region and can be easily prepared and functionalized in water. Additionally, Ag$_2$S QDs are being explored as photothermal therapy (PTT) agents[348,351,352]. Celikbas et al. investigated the modification of Ag$_2$S QDs by hemocyanine **58** (Fig. 20)[352]. Anionic Ag$_2$S QD was functionalized with folate, targeting cancer cells overexpressing folate receptors (FR)[353], and PEG to enhance solubility[352]. Compound **58** with an amino group was loaded electrostatically onto Ag$_2$S QD. Upon irradiation (640 nm; 300 mW, 0.78 W cm$^{-2}$, 5 min), the temperature of QD (Ag 300 μg mL$^{-1}$) and free **58** solution (53 μg/mL) increased by approximately 13.6 °C and 6.7 °C, respectively, while **58@QD** reached a maximum temperature increase of 17.74 °C. ROS production was observed for **58** but not for the QD alone. Significantly higher cell uptake, ROS production, and phototoxicity of **58@QD** were noted in HeLa cells with FR overexpression compared to FR-negative A549 cells. Both **58** and **58@QD** exhibited strong mitochondrial localization (P = 0.82 and 0.84, respectively). In summary, QD acted as a mild PTT agent and a delivery system for **58**, enhancing its phototoxicity.

## Sonodynamic therapy—novel therapeutic applicability of phototoxic cyanine dyes

Cyanine dyes with photodynamic activity could also be used in the sonodynamic therapy (SDT). SDT exhibits a mechanism similar to PDT[354,355]. Specific agents called sonosensitizers produce ROS from oxygen after exposure of low-intensity ultrasound (US). Most sonosensitizers are derived from PS[354]. For example, porphyrins are the first and most commonly used sonosensitizers. Nevertheless, US, unlike light, is not absorbed by tissues and can penetrate soft tissue to the depth of tens of centimeters[356]. Therefore, other polyconjugate compounds such as curcumin[357] and cyanine dyes are intensively studied for SDT. It should also be mentioned that SDT show a strong synergistic effect when combined with PDT and PTT[358].

Dyes based on trifluoromethyl-heptamethine cyanine **59** (Fig. 22) represent a promising structural motif for the design of a combined mitochondria-targeted sonosensitizer and photothermal agent[359,360]. In vitro, dyes **59** (50 mg ml$^{-1}$) were capable to kill cancer cells (4T1) with high efficiency (80%) under NIR and (US).

Chondroitin sulphate modified by self-assembled carboxylated heptamethine cyanine **60** (derivative of **41**; Fig. 22) forms nanoparticles (hydrodynamic d ~ 200 nm)[361]. These nanoparticles display significantly elevated temperature ( > 25 versus <20 °C) relative to the same concentration of the original dyes (40 ug ml$^{-1}$) after irradiation (808 nm, 1.0 W cm$^{-2}$,

**Fig. 22 | Structure of sonodynamic heptamethine cyanine dyes.** Certain photoactive cyanine dyes can also function as sonosensitizers in anticancer treatment. Heptamethine dye **59** demonstrates a dual effect, serving as both a photothermal agent and a sonodynamic sensitizer. Additionally, carboxylated dye **60** can be employed in a combined approach involving photodynamic, photothermal, and sonodynamic therapy.

5 min). The nanoparticles (corresponding to 0.5 μM IR806) decrease cell viability (to 60%) after exposure to US or laser (PDT and PTT); the combination of irradiation and ultrasound reduces viability to approximately 30%. In a mouse model with PC-3 carcinoma, the nanoparticles exposed to laser irradiation and ultrasound displayed a strong therapeutic effect; the tumor mass was sometimes smaller relative to a single therapy mode, or fully eradicated.

## Future direction

Mitochondria represent an attractive target in anticancer treatment[10,146,362]. The studies discussed above (summarized in the Table 1) suggest that cyanine dyes are a promising structural motif designing mitochondria-targeting photodynamic and photothermal agents. However, these studies were conducted under varying conditions. Therefore, it would be beneficial to propose a systematic study of different dyes, tested under the same conditions to observe and compare their therapeutic and diagnostic effects.

The presented PSs have demonstrated potent photodynamic and photothermal effects. However, the antitumor efficacy of low molecular compounds can be hindered by their limited selectivity for tumors. This challenge can be effectively addressed by enhancing their accumulation in tumor tissues and therapeutic efficiency through the utilization of suitable drug delivery systems such as nanoparticles (refer to "Delivery system"). It is worth noting that certain cyanine dyes (**37a** and **44**)[221,309] exhibit high selectivity for tumor tissues due to their interaction with serum proteins, which are preferentially captured by tumor tissues[337]. Additionally, some reported PSs remain inactive until specifically activated in tumor tissues or cancer cells, thereby enhancing the selectivity of phototherapy. For instance, compounds **23** and **29** (alanine and phosphate ester) do not demonstrate significant phototoxicity; however, their photoactive forms (**24** and **27**, respectively) are released through enzyme hydrolysis by APN and ALP[72,73], which are overexpressed in tumor tissues[192,202].

In the context of selectivity, it is essential to consider the intracellular localization of cyanine dyes. Despite initial expectations, mitochondrial localization is not observed[73,181]. One potential solution involves the conjugation of cyanine dyes with the mitochondrial targeting TPP group. This strategy has proven effective in the development of photosensitizers with high mitochondrial localization, exemplified by compounds **3b, 7** and **40**[176,177,218]. However, the localization of the anionic dye **3d** was found to be comparable to that of the original **4d**[176]. Shi et al. reported that compound **5** with the TTP group exhibited significantly lower mitochondrial accumulation than compound **6** with chloroacetyl substitution[177].

Moreover, the anticancer efficacy of these agents may not always be sufficient, especially in hypoxic tumors where oxygen levels are significantly lower, leading to decreased efficiency of type II PDT, which is the primary modality for most tested or used PSs (you can see in Table 1). However, there is a growing focus on the development of type I PSs. In this line, it should be mentioned that hemicyanines display a high potential for the preparation of type I PS[73,197,203]. For instance, Zhao et al. developed hemicyanine PS **30b** with high mitochondrial selectivity and submicromolar phototoxicity[203]. Similarly, Zhang et al. demonstrated that hemicyanines (**25-29**) present a promising structural motif for the preparation of type I PS[73]. Alternatively, type II PS such as **41** can be combined with a perfluorinated delivery system to enhance oxygen transport for improved PDT efficacy[335].

In the realm of photoactive agents, a potential strategy involves combining PDT and PTT (further detailed in "Combination of PDT and PTT"). PTT can elevate oxygen levels in cancer cells, while PDT can enhance the effects of hyperthermia. In a mouse model with 4T1 tumors, the administration of dual photodynamic and photothermal agent **39g** effectively inhibited tumor growth and recurrence, achieving a 100% overall survival rate by the 60th day[225]. Nevertheless, even the combination of PDT and PTT cannot be sufficient for the complete treatment of cancer. Therefore, various strategies to improve their therapeutic efficiency (application of other anticancer effects of cyanine dyes, improvement in the tumor selectivity of cyanine dyes and their combination with other therapeutic agents and therapies) are intensively studied.

It is well known that cyanine dyes can strongly disturb mitochondrial functionality and targeting mitochondria is an effective strategy to improve the efficiency of PDT and PTT. Nevertheless, in the case of phototoxic agents, their non-toxicity without irradiation (at least at the concentration used) is usually required[97,110,179,217]. However, cyanine dyes, including compounds targeting mitochondria, are intensively studied in the cancer treatment. Depending on their tumor selectivity, their dark cytotoxicity might not be undesirable but therapeutically beneficial. Irradiation cyanine dyes in damaged mitochondria (at higher oxygen levels) may allow an effective combination of their dark toxicity and phototoxicity with synergic effect. In accordance with proposed strategy, NIR cyanine dyes **14a** and **14b** conjugated with 5′-deoxy-5-fluorouridine (cytotoxic agents)[128] and xanthene-cyanine dyes **16** with methyl-triazene moiety (DNA methylation)[129] have been prepared and successfully tested. Similarly, it is possible to use cyanine dyes that disturb mitochondrial functionality (inhibition of dihydroorotate dehydrogenase, STAT3 phosphorylation and mitochondrial respiration complexes)[107,124,126,127]. The promising potential of this strategy was shown by Huang et al.[339]. Compound **45** (conjugate of heptamethine dye with lonidamine; CI and CII inhibitor)[271] displayed a strong efficiency in vitro and in vivo, nevertheless, the main part of activity was due to its cytotoxicity. However, the alternation of mitochondrial functionality (a decrease in the oxygen consumption and ATP content) induced by cyanine dyes may not always manifest itself in the cell sensation[125].

Lastly, it is essential to acknowledge that the effectiveness of phototherapy can be significantly impeded by the transmission of light through tissues. On the other hand, the promising results of cyanine dyes in sonodynamic therapy could redefine a design of a cyanine sensitizer. Currently, heptamethines are rather preferred due to their selectivity for cancer tissues[363]. Appropriately designed hepthamethines (e.g., Cl substitution at the γ-position) such as **37a**, unlike corresponding pentamethines, can covalently bind to albumin in vivo[9]. However, longer polymethine chain length increases its flexibility, thus increasing the non-radiative dissipation of the excited state energy and reducing the triplet state quantum yield (critical for PDT)[178], although increasing the rigidity of polymethinium chain cannot always lead to an increase in the photodynamic efficiency[180]. Some pentamethines such as **12** with a more rigid polymethinium chain can display very potent phototoxicity (in the tens of nanomoles)[68,364]. Nevertheless, substitution of cyanine dyes by flexible electron rich aromatic groups can lead to an increase in their PDT and PTT efficiency[365]. Given the similarity of both therapeutic methods[366], pentamethines could be more

## Table 1 | Photodynamic and photothermal dyes with regard to mitochondrial targeting

| Tested PS | Monitored quantities/properties (Obtained results/data; experimental conditions) |
|---|---|
| **Photodynamic agents** | |
| **1a**[174] pentamethine | **Photophys** ($\lambda_{abs}/\lambda_{em}$ = 638/657 nm and $\Phi$ = 0.38; PBS), **$^1O_2$** ($\Phi_\Delta$ = 0.003; 660 nm-DPBF, 5 mW cm$^{-2}$, 20 min, DCM) and **Phototox** (IC$_{50}$ = 0.75 µM; 660 nm-MCF-7 cells, 20 mW cm$^{-2}$, 5 min) |
| **1b**[174] pentamethine | **Photophys** ($\lambda_{abs}/\lambda_{em}$ = 638/656 nm and $\Phi$ = 0.029; PBS), **$^1O_2$** ($\Phi_\Delta$ = 0.003; 660 nm-DPBF, 5 mW cm$^{-2}$, 20 min, DCM) and **Phototox** (IC$_{50}$ = 1.2µM; 660 nm-MCF-7 cells, 20 mW cm$^{-2}$, 5 min) |
| **1c**[174] pentamethine | **Photophys** ($\lambda_{abs}/\lambda_{em}$ = 634/654 nm and $\Phi$ = 0.025; PBS), **$^1O_2$** ($\Phi_\Delta$ = 0.015; 660 nm-DPBF, 5 mW cm$^{-2}$, 20 min, DCM), **IntraLoc** (P = 0.94; Mito, MCF-7 cells) and **Phototox** (IC$_{50}$ = 0.062 uM; 660 nm-MCF-7 cells, 20 mW cm$^{-2}$, 5 min) |
| **2a**[175] pentamethine | **Photophysis** ($\lambda_{abs}/\lambda_{em}$ = 680/721 nm and $\Phi$ = 0.25; DCM), **$^1O_2$** ($\Phi_\Delta$ = n.d.[a]; 808nm-DPBF, 330 mW cm$^{-2}$, 10 min, EtOH), **SOC** (0.028 cm$^{-1}$), **Phototox** (n.d./ 2.5 uM; 808nm-4T1 cells, 330 mW cm$^{-2}$, 10 min) |
| **2b**[175] pentamethine | **Photophysis** ($\lambda_{abs}/\lambda_{em}$ = 680/721 nm and $\Phi$ = 0.07; DCM), **$^1O_2$** ($\Phi_\Delta$ = 0.014; 808 nm, 10 min, 330 mW cm$^{-2}$, EtOH), **SOC** (0.23 cm$^{-1}$), **IntraLoc** (P = 0.94 and 0.49; Mito, 4T1 cells and CCCP treated 4T1 cells, respectively), **Phototox** (IC$_{50}$ = 1.5 µM, mitochondrial damage; 808 nm-4T1 cells, 330 mW cm$^{-2}$, 10 min), **Intratumorally** (2% and 30% tumor mass (normal and 5 mm deep, respectively); 3×0.8 mg kg$^{-1}$, 808nm-4T1 tumor mice, 330 mW cm$^{-2}$, 15 min) and **I.V.** (28% tumor mass and increase OS ( < 40 vs > 60 day); 3×0.8 mg kg$^{-1}$, 800nm-4T1 tumor mice, 330 mW cm$^{-2}$, 15 min) |
| **3b**[176] heptamethine | **Photophysis** ($\lambda_{abs}/\lambda_{em}$ = 627/746 nm; MeOH), **$^1O_2$** (strong response; 662nm-SOSG, 100 mW cm$^{-2}$, 5 min, DPBS), **ItraLoc** (mitochondria/cytosol = 6.58 and 6.31; NCI-H460 and MCF-7 cells, respectively), **Phototox** (80% and 60% apoptotic cells/50 uM[b]; 662 nm-NCI-H460 and MCF-7 cells, 100 mW cm$^{-2}$, 5 min, respectively), and **I.V.** (~30% tumor volume and mass; 0.1 mM, 200 µL, 662 nm-NCI-H460 tumor mice, 100 mW cm$^{-2}$, 5 min) |
| **3d**[176] heptamethine | **Photophysis** ($\lambda_{abs}/\lambda_{em}$ = 618/748 nm; MeOH), **$^1O_2$** (strong response; 662nm-SOSG, 100 mW cm$^{-2}$, 5 min, DPBS) and **ItraLoc** (mitochondria/cytosol = 1.05 and 1.35; NCI-H460 and MCF-7 cells, respectively) |
| **3f**[176] heptamethine | **Photophysis** ($\lambda_{abs}/\lambda_{em}$ = 630/750 nm; MeOH), **$^1O_2$** (strong response; 662nm-SOSG, 100 mW cm$^{-2}$, 5 min, DPBS) and **ItraLoc** (mitochondria/cytosol = 2.45 and 1.48; NCI-H460 and MCF-7 cell, respectively) |
| **3a**[176] heptamethine | **Photophysis** ($\lambda_{abs}/\lambda_{em}$ = 636/754 nm; MeOH) and **$^1O_2$** (slow response; 662nm-SOSG, 100 mW, 5 min, DPBS) |
| **3c**[176] heptamethine | **Photophysis** ($\lambda_{abs}/\lambda_{em}$ = 626/746 nm; MeOH) and **$^1O_2$** (slow response; 662nm-SOSG, 100 mW cm$^{-2}$, 5 min, DPBS) |
| **3e**[176] heptamethine | **Photophysis** ($\lambda_{abs}/\lambda_{em}$ = 641/752 nm; MeOH) and **$^1O_2$** (slow response; 662 nm-SOSG, 100 mW cm$^{-2}$, 5 min, DPBS) |
| **5**[177] heptamethine | **Photophysis** ($\lambda_{abs}/\lambda_{em}$ = 802/810 nm and $\Phi$ = 0.0879; DMSO), **$^1O_2$** ($\Phi_\Delta$ = 0.0081; 808 nm-DPBF, 1.5 W cm$^{-2}$, 10 min, EtOH), **Phototox** (IC$_{50}$ = 21.33 uM; 800 nm-4T1 cells, 1.0 W cm$^2$, 30 s) and **Intratumorally** (~ 33% tumor volume and mass; 200uM, 808 nm-4T1 tumor mice, 1.0 W cm$^{-2}$, 30 s) |
| **6**[177] heptamethine | **Photophysis** ($\lambda_{abs}/\lambda_{em}$ = 802/810 nm and $\Phi$ = 0.08842; DMSO), **$^1O_2$** ($\Phi_\Delta$ = 0.0079; 808 nm-DPBF 1.5 W cm$^{-2}$, 10 min, EtOH), **Phototox** (IC$_{50}$ = 10.76 µM; 800 nm-4T1 cells, 1.0 W cm$^{-2}$, 30 s) and **Intratumorally** (~1/6 tumor volume and mass; 200uM, 808nm-4T1 tumor mice, 1.0 W cm$^{-2}$, 30 s) |
| **7**[177] heptamethine | **Photophysis** ($\lambda_{abs}/\lambda_{em}$ = 802/810 nm and $\Phi$ = 0.08818; DMSO), **$^1O_2$** ($\Phi_\Delta$ = 0.0079; 808 nm-DPBF, 1.5 kW cm$^{-2}$, 10 min EtOH), **Phototox** (IC$_{50}$ = 6.28 uM; 800 nm-4T1 cells, 1.0 W cm$^{-2}$, 30 s) and **Intratumorally** (< 10% tumor volume and mass; 200uM, 808nm-4T1 tumor mice, 1.0 W cm$^{-2}$, 30 s) |
| **8**[179] heptamethine | **Photophysis** ($\lambda_{abs}/\lambda_{em}$ ~ 654/700 nm; polar and nonpolar solvents), **$^1O_2$** (strong response; 600 nm-DPBF, 10 mW cm$^{-2}$, 5 min, DCM), **ItraLoc** (P = 0.914; Mito, MCF-7), **Phototox** (~50% cell viability/ 2.0 µM; 660nm-MCF-7 cells, 90 J cm$^{-2}$) and **ROS Scavenger** (NaN$_3$) |
| **9b**[181] squaraine pentamethine | **Photophysis** ($\lambda_{abs}$ = 636 nm; DMEM), **$^1O_2$** (strong response/ 5 µM; xenon lamp-DPBF, 40 min) and **Phototox** (IC$_{50}$ = 0.673 uM; 640 nm-PC-3 cells, LED, 7 min, PBS) |
| **10a**[181] squaraine pentamethine | **Photophysis** ($\lambda_{abs}$ = 647 nm; DMEM), **$^1O_2$** (strong response/ 5 µM; xenon lamp-DPBF, 40 min) and **Phototox** (IC$_{50}$ = 0.091 uM; 640 nm-PC-3 cells, LED, 7 min, PBS) |
| **10b**[181] squaraine pentamethine | **Photophysis** ($\lambda_{abs}$ = 658 nm; DMEM), **$^1O_2$** (strong response/ 5 µM; xenon lamp-DPBF, 40 min) and **Phototox** (IC$_{50}$ = 0.124 uM; 640 nm-PC-3 cells, LED, 7 min, PBS) |
| **10c**[181] squaraine pentamethine | **Photophysis** ($\lambda_{abs}$ = 657 nm; DMEM), **$^1O_2$** (strong response/ 5 µM; xenon lamp-DPBF, 40 min) and **Phototox** (IC$_{50}$ = 0.108 uM; 640 nm-PC-3 cells, LED, 7 min, PBS) |
| **11**[182] bichromophoric pentamethine | **Photophysis** ($\lambda_{abs}/\lambda_{em}$ = 632/644 nm and $\Phi$ = 0.16; PBS), **ItraLoc** (co-localization; rhodamine 123, B16F10 cells) and **Phototox** (LC$_{50}$ < 1 µM; halogen lamp-B16F10 cells, 40 J cm$^{-2}$) |
| **12**[68,108] cyclic pentamethine | **Photophysis** ($\lambda_{abs}/\lambda_{em}$ = 493/631 nm and $\Phi$ = 0.27; phosphate buffer) and **Phototox** (IC$_{50}$ = 121 nM, loss $\Delta\Psi m$ and ROS production (CM-H$_2$DCFDA); LED-630 nm-A375 cells, 5 J cm$^{-2}$) |
| **13**[186] hemicyanine | **Photophysis** ($\lambda_{abs}/\lambda_{em}$ = 723/746 nm and $\Phi$ = 0.186; CH$_2$Cl$_2$), **$^1O_2$** ($\Phi_\Delta$ = 0.323;660nm-DPBF, 90 s, CH$_2$Cl$_2$), **ItraLoc** (P = 0.950, $\Delta\Psi m$ dependent; Mito, MCF-7 cells) and **Phototox** (>50 and 20%; 660 and 700 nm-MCF-7 cells, 15 mW cm$^{-2}$, 10 min, respectively) |
| **30a**[203] hemicyanine | **Photophysis** ($\lambda_{abs}/\lambda_{em}$ = 764/681 nm; water), **$^1O_2$** (very slow response; red LED light-ABDA, 50 mW cm$^{-2}$, 6 min, Tris buffer pH 7.4), **O$_2^{•-}$** (strong response; red LED light-DHR123, 50 mW cm$^{-2}$, 100 s, Tris buffer pH 7.4), **ROS scavenger** (vitamin C), **ItraLoc** (P = 0.55; Mito, MCF-7 cells), **PhotoTox** (~30% cell survival / 1 µM; red LED light nm-MCF-7 cells, 50 mW cm$^{-2}$, 3 min) and **DarkTox** (~70% cell survival/ 1 µM; MCF-7 cells) |
| **30b**[203] hemicyanine | **Photophysis** ($\lambda_{abs}/\lambda_{em}$ = 766/6821 nm; Tris buffer pH 7.4), **$^1O_2$** (very slow response; red LED light-ABDA, 50 mW cm$^{-2}$, 6 min, Tris buffer pH 7.4) **O$_2^{•}$** (strong response; red LED light-DHR123, 50 mW cm$^{-2}$, 100 s, Tris buffer pH 7.4), **ROS Scavenger** (vitamin C), **ItraLoc** (P = 0.88; Mito, MCF-7 cells), **PhotoTox** (~ 5% cell survival / 1µM; red LED light nm-MCF-7 cells, 50 mW cm$^{-2}$, 3 min) and **DarkTox** (~70% cell survival / 1 µM; MCF-7 cells) |
| **31**[197] hemicyanine | **Photophysis** ($\lambda_{abs}/\lambda_{em}$ = 430/700 nm and $\Phi$ = 0.041; water), **ROS** (strong and middle response; white light-DCFH-DA, 30 mW cm$^{-2}$, 3 min, DPBS, I$^-$ and PF$_6^-$ anoint, respectively), **$^1O_2$** (no response; white light-ABDA, 30 mW cm$^{-2}$, 5 min, DPBS), **O$_2^{•-}$** (strong response; white light- |

**Table 1 (continued) | Photodynamic and photothermal dyes with regard to mitochondrial targeting**

| Tested PS | Monitored quantities/properties (Obtained results/data; experimental conditions) |
|---|---|
| | DHR 123, 30 mW cm$^{-2}$, 5 min, DPBS), •OH (strong response; white light-DMPO, 30 mW cm$^{-2}$, 1 min, DPBS), **ItraLoc** (P = 0.94; Mito, Hela cells), **PhotoTox-normoxia** (IC$_{50}$ = 6 µM; white light nm-Hela cells, 25.0 J cm$^{-2}$, 20% O$_2$) and **PhotoTox-hypoxia** (IC$_{50}$ = 11.5, 9.7 and 4.8 µM; white light nm-Hela, MCF-7 and A549 cells, 25.0 J cm$^{-2}$, 1% O$_2$, respectively) |
| **32**[197] hemicyanine | **Photophysis** ($\lambda_{abs}/\lambda_{em}$ = 540/795 nm and $\Phi$ = 0.012; water) and **ROS** (slow response; white light-DCFH-DA, 30 mW cm$^{-2}$, 3 min, DPBS) |
| **33**[197] hemicyanine | **Photophysis** ($\lambda_{abs}/\lambda_{em}$ = 580/710 nm and $\Phi$ = 0.002; water) and **ROS** (slow response; white light-DCFH-DA, 30 mW cm$^{-2}$, 3 min, DPBS) |
| **Multi-functional photodynamic agents** | |
| DNA methylation PS[129] | **Other funtion:** DNA methylation<br>Activation (PDT and chemotherapy): hydrolysis |
| **14a**[129] hemicyanine | **Photophysis** ($\lambda_{abs}/\lambda_{em}$ = 680/705 nm and $\Phi$ = 0.37; phosphate buffer pH 7.4), **Phototox** (IC$_{50}$ = 4.8 µM; 660nm-DA-MB-238 cells, 45 mW cm$^{-2}$, 1 h) and **Darktox** (IC$_{50}$ = 42.3 µM; DA-MB-238 cells) |
| **14b**[129] hemicyanine | **Photophysis** ($\lambda_{abs}/\lambda_{em}$ = 688/711 nm and $\Phi$ = 0.27; phosphate buffer pH 7.4), **Phototox** (IC$_{50}$ = 2.7 µM; 660 nm-DA-MB-238 cells, 45 mW cm$^{-2}$, 1 h) and **Darktox** (IC$_{50}$ = 43.7 µM; DA-MB-238 cells) |
| **16**[128] hemicyanine | **Other function:** 5-flurouracil prodrug and NIR tumor imaging<br>Activation (Fluorescence, PDT and chemotherapy): H$_2$O$_2$-oxidation of bis boronate group |
| | **Photophysis** ($\lambda_{abs}/\lambda_{em}$ = 680/710 nm; phosphate buffer pH 7.4), **$^1$O$_2$ without H$_2$O$_2$ pretreatment** (slow response; white light-DPHA and MNAH, 50 W, 3 min, phosphate buffer pH 7.4), **$^1$O$_2$ with H$_2$O$_2$ pretreatment** (strong response; white light-DPHA and MNAH, 50 W, 3 min, phosphate buffer pH 7.4), **ItraLoc** (P = 0.950; Mito, Hela cells), **PhotoTox** (IC$_{50}$ = 9.32 and 8.15 uM; white light-Hela and HepG2 cells, respectively) and **Darktox** (IC$_{50}$ = 16.6 and 14.8 µM; Hela and HepG2 cells, respectively) |
| ruthenium(II)-cyaninecomplexes[188] | **Other function:** cytotoxicity and hypoxia targeting (type I mechanism)<br>Activation: None |
| **18a**[188] heptamethine | **Photophysis** ($\lambda_{abs}/\lambda_{em}$ = 766/796 nm, $\Phi_P$ = 0.01; acetonitrile), **$^1$O$_2$** (strong response and $\Phi_\Delta$ = 0.15; 650 nm-SOSG, 95 mW cm$^{-2}$, 20 min, PBS and acetonitrile, respectively), **O$_2$•–** (strong response; 650 nm-DHR123, 95 mW cm$^{-2}$, 20 min, PBS), **•OH** (strong response; 650 nm-HPF, 95 mW cm$^{-2}$, 20 min, PBS), **ROS scavenger** (NaN$_3$, tiron and terephthalic acid), **ItraLoc** (P = 0.76; Mito, CT-26 cells), **PhotoTox** (IC$_{50}$ = 0.17; 740 nm-CT-26 cells, 3.50 mW cm$^{-2}$, 1 h), **Darktox** (IC$_{50}$ = 18.4 µM; CT-26 cells), **PhotoTox-hypoxia** (IC$_{50}$ = 0.62; 740 nm-CT-26 cells, 3.50 mW cm$^{-2}$, 1 h) and **Darktox-hypoxia** (IC$_{50}$ = 12.3 µM; CT-26 cells) |
| **18b**[188] heptamethine | **Photophysis** ($\lambda_{abs}/\lambda_{em}$ = 7709/809 nm, $\Phi_P$ = 0.01; acetonitrile), **ItraLoc** (P = 0.67; Mito, CT-26 cells), **$^1$O$_2$** ($\Phi_\Delta$ = 0.15; 650 nm-SOSG, 95 mW cm$^{-2}$, acetonitrile, respectively), **PhotoTox** (IC$_{50}$ = 0.15; 740 nm-CT-26 cells, 3.50 mW cm$^{-2}$, 1 h) and **Darktox** (IC$_{50}$ = 1.1 µM; CT-26 cells) |
| **19**[188] heptamethine | **Photophysis** ($\lambda_{abs}/\lambda_{em}$ = 765/805 nm, $\Phi_P$ = 0.01), **$^1$O$_2$** ($\Phi_\Delta$ = 0.2; 650 nm-SOSG, 95 mW cm$^{-2}$, acetonitrile, respectively) **ItraLoc** (P = 0.67; Mito, CT-26 cells), **PhotoTox** (IC$_{50}$ = 0.154; 740 nm-CT-26 cells, 3.50 mW cm$^{-2}$, 1 h) and **Darktox** (IC$_{50}$ = 5 µM; CT-26 cells) |
| **20**[189] heptamethine | **Other function:** Porphyrin PDT and Fluorescence tumor imaging<br>Activation: None |
| | **Photophysis** ($\lambda_{abs}$)/$\lambda_{em}$ ~ (400 and 830 nm)/865 nm), **PhotoTox** (~0 and 75% cell survival / 1 µM; 655 nm and 810nm-RIF cells, 1 J cm$^{-2}$, respectively) and **I.V.** (8/10 mice were tumor free at 90th day of therapy; 3.5 µmol kg$^{-1}$, 665 nm-RIF tumor mice, 135 J cm$^{-2}$) |
| **21**[191] heptamethine | **Other function:** Cys-SH recognition, inhibition cell migration (after irradiation)<br>Activation (fluorescence): Cys-SH |
| | **Photophysis** ($\lambda_{abs}/\lambda_{em}$ ~ 650/733 nm; DMSO/PBS (1:1), pH 7.4 in presence of Cys), **$^1$O$_2$** (strong response; 660 nm-DPBF, 180 s, DCM), **ItraLoc** (P = 0.873; MitoRed, Bel-7402 cells) and **PhotoTox** (IC$_{50}$ = 3.7 µM; 660nm-A549 cells, 60 min) |
| **23**[72] hemicyanine | **Other function:** APN recognition<br>Activation (PDT and fluorescence): APN hydrolysis |
| | **Photophysis** ($\lambda_{abs}/\lambda_{em}$ ~ 685/717 nm and $\Phi$ = 0.024; phosphate buffer pH 7.4), **$^1$O$_2$** (690 nm-DPBF, 3.0 mW cm$^{-2}$, 1800 s, MeoH), **ItraLoc** (P = 0.94; Mito, HepG-2 cells), **PhotoTox** (15% viability / 2 µM; 660 nm-HepG-2 cells, 20 mW cm$^{-2}$, 10 min) and **Intratumoral** (~10% tumor mass; 100 µM, 600nm-4T1 tumor mice, 100 mW cm$^{-2}$, 20 min,) |
| **29**[73] hemicyanine | **Other function:** ALP recognition<br>Activation (PDT and fluorescence): APN hydrolysis |
| | **Photophysis** ($\lambda_{abs}/\lambda_{em}$ ~ 663/686 nm and $\Phi$ = 0.07; PBS), **$^1$O$_2$** (slow response; 660nm-ABDA, 30 mW cm$^{-2}$, 10 min, PBS), **O$_2$•–** (strong response; 660nm-DHR 123, 30 mW/cm$^{-2}$, 10 min, PBS), **•OH** (strong response; 660mn-DMPO, 30 mW cm$^{-2}$, 5 min, PBS), **ItraLoc** (P = 0.75, 032 and 0.25; Lysotracker, ER Tracker and Mito, HepG2 cells, respectively), **PhotoTox-normoxia** (~40% cell viability/ 3 µM; 660 nm-HepG-2 cells, 30 mW cm$^{-2}$, 10 min, 21% O$_2$), **PhotoTox-hypoxia** (~50% cell viability/ 4 µM; 660 nm-HepG-2 cells, 30 mW cm$^{-2}$, 10 min, 2% O$_2$) and **Intratumoral** (tumor eradation and ΔT < 5 °C; 100 µM, 100 µL, 650nm-HepG2 rhabdomyolysis mice, 0.5 W cm$^{-2}$, 10 min) |
| **48**[313] Squaraine dye pentamethine | **Other function:** COX-2 fluorescence labelling and inhibition<br>Activation (PDT and fluorescence): de-aggregation via COX-2 binding |
| | **Photophysis** ($\lambda_{abs}/\lambda_{em}$ ~ 663/686 nm and $\Phi$ = 0.10; EtOH), **$^1$O$_2$** ($\Phi_\Delta$ = 0.0057 and 0.035; 630nm-ABDA, 25 mW cm$^{-2}$, 25 min in the absence and presence of COX-2, respectively), **IntraLoc** (P = 0.96; Rho 123, HepG-2 cells) and **PhotoTox** (IC$_{50}$ = 0.59 µM; 630 nm-HepG-2 cells, 25 mW cm$^{-2}$, 12.5 min) |
| **52**[333] Bis-heptamethine | **Other function:** GSH targeting and oxidation<br>Activation (PDT and fluorescence): GSH oxidation |
| | **Photophysis** ($\lambda_{abs}/\lambda_{em}$ = 641/755 and 787/805 nm; in the absence and presence of GSH, DPBS pH 7.4, respectively) and **$^1$O$_2$** (no response and strong response; 808 nm-ABDA, 90 s, in the absence and presence of GSH, respectively) |

**Table 1 (continued) | Photodynamic and photothermal dyes with regard to mitochondrial targeting**

| Tested PS | Monitored quantities/properties (Obtained results/data; experimental conditions) |
|---|---|
| **Photothermal agents** | |
| **34**[218] hemicyanine | **Photophysis** ($\lambda_{abs}/\lambda_{em}$ = 575/606 nm and $\Phi$ = 0.029; phosphate buffer pH 7.4), **Phothermal** ($\Delta$T ~ 19.2 °C/ 0.5 mM and $\eta$ = 35.6%; 600 nm, 1.5 W cm$^{-2}$, 300 s, phosphate buffer pH 7.4), **PhotoTox** (~20% viability and $\Delta$T ~ 20 °C/ 50 ug ml$^{-1}$; 600 nm MCF-7 or U87 cell, 1.5 W cm$^{-2}$, 5 min) and **Intratumoral** (~10% tumor mass and $\Delta$T ~ 24.5 °C; 5 mg kg$^{-1}$, 600 nm- MCF-7 rhabdomyolysis mice, 1.5 W cm$^{-2}$, 5 min) |
| **35a**[219] heptamethine | **Photophysis** ($\lambda_{abs}/\lambda_{em}$ 792/788 and 755/788 nm, $\Phi$ = 0.0003 and 0.0919; phosphate buffer pH 7.4 and 2.4 respectively) and **Phothermal** ($\Delta$T ~ 21 °C / 10 µM; 750 nm, 6.0 W cm$^{-2}$, 160 s, phosphate buffer pH 7.4) |
| **35b**[219] heptamethine | **Photophysis** ($\lambda_{abs}/\lambda_{em}$ 774/784 and 761/784 nm, $\Phi$ = 0.0005 and 0.04.63; phosphate buffer pH 7.4 and 2.4 respectively), **Phothermal** ($\Delta$T ~ 37 °C/10 µM; 750 nm, 6.0 W cm$^{-2}$, 160 s, phosphate buffer pH 7.4) and **PhotoTox** (15 and 17% cell viability, $\Delta\Psi m$ loss / 20 µM ;750 nm- Hela and HepG2 cells, 6.0 W cm$^{-2}$,10 min, respectively) |
| **35c**[219] heptamethine | **Photophysis** ($\lambda_{abs}/\lambda_{em}$ 7771/780 and 756/780 nm, $\Phi$ = 0.0005 and 0.0607, phosphate buffer pH 7.4 and 2.4 respectively) and **Phothermal** ($\Delta$T ~ 35 °C/ 10 µM; 750 nm, 6.0 W cm$^{-2}$, 160 s, phosphate buffer pH 7.4) |
| **35d**[219] heptamethine | **Photophysis** ($\lambda_{abs}/\lambda_{em}$ 677/783 and 760/783 nm, $\Phi$ = 0.0001 and 0.1020, phosphate buffer pH 7.4 and 2.4 respectively) and **Phothermal** ($\Delta$T ~ 38 °C/10 µM; 750 nm, 6.0 W cm$^{-2}$, 160 s, phosphate buffer pH 7.4) |
| **36**[220] heptamethine | **Photophysis** ($\lambda_{abs}/\lambda_{em}$ ~ 580/720 and 828/850 nm, $\Phi$= 0.43 and 0.0014, phosphate buffer pH 7.4 and 2.4 respectively), **Phothermal** ($\Delta$T ~ 17 °C 20 µg ml$^{-1}$ and $\eta$ = 17.4; 808 nm, 5.0 W cm$^{-2}$, 300 s, PBS), **IntraLoc** (P = 0.94-96, $\lambda_{em}$ = 610 nm $\lambda_{ex}$ = 495 nm), $\Delta\Psi m$ dependent; Rhod 123, HeLa, A549, HepG2, and MCF-7 cells) and **PhotoTox** (~10% cell viability / 8 µg ml$^{-1}$; 808 nm - Hela cells, 1.0 W cm$^{-2}$, 5 min) |
| **44**[303] heptamethine | **Photophysis** ($\lambda_{abs}/\lambda_{em}$ ~ 780/820 nm; EtOH), **Phothermal** ($\Delta$T ~ 17 °C/ 50 µM; 808 nm, 0.5 W cm$^{-2}$, 10 min, EtOH), **PhotoTox** (~40% cell viability; 1.2 µM; 808 nm-Hela cells, 0.5 W cm$^{-2}$, 5 min) and **Darktox** (IC$_{50}$ = 10 uM; Hela cells) |
| **46**[303] heptamethine | **Photophysis** ($\lambda_{abs}/\lambda_{em}$ ~ 780/820 nm; EtOH), **Phothermal** ($\Delta$T ~ 8 °C/ 50 µM; 808 nm, 0.5 W cm$^{-2}$, 10 min, EtOH), **PhotoTox** (~ 30% cell viability; 0.3 µM; 808 nm-Hela cells, 0.5 W cm$^{-2}$, 5 min) and **Darktox** (IC$_{50}$ = 3.3 uM, Hela cells) |
| **Dual photodynamic and photothermal agents** | |
| **17**[232] hemicyanine | **Photophysis** ($\lambda_{abs}/\lambda_{em}$ = 693/714 nm and $\Phi$ = 0.01; phosphate buffer 7.4), **¹O₂** ($\Phi_\Delta$ = 0.04; 640 nm-SOSG, 300 mW cm$^{-2}$, 1 min, EtOH), **Phothermal** ($\Delta$T ~ 17 °C/2 mg ml$^{-1}$ and $\eta$ = 50; 808 nm, 1 W cm$^{-2}$, 10 min, DMSO), **PhotoTox-640nm** (14.8% viability /10 µg ml$^{-1}$; 640 nm - SW480 cells, 0.3 W cm$^{-2}$, 5 min), **PhotoTox-808nm** (62.3% viability / 10 µg ml$^{-1}$; 800 nm-SW480 cells, 1.0 W cm$^{-2}$, 5 min) and **PhotoTox-640 and 808 nm** (14.4% cell viability / 10 µg ml$^{-1}$; 640 and 800 nm-SW480 cells, 0.3 and 1.0 W cm$^{-2}$, 3 min, respectively) |
| **39 g**[233], heptamethine | **Photophysis** ($\lambda_{abs}/\lambda_{em}$ = 781/799 nm and $\Phi$ = 0.039; MeOH), **¹O₂** (strong response; 808 nm-SOSG, 1.5 W cm$^{-2}$, 5 min, water), **Phothermal** ($\Delta$T ~ 17 °C / 10 µM; 808 nm, 1.5 W cm$^{-2}$, 5 min), **PhotoTox-PDT** (67.4% cell viability / 12 µM; 808 nm-A549 cells, 1.5 W cm$^{-2}$, 5 min, under ice), **PhotoTox-PTT** (29.3% cell viability 12 µM; 808 nm-A549 cells, 1.5 W cm$^{-2}$, 5 min, N-acetylcysteine supplement), **PhotoTox-PDT and PTT** ( ~ 5% cell viability; 12 µM; 808 nm-A549 cells, 1.5 W cm$^{-2}$, 5 min) and **I.V.** (tumor eradiation and 100% OS (60 day); 5 mg kg$^{-1}$, 808 nm-4T1 tumor mice, 800 mW cm$^{-2}$, 5 min,) |
| **40**[234] heptamethine | **Photophysis** ($\lambda_{abs}/\lambda_{em}$ = 650/NIR nm, water), **¹O₂** (slow response; 660 nm-DPBF, 1.5 W cm$^{-2}$, 5 min, water), **Phothermal** ($\Delta$T ~ 14 °C / 50 µM; 660 nm, 1.0 W cm$^{-2}$, 5 min, water) and **I.V.** (reduction in tumor volume and mass on the half, $\Delta$T ~ 15 °C and improvement in OS; 2 µmol kg$^{-1}$, 660 nm-4T1 tumor mice, 1.0 W cm$^{-2}$, 5 min) |
| **42a**[235] heptamethine | **Photophysis** ($\lambda_{abs}/\lambda_{em}$ ~ 700/760 nm; water), **¹O₂** (strong response; 750 nm-SOSG, 1.0 W cm$^{-2}$, 5 min, water) and **Phothermal** ($\Delta$T ~ 21 °C / 50 µM; 750 nm, 1.0 W cm$^{-2}$, 5 min, water) |
| **42b**[235] heptamethine | **Photophysis** ($\lambda_{abs}/\lambda_{em}$ = 696/764 nm, water), **¹O₂** (strong response; 750 nm-SOSG, 1.0 W cm$^{-2}$, 5 min, water) and **Phothermal** ($\Delta$T ~ 25 °C / 50 µM; 750 nm, 1.0 W cm$^{-2}$, 5 min, water) |
| **47**[304] Bis-heptamethine | **Photophysis** ($\lambda_{abs}/\lambda_{em}$ ~ 701/828 nm and $\Phi$ = 0.05; water), **¹O₂** (strong response; 808nm-SOSG, 0.3 W cm$^{-2}$, 5 min, DMSO), **Phothermal** ($\Delta$T ~ 12 °C / 50 µM; 808 nm, 0.3 W cm$^{-2}$, 5 min, DMSO), **PhotoTox-PDT** (5% cell viability / 5 µM; 808 nm-Hela cells, 0.3 W cm$^{-2}$, 5 min, under ice), **PhotoTox-PTT** (52% cell viability/ 5 µM; 808 nm-Hela cells, 0.3 W cm$^{-2}$, 5 mi, N-acetylcysteine supplement), **PhotoTox-PDT and PTT** (9% cell viability; 5 µM; 808 nm-Hela cells, 0.3 W cm$^{-2}$, 5 min) and **I.V.** (tumor eradiation, $\Delta$T ~ 17 °C; 0.8 mg kg$^{-1}$, 2×808 nm-4T1 tumor mice (24 and 48 h after application), 0.8 W cm$^{-2}$, 5 min) |
| **49a**[314] pentamethine | **Photophysis** ($\lambda_{abs}/\lambda_{em}$ = 637/657 nm and $\Phi$ = 0.09; water), **¹O₂** (middle response; 65 nm-SOSG, 0.3 W cm$^{-2}$, 5 min, EtOH), **Phothermal** ($\Delta$T ~ 25 °C / 100 µmg ml$^{-1}$; 655 nm, 0.3 W cm$^{-2}$, 10 min, water) and **PhotoTox** (60 and 34% cell viability/ 20 and 100 µg ml$^{-1}$; 655 nm-4T1 cells, 2 W cm$^{-2}$, 5 min) |
| **49b**[314] pentamethine | **Photophysis** ($\lambda_{abs}/\lambda_{em}$ = 647/660 nm and $\Phi$ = 0.09; water), **¹O₂** (middle response; 65 nm-SOSG, 0.3 W cm$^{-2}$, 5 min, EtOH), **Phothermal** ($\Delta$T ~ 25 °C / 100 µmg/ml; 655 nm, 0.3 W cm$^{-2}$, 10 min, water) and **PhotoTox** (71 and 40% cell viability/ 20 and 100 µg ml$^{-1}$; 655 nm-4T1 cells, 2 W cm$^{-2}$, 5 min) |
| **49c**[314] pentamethine | **Photophysis** ($\lambda_{abs}/\lambda_{em}$ = 668/(655, 720 and 796 nm) and $\Phi$ = 0.09; water), (strong response; 65 nm-SOSG, 0.3 W cm$^{-2}$, 5 min, EtOH), **Phothermal** ($\Delta$T ~ 17 °C / 100 µmg ml$^{-1}$; 655 nm, 0.3 W cm$^{-2}$, 10 min, water), and **PhotoTox** (45 and 23% cell viability / 20 and 100 µg ml$^{-1}$; 655nm-4T1 cells, 2 W cm$^{-2}$, 5 min) and **Intratumorally** (<5% tumor mass; 200 µg ml$^{-1}$, 100 µl, 655 nm-4T1 tumor mice, 0.3 W cm$^{-2}$, 5 min) |
| **Multi-functional photodynamic and photothermal agents** | |
| **33**[295] heptamethine | **Other function:** Radiosenzation <br> Activation: None <br><br> **Photophysis** ($\lambda_{abs}/\lambda_{em}$ = 782/810 nm; MeOH), **¹O₂** (strong response, O₂ level dependent; 808 nm-SOSG, 2.0 W cm$^{-2}$, 5 min, PBS), **Phothermal** ($\Delta$T ~ 29.4 °C / 2.5 µM, $\eta$ = 54.1%; 807 nm, 2.0 W cm$^{-2}$, 5 min, PBS), **IntraLoc** (P = 0.981,Mito-tracker probe, LLC cells), **PhotoTox** (< 50% cell viability/ 2.5 µM; 808 nm - LLC cells, 2 W cm$^{-2}$, 5 min) and **I.V.** (decrease in Ki-67 positive cell, $\Delta$T ~ 32 °C, tumor erudition and increase in OS; 2.5 mg kg$^{-1}$, 808 nm-LLC tumor mice, 0.8 W cm$^{-2}$, 5 min, radiotherapy) |

$\Phi$ fluorescence quantum yield, $\Phi_\Delta$ ¹O₂ quantum yield, $\Phi_P$ phosphorescence quantum yield, $\lambda_{abs}$ wavelength of absorption maximum, $\lambda_{em}$ wavelength of emission maximum, $\eta$ light-to-heat conversion, *ABDA* 9,10-anthracenediyl-bis(methylene)-dimalonic acid (¹O₂ probe), *CM-H₂DCFDA* chloromethyl-2′,7′-dichlorodihydrofluorescein (ROS probe), *DarkTox* dark toxicity, *DCFH-DA* 2′,7′-dichlorodihydrofluorescein diacetate (ROS probe), *DMPO* 5,5-dimethyl-1-pyrroline-N-oxide (•OH Probe); *DPBS, PBS with 1% DMSO* DHR123, dihydrorhodamine 123 (O₂•⁻ probe), *DPHA* 9,10-diphenylanthracene (¹O₂ probe), *ItraLoc* intracellular localization, *HPF* hydroxyphenyl fluorescein (•OH probe), *I.V.* intravenous, *Mito* Mito-Tracker Green FM, *MNAH* ¹O₂ probe, *¹O₂* singlet oxygen, *O₂•⁻* superoxide anion, *•OH* hydroxyl radical, *Rho 123* rhodamine 123, *SOSG Singlet Oxygen Sensor Green* (¹O₂ probe), *DPBF* 1,3-diphenylisobenzofuran (¹O₂ probe), *Photophys* photophysical properties, *Phothermal* photothermal efficiency, *Phototox* phototoxicity.

a n.d. value is too low for the determination.

b The effect corresponds to the presented concentration of the cyanine dye.

potent sonodynamic agents. In addition, some of them display strong affinity to serum albumin[108,160], at least in the combination with suitable drug delivery systems, they could represent a promising structural motif for novel sonodynamic agents. Nevertheless, the photophysical properties of cyanine dyes can be significantly affected by an interaction with the target structure such as the inner mitochondrial membrane. On the other hand, an anionic membrane can decrease the aggregation of cyanine dyes and thus could reduce their phototoxicity, although hydrophobic cations can form dimer forms (at higher concentrations of hydrophobic cation, or higher $\Delta\Psi m$) in the inner mitochondrial membrane[122,172]. However, the viscosity of the inner mitochondrial membrane is strongly higher than that of the extra-matrix surroundings and the mitochondrial matrix[367]. It was reported that non-radiative transitions of cyanine dyes can be significantly decreased and fluorescence quantum yield is increased in more viscous surroundings. It could be expected that in the case of more rigid pentamenthine dyes, this effect may not be as strong as in the case of more flexible heptamethine dyes. It should be also mentioned that penthamethinium usually displays an absorbance maximum at about 650 nm, whereas heptamethinium at about 800 nm[110]. Irradiation with 808 nm have deeper penetration into tissue than irradiation with 650 nm[368]. Sonodynamic therapy shows a potent synergic effect with PDT and PTT, and hepthamethines could have higher therapeutic potential. However, even in this case, light penetration is no greater than 1 cm[368]. A combination of sonodynamic and cytostatic therapy might be more effective for reaching deeper localized tumor tissues. Nevertheless, it can be said that penthamethinium can represent a suitable structural motif for the design of novel sonosensitizers. Briefly, in addition to the position of absorption maximum, other properties (e.g., aggregation and interaction with mitochondrial structure), or the type of PDT mechanism should be considered. In certain tumor types, such as head and neck carcinoma, it has been documented that the hypoxic volume experiences a notable increase in correlation with the primary tumor volume[369]. It should be mentioned, that hypoxia is associated with resistance against sonosensitizers (e.g., heptacyanine **41** (IR-780))[370]. For instance, the application of type I photosensitizers like suitable hemicyanines (**29, 30b** and **31**)[73,197,203] may enable the effective targeting of hypoxic tumors or their hypoxic regions. However, this final statement remains unverified experimentally, and unfortunately, sonosensitizers based on this structural motif have not yet been prepared and tested.

## Conclusion

Mitochondria play an important role in the cancer biology and represent very important target for the anticancer therapy. On the other hand, they are vulnerable against PDT and PTT. In conclusion, cyanine dyes present a promising avenue for enhancing the efficacy of PDT and PTT in cancer treatment. Their ability to target mitochondria and their high selectivity for tumor tissues make them potent agents in disrupting cancer cell viability and metastatic potential. However, the antitumor efficacy of these low molecular compounds can be limited by their selectivity and non-toxicity without irradiation. Strategies to improve their therapeutic efficiency, such as the development of suitable drug delivery systems like nanoparticles and the conjugation with cytotoxic agents, have shown significant promise. The studies reviewed indicate that enhancing the accumulation of cyanine dyes in tumor tissues and their activation specifically within cancer cells can lead to improved therapeutic outcomes. Furthermore, the exploration of type I photosensitizers and their combination with perfluorinated delivery systems to enhance oxygen transport presents new opportunities for overcoming the limitations of type II PDT, particularly in hypoxic tumor environments. Additionally, the combination of PDT and PTT has demonstrated a synergistic effect in preclinical models, suggesting a potential for more effective cancer treatment protocols. However, the challenge of light penetration in tissues remains a significant barrier, emphasizing the need for continued research into sonodynamic therapy and the design of novel sonosensitizers that can target deeper tumor tissues. Overall, while the potential of cyanine dyes in cancer therapy is substantial, further experimental verification and systematic studies are essential to optimize their use and fully realize their therapeutic benefits. Future research should focus on refining these compounds, improving their selectivity, and exploring innovative delivery mechanisms to enhance their clinical applicability.

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

## Acknowledgements

This work was supported by projects of Charles University in Prague [SVV260637; SVV260521; UNCE 204064; Progres LF1 Q38 and Q27, Cooperatio ONCO]. The work was also supported by the Ministry of Education, Youth, and Sports grants no. LM2023053 (EATRIS-CZ) and the Technology Agency of the Czech Republic within projects TN02000109. The work was also supported by the Ministry of Health grants nos. RVO-VFN 64165, NU22-08-00160 and NU21-08-00407. We are also grateful for the support from project National Institute for Cancer Research (Programme EXCELES, ID Project No. LX22NPO5102) - funded by the European- Union - Next Generation EU. We also thank for the support Ministry of Interior grant nos. VB02000056. This work was also supported by the National Institute for Neurological Research (Programme EXCELES, ID Project No. LX22NPO5107) funded by the European Union—Next Generation EU. This work has also received support from the Masaryk University Foundation (Grant No. MUNI/A/1587/2023). Additionally, we would also like to thank Dr. Tomáš Bříza and Prof. Vladimír Král, who played an irreplaceable role in the design and synthesis of many polymethinium salts, including photoactive dyes. We honor their memory.

## Author contributions

Zdeněk Kejík: writing original draft, supervision, and reviewing the draft. Jan Hajduch: writing original draft. Nikita Abramenko: writing original draft. Frédéric Vellieux: writing original draft. Kateřina Veselá: writing original draft. Jindřiška Leischner Fialová: writing original draft. Kateřina Petrláková: writing original draft. Kateřina Kučnirová: writing original draft. Robert Kaplánek: writing original draft. Ameneh Tatar: writing original draft. Markéta Skaličková: writing original draft. Michal Masařík: Conceptualization, reviewing the draft. Petr Babula: Conceptualization. Petr Dytrych: writing original draft. David Hoskovec: reviewing the draft. Pavel Martasek: Reviewing the draft. Milan Jakubek: Conceptualization, supervision, and reviewing the draft.

## Competing interests

The authors declare no competing interest.
