## [Peer Review File · Communications Chemistry]

Reviewers' comments:

Reviewer #1 (Remarks to the Author):

This review article summarizes the mitochondria targeted cyanine-based photosensitizers, which are shown to offer improved therapeutic outcome, by highlighting the importance of mitochondria as a target in phototherapy applications. The topic is quite important and attractive. The manuscript is well-designed and it covers critical basic principles along with recent examples. However, it still requires a major revision before considering publication.

- 1) The language needs to be improved. There are many typos and some sentences are hard to follow.
- 2) Currently, activity-based PSs and type I PSs are highly popular in the field of PDT. A small section can be devoted to discuss the design principles of activity-based mitochondria targeted cyanines. Additionally, recent cyanine-based type I PSs can be included in the manuscript.
- 3) For most of the examples, the structure-activity relationship discussion needs to be improved.
- 4) A clear outlook should be given for the future designs. Drawbacks of the current PSs need to be listed clearly.
- 5) Following papers can be added as additional examples to the proper sections.
doi.org/10.1002/adhm.202303432, doi.org/10.1021/acs.bioconjchem.3c00096
- 6) A table is needed to summarize the mode of action (PDT, PTT or dual), application (in vitro/in vivo), photophysical properties and type of ROS etc..

Reviewer #2 (Remarks to the Author):

In this review paper, authors reviewed a large number of references to thoroughly describe the state-of-the-art of synthesis and applications of cyanine dyes for photodynamic and photothermal applications with a focus on mitochondria targeting. Still, a revision is needed to improve the manuscript's quality and consider the publication of the manuscript. The general issues are:

1. The introduction consists of repeating sentences listing the chapters. This can be more efficiently and clearly done by using a table of contents.
2. The second paragraph of the introduction (page 1, line 35), which tells about the mitochondrial role, can be a subsection entitled Mitochondria role in biological processes or similar.
3. Page 2, line 51, sentence: "African American patient have the worst therapeutic prognosis (higher cancer mortality rates and shorter survival times) and higher mitochondrial activity compared to European American patients"... Is this true for all cancer types?
4. Page 2, line 76, unclear sentence: "It is well known, that their anticancer is partially caused by higher ROS activity."
5. Page 2, line 77, sentence: "On the other hand, higher ROS levels in cancer cells make them more sensitive to the ROS-induction treatment." Here it is stated that higher ROS production means that cells are more sensitive to treatment, but a few sentences above (line 60) it is written that more ROS means higher invasiveness of cancer and positive effect on cancer cell viability. How is this connected?
6. Page 3, line 91: please define the abbreviation "PS" when first used.
7. Page 3, line 89, in addition to the first biological window, mention second biological window as well.
8. Page 3, line 107, sentence "It should be noted that some cyanine dyes display selective localization in the inner mitochondrial membrane". Can you comment the localization in other parts of the cell? What about endocytosis and accumulation in endosomes/lysosomes?
9. Page 4, line 118: In case of a temperature increase that is too high, shock proteins are activated. Please mention these, because they can strongly affect the outcome of the treatment. Also, discuss the optimal elevated temperature for the treatment, i.e mild hyperthermia vs. hyperthermia vs. thermal ablation. Usually, thermal ablation is not the desired result.
10. Page 5, line 159, sentence: "As some of them show significant selectivity for tumour tissue and cytoselective effect against cancer cells, they are intensively studied as anticancer agents.". can you comment more on how they selectively target cancer cells?
11. Section 3.1. can you comment on the possible internalization mechanisms and how cyanine

dyes end up in mitochondria? Comment the difference between normal and cancer cells. You mentioned the difference in mitochondrial membrane potential, but before dyes come to the mitochondria, they must overcome several other barriers, determining the final yield in mitochondria.

12. Page 6, line 207: IR-61 showed to have a protective function for mitochondria. Why this and why not any others?

13. Page 6, line 226: "Mitochondria might represent a more suitable organelle than lysosome and endoplasmic reticulum in the case of PDT-stimulated immunogenic cell death. [91] Although the effect of lysosomal and endoplasmic reticulum-PS on tumour mass can be comparable to mitochondrial PS". Why is it more beneficial to target mitochondria if targeting lysosomes is easier and you state that the effect is similar? Please comment on that.

14. Page 7, line 154: ". A synergic effect has also been observed for the combination of mitochondrial and lysosomal targeted PDT. [104-108]". Combination is better, but which one is better as a single therapy?

15. General comment on section 4. You nicely presented a huge literature survey and many different cyanine dyes. However, all of them were tested under different conditions, so it would be beneficial to propose a systematic study of different dyes, tested under the same condition to observe and compare their therapeutic and diagnostic effect.

16. Chapter 8: Target transport system. Please rephrase into the delivery system

17. Chapter 8: you mention only silica, sulphide and phosphate nanoparticles. What about other inorganic and organic nanostructures?

We would like to thank the reviewers and the editor for their thorough reading and reviewing of our manuscript. We greatly appreciate their insightful remarks, which have significantly improved our work. We have carefully taken the reviewer's advice and comments into account, addressing each point carefully. In consideration of the reviewer's recommendation implemented in the manuscript, the conclusion has been correspondingly adjusted. The following changes and corrections have been made in the revised manuscript:

Reviewer #1 (Remarks to the Author):

This review article summarizes the mitochondria targeted cyanine-based photosensitizers, which are shown to offer improved therapeutic outcome, by highlighting the importance of mitochondria as a target in phototherapy applications. The topic is quite important and attractive. The manuscript is well-designed and it covers critical basic principles along with recent examples. However, it still requires a major revision before considering publication.

1) The language needs to be improved. There are many typos and some sentences are hard to follow.

- ✓ Thank you for your valuable feedback. The language throughout the manuscript has been thoroughly reviewed and corrected to address typos and improve readability.

2) Currently, activity-based PSs and type I PSs are highly popular in the field of PDT. A small section can be devoted to discuss the design principles of activity-based mitochondria targeted cyanines. Additionally, recent cyanine-based type I PSs can be included in the manuscript.

- ✓ A presentation of the basic principles of type I PSs, along with relevant examples of cyanine-based PSs, has been added to subchapter 4.1.2, titled 'Cyanine Dyes for Type I PDT'. Furthermore, the discussion on transition metal cation nanoaggregates of type I heptamethinium PS is provided in subchapter 8.1. (**page 34, line 1063-1077**)

3) For most of the examples, the structure-activity relationship discussion needs to be improved.

- ✓ Each presented structural formula of the cyanine dyes has been assigned a unique identifier corresponding to their order in the text. This has significantly clarified and improved the discussion on structure-activity relationships.
- ✓ To aid comprehension, the chemical structures of the cyanine dyes under consideration will be collectively presented (**Fig. 5, Fig. 6, Fig. 11, Fig. 13, Fig. 15, Fig. 19. and Fig. 20.**) at the beginning of the relevant chapters and subsections.

4) A clear outlook should be given for the future designs. Drawbacks of the current PSs need to be listed clearly.

- ✓ Future Design Outlook: The chapter "Future Directions" has been supplemented with an outlook on future design strategies (**page 48 and 49, line 1429-1461; page 49, line 1483-1484 and page 50, line 1513-1521**).

Applicability of Cyanine PSs: We have included a discussion on the applicability of cyanine PSs, focusing on their selectivity for tumors and their effectiveness in hypoxic tumor environments

Design Strategies: This section now delineates strategies for designing multifunctional photodynamic agents, highly mitochondrial selective PSs, and type I PSs.

5) Following papers can be added as additional examples to the proper sections.
doi.org/10.1002/adhm.202303432, doi.org/10.1021/acs.bioconjchem.3c00096

- ✓ doi.org/10.1002/adhm.202303432: This paper (ref. 365) is discussed in the "Future Directions" chapter in connection with the influence of aromatic substitution of cyanine dyes on their photodynamic and photothermal efficiency (**page 49, line 1493-1494**).
- ✓ doi.org/10.1021/acs.bioconjchem.3c00096: This paper (ref. 352) is presented in Chapter 8.5 (**page 41, line 1365-1379**) as an example of inorganic nanoparticles.

6) A table is needed to summarize the mode of action (PDT, PTT or dual), application (in vitro/in vivo), photophysical properties and type of ROS etc..

- ✓ Thank you for the suggestion. We have included a table in the "Future Directions" chapter (**page 41, line 1414**). This table summarizes the photophysical properties, photodynamic and photothermal efficiency, and *in vitro* and *in vivo* studies of the discussed cyanine dyes, as well as their modes of action (PDT, PTT, or dual) and types of reactive oxygen species (ROS).

Reviewer #2 (Remarks to the Author):

In this review paper, authors reviewed a large number of references to thoroughly describe the state-of-the-art of synthesis and applications of cyanine dyes for photodynamic and photothermal applications with a focus on mitochondria targeting. Still, a revision is needed to improve the manuscript's quality and consider the publication of the manuscript. The general issues are:

1) The introduction consists of repeating sentences listing the chapters. This can be more efficiently and clearly done by using a table of contents.

- ✓ Thank you for your valuable recommendation. In the current version of the manuscript, we have included a table of contents at the end of the introduction to address this issue.

2) The second paragraph of the introduction (page 1, line 35), which tells about the mitochondrial role, can be a subsection entitled Mitochondria role in biological processes or similar.

- ✓ Thank you for your suggestion. We have created a new subsection entitled "The role of mitochondria in biological processes" to present the content from the second paragraph of the introduction.
- ✓ In alignment with the journal's guideline, a new s introduction has been written.

3) Page 2, line 51, sentence: African American patient have the worst therapeutic prognosis (higher cancer mortality rates and shorter survival times) and higher mitochondrial activity compared to European American patients Is this true for all cancer types?

- ✓ The sentence (**page 2 and 3, line 84-86**) has been edited to clarify that the study was conducted across multiple cancer types (referred to as a pan-cancer study). The revised sentence is: "Additionally, higher mitochondrial activity and mitochondrial biogenesis in African American patients correlate with worse therapeutic prognoses (higher cancer mortality rates and shorter survival times) across multiple cancer types compared to European American patients with lower mitochondrial activity."
- ✓ More detailed this difference is described **on the page 3, line 88-97**.

4) Page 2, line 76, unclear sentence: It is well known, that their anticancer is partially caused by higher ROS activity.

Thank you for pointing out the unclear sentence. To improve cohesion with the previous sentences, we have revised and relocated the sentence (**page 3, line 112-113**) as follows: "Many anticancer drugs exploit ROS-induced cell death as their mechanism of action."

5) Page 2, line 77, sentence: On the other hand, higher ROS levels in cancer cells make them more sensitive to the ROS-induction treatment. Here it is stated that higher ROS production means that cells are more sensitive to treatment, but a few sentences above (line 60) it is written that more ROS means higher invasiveness of cancer and positive effect on cancer cell viability. How is this connected?

- ✓ A short paragraph (**page 3, line 104-112**) summarizing the dual role of ROS in cancer has been placed in this chapter, discussing both aspects and introducing the concept of why cancer cells are more susceptible to ROS generation induced by anticancer drugs. In essence, cancer cells necessitate elevated ROS levels compared to normal cells. However, excessively high ROS levels induced by certain therapies such as PDT can effectively eliminate cancer cells.

6) Page 3, line 91: please define the abbreviation PS when first used.

- ✓ We have reviewed all abbreviations in the manuscript and made corrections where necessary. Specifically, the abbreviation "PS" (Photosensitizer) has been defined in the last paragraph (**page 1, line 23**) of the introduction to ensure clarity for the readers.

7) Page 3, line 89, in addition to the first biological window, mention second biological window as well.

- ✓ Thank you for your suggestion. We have expanded the discussion on biological windows as follows: "In addition to the first biological window, photosensitizers (PS) used for PDT can also be irradiated within the second biological window (1000–1350 nm) and the third biological window (1550–1850 nm)." **Page 4, line 143-145**

8) Page 3, line 107, sentence It should be noted that some cyanine dyes display selective localization in the inner mitochondrial membrane. Can you comment the localization in other parts of the cell? What about endocytosis and accumulation in endosomes/lysosomes?

- ✓ Thank you for raising this excellent point. Indeed, it is accurate to note that cyanine dyes exhibit lysosomal co-localization/localization (**page 5, line 172-183**). Furthermore, cyanine dyes (in the form of nanoparticles) can enter cells through endocytosis, with mitochondrial localization becoming evident only over an extended period. Notably, studies indicate that mitochondrial PDT may be more efficacious than lysosomal PDT. While this subject is acknowledged in the current manuscript, a more comprehensive exploration of the correlation between photosensitizer intracellular localization and its effectiveness is extensively elucidated by Wang et al. (ref. 69) in their seminal works. For a more thorough understanding, we recommend referring to their research.

9) Page 4, line 118: In case of a temperature increase that is too high, shock proteins are activated. Please mention these, because they can strongly affect the outcome of the treatment. Also, discuss the optimal elevated temperature for the treatment, i.e. mild hyperthermia vs. hyperthermia vs. thermal ablation. Usually, thermal ablation is not the desired result.

- ✓ We agree that this topic was discussed too briefly in the previous version of the manuscript. In the current version, we have expanded the discussion on the relationship between temperature and PTT efficiency (**page 5 and 6, line 188-211**). Specifically, we have included the following points:

Heat Shock Proteins (HSPs): We now mention the activation of heat shock proteins (HSPs) when the temperature increase is too high. HSPs can strongly affect the outcome of the treatment by helping cancer cells survive thermal stress.

Optimal Temperature for Treatment: We have discussed the optimal elevated temperatures for treatment, distinguishing between mild hyperthermia, hyperthermia, and thermal ablation.

Balancing Temperature and Efficiency: Balancing the PTT temperature for effective tumor destruction while minimizing harm to healthy tissues is crucial. This balance is essential to maximize the therapeutic efficacy and minimize side effects.

10) Page 5, line 159, sentence: As some of them show significant selectivity for tumor tissue and cytospecific effect against cancer cells, they are intensively studied as anticancer agents. can you comment more on how they selectively target cancer cells?

- ✓ We have expanded the discussion to explain how certain dyes selectively target cancer cells (**page 7, line 251-261**): Some dyes, particularly heptamethines display a strong affinity for serum albumin, which is known to accumulate in tumor tissue. Additionally, oncogenic transformation is associated with high levels of anionic phospholipids, which are typical binding partners for cationic cyanine dyes. This affinity for serum albumin and anionic phospholipids contributes to the selective targeting of cancer cells by these dyes.

11. Section 3.1. can you comment on the possible internalization mechanisms and how cyanine dyes end up in mitochondria? Comment the difference between normal and cancer cells. You mentioned the difference in mitochondrial membrane potential, but before dyes come to the mitochondria, they must overcome several other barriers, determining the final yield in mitochondria.

- ✓ We have expanded Section 3.1 to discuss the possible internalization mechanisms and how cyanine dyes end up in mitochondria, including the differences between normal and cancer cells. The revised section (**page 8, line 282-286**) includes the following points:
Since cyanine dyes are lipophilic and cationic, and cellular membrane of cancer cells contain higher level of anionic phospholipids than normal cells. Membrane cancer cells can be more attractive for normal cells and passive diffuse could be explain their uptake. Nevertheless, existence of other mechanism such as endocytosis (especially at higher concentration) cannot be excluded.
In summary, while passive diffusion plays a significant role in the uptake of cyanine dyes due to their lipophilic and cationic nature, other mechanisms such as endocytosis may also contribute, particularly at higher concentrations. The selective targeting of cancer cells is facilitated by the differences in membrane composition and mitochondrial membrane potential between normal and cancer cells.

12) Page 6, line 207: IR-61 showed to have a protective function for mitochondria. Why this and why not any others?

- ✓ In the investigated mice model of diabetes, the efficacy of IR-61 was linked to the activation of NRF2. It is well-established that NRF2 plays a pivotal role in the regulation of mitochondrial homeostasis, including the induction of processes such as mitophagy. The suppression of mitophagy is a critical aspect of the pathogenesis of type 2 diabetes. Consistent with the proposed hypothesis, mitophagy induction was observed for the heptamethine dye dc-IR825. Nevertheless, the current understanding of this phenomenon remains limited, necessitating further research to provide more precise and detailed elucidation. -**page 9, line 321-328**

13) Page 6, line 226: Mitochondria might represent a more suitable organelle than lysosome and endoplasmic reticulum in the case of PDT-stimulated immunogenic cell death. [91] Although the effect of lysosomal and endoplasmic reticulum-PS on tumor mass can be comparable to mitochondrial PS.

Why is it more beneficial to target mitochondria if targeting lysosomes is easier and you state that the effect is similar? Please comment on that.

- ✓ In the latest version of the manuscript, this topic was briefly addressed (**page 9, line 340-357**). The previous sentence focused solely on the effect of PDT on the primary tumor. However, when considering distant tumors and immune system stimulation, a significant difference was observed. Possible mechanisms, as discussed in the current version of the manuscript, suggest that mitochondrial ROS may not only induce mitochondria-dependent immunogenic cell death (ICD) but also endoplasmic reticulum-dependent ICD. Building upon this, it is hypothesized that mitochondria present an appealing target for PDT-induced ICD.

14) Page 7, line 154: A synergic effect has also been observed for the combination of mitochondrial and lysosomal targeted PDT. [104-108]. Combination is better, but which one is better as a single therapy?

- ✓ In the current version of the manuscript, this topic is elaborated upon in the aforementioned sentence (**page 10, line 380-387**). Examples (ref. 139, 70 and 71) provided in the discussion indicate that PDT targeted at the mitochondria exhibits greater cytotoxicity compared to when targeting the lysosomes. Therefore, while a combination of mitochondrial and lysosomal targeted PDT shows a synergistic effect, mitochondrial-targeted PDT is more effective as a single therapy in terms of cytotoxicity.

15) General comment on section 4. You nicely presented a huge literature survey and many different cyanine dyes. However, all of them were tested under different conditions, so it would be beneficial to propose a systematic study of different dyes, tested under the same condition to observe and compare their therapeutic and diagnostic effect.

- ✓ We totally agree with this suggestion. The validity of this statement is indeed broader and applicable to all presented photoactive cyanine dye. Therefore, this statement has been inserted below Table 1 (**page 42, line 1409-1412**), which summarizes properties and applications of photoactive dyes.
We acknowledge the need for a systematic study of different cyanine dyes tested under the same conditions to accurately observe and compare their therapeutic and diagnostic effects. Such a comparative analysis would provide more reliable insights into their relative efficacies and potential clinical applications.

16) Chapter 8: Target transport system. Please rephrase into the delivery system

- ✓ Thank you for your suggestion. The phrase "Target transport system" has been rephrased to "delivery system" in the current version of the manuscript.

17) Chapter 8: you mention only silica, sulphide and phosphate nanoparticles. What about other inorganic and organic nanostructures?

- ✓ This topic has been discussed more comprehensively in the current version of the manuscript. In the part of Chapter 8, we have also briefly introduced various supramolecular nanosystems, including lipid, polymeric nanoparticles (**page 29 and 30, line 980-981**), and self-assembly nanoparticles (**page 29, line 972 and 974**). In addition, complexes of cyanine dyes with serum proteins (in nano size), such as serum albumin, are also mentioned (**page 30, line 985-987**).

REVIEWERS' COMMENTS:

Reviewer #1 (Remarks to the Author):

The authors have addressed all of my concerns. The manuscript can be accepted as is.

Reviewer #2 (Remarks to the Author):

The authors have addressed all previously pointed-out issues in the manuscript. Each comment and suggestion has been carefully considered and incorporated, resulting in significant improvements. The manuscript has been refined to meet the journal's standards, and it is now ready for publication.